# Differentiable Causal Discovery
# from Interventional Data

**Philippe Brouillard**[*]
Mila, Université de Montréal

**Sébastien Lachapelle**[*]
Mila, Université de Montréal

**Alexandre Lacoste**
Element AI

**Simon Lacoste-Julien**
Mila, Université de Montréal
Canada CIFAR AI Chair

**Alexandre Drouin**
Element AI

## Abstract

Learning a causal directed acyclic graph from data is a challenging task that involves solving a combinatorial problem for which the solution is not always identifiable. A new line of work reformulates this problem as a continuous constrained optimization one, which is solved via the augmented Lagrangian method. However, most methods based on this idea do not make use of interventional data, which can significantly alleviate identifiability issues. This work constitutes a new step in this direction by proposing a theoretically-grounded method based on neural networks that can leverage interventional data. We illustrate the flexibility of the continuous-constrained framework by taking advantage of expressive neural architectures such as normalizing flows. We show that our approach compares favorably to the state of the art in a variety of settings, including perfect and imperfect interventions for which the targeted nodes may even be unknown.

## 1 Introduction

The inference of causal relationships is a problem of fundamental interest in science. In all fields of research, experiments are systematically performed with the goal of elucidating the underlying causal dynamics of systems. This quest for causality is motivated by the desire to take actions that induce a controlled change in a system. Achieving this requires to answer questions, such as "what would be the impact on the system if this variable were changed from value $x$ to $y$?", which cannot be answered without causal knowledge [33].

In this work, we address the problem of data-driven causal discovery [16]. Our goal is to design an algorithm that can automatically discover causal relationships from data. More formally, we aim to learn a *causal graphical model* (CGM) [36], which consists of a joint distribution coupled with a directed acyclic graph (DAG), where edges indicate direct causal relationships. Achieving this based on observational data alone is challenging since, under the faithfulness assumption, the true DAG is only identifiable up to a *Markov equivalence class* [46]. Fortunately, identifiability can be improved by considering interventional data, i.e., the outcome of some experiments. In this case, the DAG is identifiable up to an *interventional Markov equivalence class*, which is a subset of the Markov equivalence class [48, 15], and, when observing enough interventions [9, 11], the DAG is exactly identifiable. In practice, it may be possible for domain experts to collect such interventional data, resulting in clear gains in identifiability. For instance, in genomics, recent advances in gene editing technologies have given rise to high-throughput methods for interventional gene expression data [6].

---

[*] Equal contribution.
Correspondence to: {philippe.brouillard, sebastien.lachapelle}@umontreal.ca

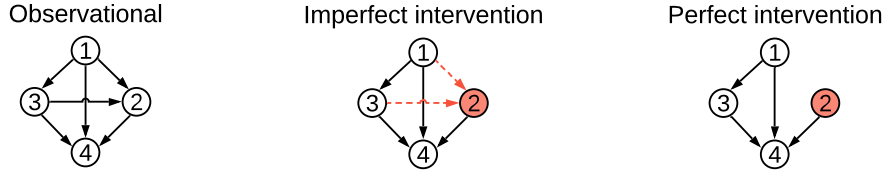

Figure 1: Different intervention types (shown in red). In imperfect interventions, the causal relationships are altered. In perfect interventions, the targeted node is cut out from its parents.

Nevertheless, even with interventional data at hand, finding the right DAG is challenging. The solution space is immense and grows super-exponentially with the number of variables. Recently, Zheng et al. [52] proposed to cast this search problem as a constrained continuous-optimization problem, avoiding the computationally-intensive search typically performed by score-based and constraint-based methods [36]. The work of Zheng et al. [52] was limited to linear relationships, but was quickly extended to nonlinear ones via neural networks [27, 49, 53, 32, 21, 54]. Yet, these approaches do not make use of interventional data and must therefore rely on strong parametric assumptions (e.g., gaussian additive noise models). Bengio et al. [1] leveraged interventions and continuous optimization to learn the causal direction in the bivariate setting. The follow-up work of Ke et al. [23] generalized to the multivariate setting by optimizing an unconstrained objective with regularization inspired by Zheng et al. [52], but lacked theoretical guarantees. In this work, we propose a theoretically-grounded differentiable approach to causal discovery that can make use of *interventional* data (with potentially unknown targets) and that relies on the constrained-optimization framework of [52] without making strong assumptions about the functional form of causal mechanisms, thanks to expressive density estimators.

## 1.1 Contributions

- We propose Differentiable Causal Discovery with Interventions (DCDI): a general differentiable causal structure learning method that can leverage perfect, imperfect and unknown-target interventions (Section 3). We propose two instantiations, one of which is a universal density approximator that relies on normalizing flows (Section 3.4).

- We show that the exact maximization of the proposed score will identify the $\mathcal{I}$-Markov equivalence class [48] of the ground truth graph (under regularity conditions) for both the known- and unknown-target settings (Thm. 1 in Section 3.1 & Thm. 2 in Section 3.3, respectively).

- We provide an extensive comparison of DCDI to state-of-the-art methods in a wide variety of conditions, including multiple functional forms and types of interventions (Section 4).

## 2 Background and related work

### 2.1 Definitions

**Causal graphical models.** A CGM is defined by a distribution $P_X$ over a random vector $X = (X_1, \cdots, X_d)$ and a DAG $\mathcal{G} = (V, E)$. Each node $i \in V = \{1, \cdots d\}$ is associated with a random variable $X_i$ and each edge $(i, j) \in E$ represents a direct causal relation from variable $X_i$ to $X_j$. The distribution $P_X$ is Markov to the graph $\mathcal{G}$, which means that the joint distribution can be factorized as

$$p(x_1, \cdots, x_d) = \prod_{j=1}^{d} p_j(x_j | x_{\pi_j^{\mathcal{G}}}), \tag{1}$$

where $\pi_j^{\mathcal{G}}$ is the set of parents of the node $j$ in the graph $\mathcal{G}$, and $x_B$, for a subset $B \subseteq V$, denotes the entries of the vector $x$ with indices in $B$. In this work, we assume *causal sufficiency*, i.e., there is no hidden common cause that is causing more than one variable in $X$ [36].

**Interventions.** In contrast with standard Bayesian Networks, CGMs support interventions. Formally, an intervention on a variable $x_j$ corresponds to replacing its conditional $p_j(x_j | x_{\pi_j^{\mathcal{G}}})$ by a new

conditional $\tilde{p}_j(x_j|x_{\pi_j^{\mathcal{G}}})$ in Equation (1), thus modifying the distribution only locally. Interventions can be performed on multiple variables simultaneously and we call the *interventional target* the set $I \subseteq V$ of such variables. When considering more than one intervention, we denote the interventional target of the $k$th intervention by $I_k$. Throughout this paper, we assume that the observational distribution (the original distribution without interventions) is observed, and denote it by $I_1 := \emptyset$. We define the *interventional family* by $\mathcal{I} := (I_1, \cdots, I_K)$, where $K$ is the number of interventions (including the observational setting). Finally, the $k$th interventional joint density is

$$p^{(k)}(x_1, \cdots, x_d) := \prod_{j \notin I_k} p_j^{(1)}(x_j|x_{\pi_j^{\mathcal{G}}}) \prod_{j \in I_k} p_j^{(k)}(x_j|x_{\pi_j^{\mathcal{G}}}), \tag{2}$$

where the assumption of causal sufficiency is implicit to this definition of interventions.

**Type of interventions.** The general type of interventions described in (2) are called imperfect (or soft, parametric) [36, 7, 8]. A specific case that is often considered is (stochastic) perfect interventions (or hard, structural) [10, 48, 26] where $p_j^{(k)}(x_j|x_{\pi_j^{\mathcal{G}}}) = p_j^{(k)}(x_j)$ for all $j \in I_k$, thus removing the dependencies with their parents (see Figure 1). Real-world examples of these types of interventions include gene knockout/knockdown in biology. Analogous to a perfect intervention, a gene knockout completely suppresses the expression of one gene and removes dependencies to regulators of gene expression. In contrast, a gene knockdown hinders the expression of one gene without removing dependencies with regulators [55], and is thus an imperfect intervention.

## 2.2 Causal structure learning

In causal structure learning, the goal is to recover the causal DAG $\mathcal{G}$ using samples from $P_X$ and, when available, from interventional distributions. This problem presents two main challenges: 1) the size of the search space is super-exponential in the number of nodes [5] and 2) the true DAG is not always identifiable (more severe without interventional data). Methods for this task are often divided into three groups: constraint-based, score-based, and hybrid methods. We briefly review these below.

**Constraint-based methods** typically rely on conditional independence testing to identify edges in $\mathcal{G}$. The PC algorithm [41] is a classical example that works with observational data. It performs conditional independence tests with a conditioning set that increases at each step of the algorithm and finds an equivalence class that satisfies all independencies. Methods that support interventional data include COmbINE [45], HEJ [19], which both rely on Boolean satisfiability solvers to find a graph that satisfies all constraints; and [24], which proposes an algorithm inspired by FCI [41]. In contrast with our method, these methods account for latent confounders. The *Joint causal inference* framework (JCI) [31] supports latent confounders and can deal with interventions with unknown targets. This framework can be used with various observational constraint-based algorithms such as PC or FCI. Another type of constraint-based method exploits the invariance of causal mechanisms across interventional distributions, e.g., ICP [35, 17]. As will later be presented in Section 3, our loss function also accounts for such invariances.

**Score-based methods** formulate the problem of estimating the ground truth DAG $\mathcal{G}^*$ by optimizing a score function $\mathcal{S}$ over the space of DAGs. The estimated DAG $\hat{\mathcal{G}}$ is given by

$$\hat{\mathcal{G}} \in \underset{\mathcal{G} \in \text{DAG}}{\arg\max} \, \mathcal{S}(\mathcal{G}) \,. \tag{3}$$

A typical choice of score in the purely observational setting is the regularized maximum likelihood:

$$\mathcal{S}(\mathcal{G}) := \max_{\theta} \mathbb{E}_{X \sim P_X} \log f_{\theta}(X) - \lambda|\mathcal{G}| \,, \tag{4}$$

where $f_{\theta}$ is a density function parameterized by $\theta$, $|\mathcal{G}|$ is the number of edges in $\mathcal{G}$ and $\lambda$ is a positive scalar.[1] Since the space of DAGs is super-exponential in the number of nodes, these methods often rely on greedy combinatorial search algorithms. A typical example is GIES [15], an adaptation of GES [5] to perfect interventions. In contrast with our method, GIES assumes a *linear* gaussian model and optimizes the Bayesian information criterion (BIC) over the space of $\mathcal{I}$-Markov equivalence

classes (see Definition 6 in Appendix A.1). CAM [4] is also a score-based method using greedy search, but it is nonlinear: it assumes an additive noise model where the nonlinear functions are additive. In the original paper, CAM only addresses the observational case where additive noise models are identifiable, however code is available to support perfect interventions.

**Hybrid methods** combine constraint and score-based approaches. Among these, IGSP [47, 48] is a method that optimizes a score based on conditional independence tests. Contrary to GIES, this method has been shown to be consistent under the faithfulness assumption. Furthermore, this method has recently been extended to support interventions with unknown targets (UT-IGSP) [42], which are also supported by our method.

## 2.3 Continuous constrained optimization for structure learning

A new line of research initiated by Zheng et al. [52], which serves as the basis for our work, reformulates the combinatorial problem of finding the optimal DAG as a continuous constrained-optimization problem, effectively avoiding the combinatorial search. Analogous to standard score-based approaches, these methods rely on a model $f_\theta$ parametrized by $\theta$, though $\theta$ also encodes the graph $\mathcal{G}$. Central to this class of methods are both the use a *weighted adjacency matrix* $A_\theta \in \mathbb{R}_{\geq 0}^{d \times d}$ (which depends on the parameters of the model) and the acyclicity constraint introduced by Zheng et al. [52] in the context of linear models:

$$\operatorname{Tr} e^{A_\theta} - d = 0 \,. \tag{5}$$

The weighted adjacency matrix encodes the DAG estimator $\hat{\mathcal{G}}$ as $(A_\theta)_{ij} > 0 \iff i \to j \in \hat{\mathcal{G}}$. Zheng et al. [52] showed, in the context of linear models, that $\hat{\mathcal{G}}$ is acyclic if and only if the constraint $\operatorname{Tr} e^{A_\theta} - d = 0$ is satisfied. The general optimization problem is then

$$\max_\theta \mathbb{E}_{X \sim P_X} \log f_\theta(X) - \lambda \Omega(\theta) \ \text{ s.t. } \ \operatorname{Tr} e^{A_\theta} - d = 0 \,, \tag{6}$$

where $\Omega(\theta)$ is a regularizing term penalizing the number of edges in $\hat{\mathcal{G}}$. This problem is then approximately solved using an augmented Lagrangian procedure, as proposed by Zheng et al. [52]. Note that the problem in Equation (6) is very similar to the one resulting from Equations (3) and (4).

Continuous-constrained methods differ in their choice of model, weighted adjacency matrix, and the specifics of their optimization procedures. For instance, NOTEARS [52] assumes a Gaussian linear model with equal variances where $\theta := W \in \mathbb{R}^{d \times d}$ is the matrix of regression coefficients, $\Omega(\theta) := ||W||_1$ and $A_\theta := W \odot W$ is the weighted adjacency matrix. Several other methods use neural networks to model nonlinear relations via $f_\theta$ and have been shown to be competitive with classical methods [27, 53]. In some methods, the parameter $\theta$ can be partitioned into $\theta_1$ and $\theta_2$ such that $f_\theta = f_{\theta_1}$ and $A_\theta = A_{\theta_2}$ [21, 32, 23] while in others, such a decoupling is not possible, i.e., the adjacency matrix $A_\theta$ is a function of the neural networks parameters [27, 53]. In terms of scoring, most methods rely on maximum likelihood or variants like implicit maximum likelihood [21] and evidence lower bound [49]. Zhu and Chen [54] also rely on the acyclicity constraint, but use reinforcement learning as a search strategy to estimate the DAG. Ke et al. [23] learn a DAG from interventional data by optimizing an unconstrained objective with a regularization term inspired by the acyclicity constraint, but that penalizes only cycles of length two. However, their work is limited to discrete distributions and single-node interventions. To the best of our knowledge, no work has investigated, in a general manner, the use of continuous-constrained approaches in the context of interventions as we present in the next section.

## 3 DCDI: Differentiable causal discovery from interventional data

In this section, we present a score for imperfect interventions, provide a theorem showing its validity, and show how it can be maximized using the continuous-constrained approach to structure learning. We also provide a theoretically grounded extension to interventions with unknown targets.

### 3.1 A score for imperfect interventions

The model we consider uses neural networks to model conditional densities. Moreover, we encode the DAG $\mathcal{G}$ with a binary adjacency matrix $M^{\mathcal{G}} \in \{0, 1\}^{d \times d}$ which acts as a mask on the neural networks

inputs. We similarly encode the interventional family $\mathcal{I}$ with a binary matrix $R^{\mathcal{I}} \in \{0,1\}^{K \times d}$, where $R^{\mathcal{I}}_{kj} = 1$ means that $X_j$ is a target in $I_k$. In line with the definition of interventions in Equation (2), we model the joint density of the $k$th intervention by

$$f^{(k)}(x; M^{\mathcal{G}}, R^{\mathcal{I}}, \phi) := \prod_{j=1}^{d} \tilde{f}(x_j; \text{NN}(M^{\mathcal{G}}_j \odot x; \phi^{(1)}_j))^{1-R^{\mathcal{I}}_{kj}} \tilde{f}(x_j; \text{NN}(M^{\mathcal{G}}_j \odot x; \phi^{(k)}_j))^{R^{\mathcal{I}}_{kj}}, \quad (7)$$

where $\phi := \{\phi^{(1)}, \cdots, \phi^{(K)}\}$, the NN's are neural networks parameterized by $\phi^{(1)}_j$ or $\phi^{(k)}_j$, the operator $\odot$ denotes the Hadamard product (element-wise) and $M^{\mathcal{G}}_j$ denotes the $j$th column of $M^{\mathcal{G}}$, which enables selecting the parents of node $j$ in the graph $\mathcal{G}$. The neural networks output the parameters of a density function $\tilde{f}$, which in principle, could be any density. We experiment with Gaussian distributions and more expressive normalizing flows (see Section 3.4).

We denote $\mathcal{G}^*$ and $\mathcal{I}^* := (I_1^*, ..., I_K^*)$ to be the ground truth causal DAG and ground truth interventional family, respectively. In this section, we assume that $\mathcal{I}^*$ is known, but we will relax this assumption in Section 3.3. We propose maximizing with respect to $\mathcal{G}$ the following regularized maximum log-likelihood score:

$$\mathcal{S}_{\mathcal{I}^*}(\mathcal{G}) := \sup_{\phi} \sum_{k=1}^{K} \mathbb{E}_{X \sim p^{(k)}} \log f^{(k)}(X; M^{\mathcal{G}}, R^{\mathcal{I}^*}, \phi) - \lambda |\mathcal{G}|, \quad (8)$$

where $p^{(k)}$ stands for the $k$th ground truth interventional distribution from which the data is sampled. A careful inspection of (7) reveals that the conditionals of the model are invariant across interventions *in which they are not targeted*. Intuitively, this means that maximizing (8) will favor graphs $\mathcal{G}$ in which a conditional $p(x_j | x_{\pi^{\mathcal{G}}_j})$ is invariant across all interventional distributions in which $x_j$ is not a target, i.e., $j \notin I_k^*$. This is a fundamental property of causal graphical models.

We now present our first theoretical result (see Appendix A.2 for the proof). This theorem states that, under appropriate assumptions, maximizing $\mathcal{S}_{\mathcal{I}^*}(\mathcal{G})$ yields an estimated DAG $\hat{\mathcal{G}}$ that is $\mathcal{I}^*$-Markov equivalent to the true DAG $\mathcal{G}^*$. We use the notion of $\mathcal{I}^*$-Markov equivalence introduced by [48] and recall its meaning in Definition 6 of Appendix A.1. Briefly, the $\mathcal{I}^*$-Markov equivalence class of $\mathcal{G}^*$ is a set of DAGs which are indistinguishable from $\mathcal{G}^*$ given the interventional targets in $\mathcal{I}^*$. This means identifying the $\mathcal{I}^*$-Markov equivalence class of $\mathcal{G}^*$ is the *best* one can hope for given the interventions $\mathcal{I}^*$ *without making further distributional assumptions*.

**Theorem 1 (Identification via score maximization)** *Suppose the interventional family $\mathcal{I}^*$ is such that $I_1^* := \emptyset$. Let $\mathcal{G}^*$ be the ground truth DAG and $\hat{\mathcal{G}} \in \arg\max_{\mathcal{G} \in DAG} \mathcal{S}_{\mathcal{I}^*}(\mathcal{G})$. Assume that the density model has enough capacity to represent the ground truth distributions, that $\mathcal{I}^*$-faithfulness holds, that the density model is strictly positive and that the ground truth densities $p^{(k)}$ have finite differential entropy, respectively Assumptions 1, 2, 3 & 4 (see Appendix A.2 for precise statements). Then for $\lambda > 0$ small enough, we have that $\hat{\mathcal{G}}$ is $\mathcal{I}^*$-Markov equivalent to $\mathcal{G}^*$.*

*Proof idea.* Using the graphical characterization of $\mathcal{I}$-Markov equivalence from Yang et al. [48], we verify that every graph outside the equivalence class has a lower score than that of the ground truth graph. We show this by noticing that any such graph will either have more edges than $\mathcal{G}^*$ or limit the distributions expressible by the model in such a way as to prevent it from properly fitting the ground truth. Moreover, the coefficient $\lambda$ must be chosen small enough to avoid too sparse solutions. $\square$

$\mathcal{I}^*$-faithfulness (Assumption 2) enforces two conditions. The first one is the usual faithfulness condition, i.e., whenever a conditional independence statement holds in the observational distribution, the corresponding d-separation holds in $\mathcal{G}^*$. The second one requires that the interventions are non-pathological in the sense that every variable that can be potentially affected by the intervention are indeed affected. See Appendix A.2 for more details and examples of $\mathcal{I}^*$-faithfulness violations.

To interpret this result, note that the $\mathcal{I}^*$-Markov equivalence class of $\mathcal{G}^*$ tends to get smaller as we add interventional targets to the interventional family $\mathcal{I}^*$. As an example, when $\mathcal{I}^* = (\emptyset, \{1\}, \cdots, \{d\})$, i.e., when each node is individually targeted by an intervention, $\mathcal{G}^*$ is alone in its equivalence class and, if assumptions of Theorem 1 hold, $\hat{\mathcal{G}} = \mathcal{G}^*$. See Corollary 11 in Appendix A.1 for details.

**Perfect interventions.** The score $\mathcal{S}_{\mathcal{I}^*}(\mathcal{G})$ can be specialized for perfect interventions, i.e., where the targeted nodes are completely disconnected from their parents. The idea is to leverage the fact that the

conditionals targeted by the intervention in Equation (7) should not depend on the graph $\mathcal{G}$ anymore. This means that these terms can be removed without affecting the maximization w.r.t. $\mathcal{G}$. We use this version of the score when experimenting with perfect interventions and present it in Appendix A.4.

## 3.2 A continuous-constrained formulation

To allow for gradient-based stochastic optimization, we follow [21, 32] and treat the adjacency matrix $M^{\mathcal{G}}$ as *random*, where the entries $M_{ij}^{\mathcal{G}}$ are independent Bernoulli variables with success probability $\sigma(\alpha_{ij})$ ($\sigma$ is the sigmoid function) and $\alpha_{ij}$ is a scalar parameter. We group these $\alpha_{ij}$'s into a matrix $\Lambda \in \mathbb{R}^{d \times d}$. We then replace the score $\mathcal{S}_{\mathcal{I}^*}(\mathcal{G})$ (8) with the following relaxation:

$$\hat{\mathcal{S}}_{\mathcal{I}^*}(\Lambda) := \sup_{\phi} \mathop{\mathbb{E}}_{M \sim \sigma(\Lambda)} \left[ \sum_{k=1}^{K} \mathop{\mathbb{E}}_{X \sim p^{(k)}} \log f^{(k)}(X; M, R^{\mathcal{I}^*}, \phi) - \lambda ||M||_0 \right], \tag{9}$$

where we dropped the $\mathcal{G}$ superscript in $M$ to lighten notation. This score tends asymptotically to $\mathcal{S}_{\mathcal{I}^*}(\mathcal{G})$ as $\sigma(\Lambda)$ progressively concentrates its mass on $\mathcal{G}$.[2] While the expectation of the log-likelihood term is intractable, the expectation of the regularizing term simply evaluates to $\lambda ||\sigma(\Lambda)||_1$. This score can then be maximized under the acyclicity constraint presented in Section 2.3:

$$\sup_{\Lambda} \hat{\mathcal{S}}_{\mathcal{I}^*}(\Lambda) \quad \text{s.t.} \quad \operatorname{Tr} e^{\sigma(\Lambda)} - d = 0. \tag{10}$$

This problem presents two main challenges: it is a constrained problem and it contains intractable expectations. As proposed by [52], we rely on the *augmented Lagrangian* procedure to optimize $\phi$ and $\Lambda$ jointly under the acyclicity constraint. This procedure transforms the constrained problem into a sequence of unconstrained subproblems which can themselves be optimized via a standard stochastic gradient descent algorithm for neural networks such as RMSprop. The procedure should converge to a stationary point of the original constrained problem (which is not necessarily the global optimum due to the non-convexity of the problem). In Appendix B.3, we give details on the augmented Lagrangian procedure and show the learning process in details with a concrete example.

The gradient of the likelihood part of $\hat{\mathcal{S}}_{\mathcal{I}^*}(\Lambda)$ w.r.t. $\Lambda$ is estimated using the Straight-Through Gumbel estimator. This amounts to using Bernoulli samples in the forward pass and Gumbel-Softmax samples in the backward pass which can be differentiated w.r.t. $\Lambda$ via the reparametrization trick [20, 29]. This approach was already shown to give good results in the context of continuous optimization for causal discovery in the purely observational case [32, 21]. We emphasize that our approach belongs to the general framework presented in Section 2.3 where the global parameter $\theta$ is $\{\phi, \Lambda\}$, the weighted adjacency matrix $A_\theta$ is $\sigma(\Lambda)$ and the regularizing term $\Omega(\theta)$ is $||\sigma(\Lambda)||_1$.

## 3.3 Interventions with unknown targets

Until now, we have assumed that the ground truth interventional family $\mathcal{I}^*$ is known. We now consider the case were it is unknown and, thus, needs to be learned. To do so, we propose a simple modification of score (8) which consists in adding regularization to favor sparse interventional families.

$$\mathcal{S}(\mathcal{G}, \mathcal{I}) := \sup_{\phi} \sum_{k=1}^{K} \mathbb{E}_{X \sim p^{(k)}} \log f^{(k)}(X; M^{\mathcal{G}}, R^{\mathcal{I}}, \phi) - \lambda |\mathcal{G}| - \lambda_R |\mathcal{I}|, \tag{11}$$

where $|\mathcal{I}| = \sum_{k=1}^{K} |I_k|$. The following theorem, proved in Appendix A.3, extends Theorem 1 by showing that, under the same assumptions, maximizing $\mathcal{S}(\mathcal{G}, \mathcal{I})$ with respect to both $\mathcal{G}$ and $\mathcal{I}$ recovers both the $\mathcal{I}^*$-Markov equivalence class of $\mathcal{G}^*$ and the ground truth interventional family $\mathcal{I}^*$.

**Theorem 2 (Unknown targets identification)** *Suppose $\mathcal{I}^*$ is such that $I_1^* := \emptyset$. Let $\mathcal{G}^*$ be the ground truth DAG and $(\hat{\mathcal{G}}, \hat{\mathcal{I}}) \in \arg\max_{\mathcal{G} \in DAG, \mathcal{I}} \mathcal{S}(\mathcal{G}, \mathcal{I})$. Under the same assumptions as Theorem 1 and for $\lambda, \lambda_R > 0$ small enough, $\hat{\mathcal{G}}$ is $\mathcal{I}^*$-Markov equivalent to $\mathcal{G}^*$ and $\hat{\mathcal{I}} = \mathcal{I}^*$.*

*Proof idea.* We simply append a few steps at the beginning of the proof of Theorem 1 which show that whenever $\mathcal{I} \neq \mathcal{I}^*$, the resulting score is worse than $\mathcal{S}(\mathcal{G}^*, \mathcal{I}^*)$, and hence is not optimal. This is done using arguments very similar to Theorem 1 and choosing $\lambda$ and $\lambda_R$ small enough. $\square$

Theorem 2 informs us that ignoring which nodes are targeted during interventions does not affect identifiability. However, this result assumes implicitly that the learner knows which data set is the observational one.

Similarly to the development of Section 3.2, the score $\mathcal{S}(\mathcal{G}, \mathcal{I})$ can be relaxed by treating entries of $M^{\mathcal{G}}$ and $R^{\mathcal{I}}$ as independent Bernoulli random variables parameterized by $\sigma(\alpha_{ij})$ and $\sigma(\beta_{kj})$, respectively. We thus introduced a new learnable parameter $\beta$. The resulting relaxed score is similar to (9), but the expectation is taken w.r.t. to $M$ and $R$. Similarly to $\Lambda$, the Straight-Through Gumbel estimator is used to estimate the gradient of the score w.r.t. the parameters $\beta_{kj}$. For perfect interventions, we adapt this score by masking all inputs of the neural networks under interventions.

The related work of Ke et al. [23], which also support unknown targets, bears similarity to DCDI but addresses a different setting in which interventions are obtained sequentially in an online fashion. One important difference is that their method attempts to identify the *single node* that has been intervened upon (as a hard prediction), whereas DCDI learns a distribution over all potential interventional families via the continuous parameters $\sigma(\beta_{kj})$, which typically becomes deterministic at convergence. Ke et al. [23] also use random masks to encode the graph structure but estimates the gradient w.r.t. their distribution parameters using the log-trick which is known to have high variance [39] compared to reparameterized gradient [29].

### 3.4 DCDI with normalizing flows

In this section, we describe how the scores presented in Sections 3.2 & 3.3 can accommodate powerful density approximators. In the purely observational setting, very expressive models usually hinder identifiability, but this problem vanishes when enough interventions are available. There are many possibilities when it comes to the choice of the density function $\tilde{f}$. In this paper, we experimented with simple Gaussian distributions as well as *normalizing flows* [38] which can represent complex causal relationships, e.g., multi-modal distributions that can occur in the presence of latent variables that are parent of only one variable.

A normalizing flow $\tau(\cdot; \omega)$ is an invertible function (e.g., a neural network) parameterized by $\omega$ with a tractable Jacobian, which can be used to model complex densities by transforming a simple random variable via the change of variable formula:

$$\tilde{f}(z; \omega) := \left| \det \left( \frac{\partial \tau(z; \omega)}{\partial z} \right) \right| p(\tau(z; \omega)), \tag{12}$$

where $\frac{\partial \tau(z;\omega)}{\partial z}$ is the Jacobian matrix of $\tau(\cdot; \omega)$ and $p(\cdot)$ is a simple density function, e.g., a Gaussian. The function $\tilde{f}(\cdot; \omega)$ can be plugged directly into the scores presented earlier by letting the neural networks $\mathrm{NN}(\cdot; \phi_j^{(k)})$ output the parameter $\omega_j$ of the normalizing flow $\tau_j$ for each variable $x_j$. In our implementation, we use *deep sigmoidal flows* (DSF), a specific instantiation of normalizing flows which is a universal density approximator [18]. Details about DSF are relayed to Appendix B.2.

## 4 Experiments

We tested DCDI with Gaussian densities (DCDI-G) and with normalizing flows (DCDI-DSF) on a real-world data set and several synthetic data sets. The real-world task is a flow cytometry data set from Sachs et al. [40]. Our results, reported in Appendix C.1, show that our approach performs comparably to state-of-the-art methods. In this section, we focus on synthetic data sets, since these allow for a more systematic comparison of methods against various factors of variation (type of interventions, graph size, density, type of mechanisms).

We consider synthetic data sets with three interventional settings: perfect/known, imperfect/known, and perfect/unknown. Each data set has one of the three different types of causal mechanisms: i) linear [42], ii) nonlinear additive noise model (ANM) [4], and iii) nonlinear with non-additive noise using neural networks (NN) [21]. For each data set type, graphs vary in size ($d = 10$ or $20$) and density ($e = 1$ or $4$ where $e \cdot d$ is the average number of edges). For conciseness, we present results for 20-node graphs in the main text and report results on 10-node graphs in Appendix C.7; conclusions are similar for all sizes. For each condition, ten graphs are sampled with their causal mechanisms and then observational and interventional data are generated. Each data set has 10 000 samples uniformly

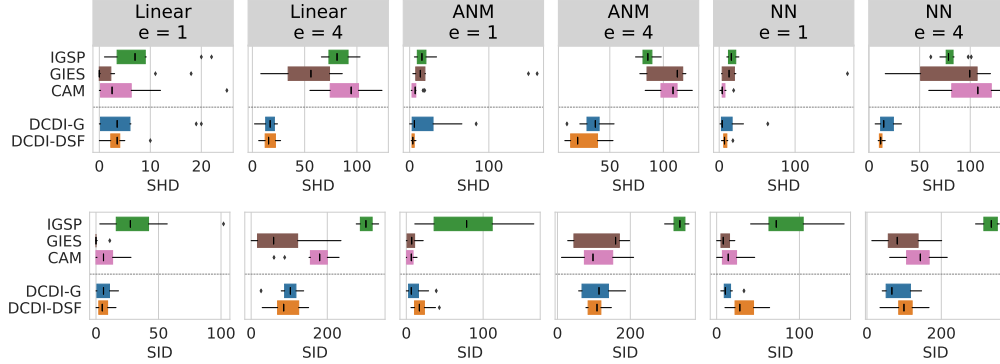

Figure 2: **Perfect interventions.** SHD and SID (lower is better) for 20-node graphs

distributed in the different interventional settings. A total of $d$ interventions were performed, each by sampling up to $0.1d$ target nodes. For more details on the generation process, see Appendix B.1.

Most methods have an hyperparameter controlling DAG sparsity. Although performance is sensitive to this hyperparameter, many papers do not specify how it was selected. For score-based methods (GIES, CAM and DCDI), we select it by maximizing the held-out likelihood as explained in Appendix B.5 (without using the ground truth DAG). In contrast, since constraint-based methods (IGSP, UT-IGSP, JCI-PC) do not yield a likelihood model to evaluate on held-out data, we use a fixed cutoff parameter ($\alpha = 1\text{e}{-}3$) that leads to good results. We report additional results with different cutoff values in Appendix C.7. For IGSP and UT-IGSP, we always use the independence test well tailored to the data set type: partial correlation test for Gaussian linear data and KCI-test [50] for nonlinear data.

The performance of each method is assessed by two metrics comparing the estimated graph to the ground truth graph: i) the *structural Hamming distance* (SHD) which is simply the number of edges that differ between two DAGs (either reversed, missing or superfluous) and ii) the *structural interventional distance* (SID) which assesses how two DAGs differ with respect to their causal inference statements [34]. In Appendix C.6, we also report how well the graph can be used to predict the effect of unseen interventions [13]. Our implementation is available here and additional information about the baseline methods is provided in Appendix B.4.

### 4.1  Results for different intervention types

**Perfect interventions.** We compare our methods to GIES [15], a modified version of CAM [4] that support interventions and IGSP [47]. The conditionals of targeted nodes were replaced by the marginal $\mathcal{N}(2,1)$ similarly to [15, 42]. Boxplots for SHD and SID over 10 graphs are shown in Figure 2. For all conditions, DCDI-G and DCDI-DSF shows competitive results in term of SHD and SID. For graphs with a higher number of average edges, DCDI-G and DCDI-DSF outperform all methods. GIES often shows the best performance for the linear data set, which is not surprising given that it makes the right assumptions, i.e., linear functions with Gaussian noise.

**Imperfect interventions.** Our conclusions are similar to the perfect intervention setting. As shown in Figure 3, DCDI-G and DCDI-DSF show competitive results and outperform other methods for graphs with a higher connectivity. The nature of the imperfect interventions are explained in Appendix B.1.

**Perfect unknown interventions.** We compare to UT-IGSP [42], an extension of IGSP that deal with unknown interventions. The data used are the same as in the perfect intervention setting, but the intervention targets are hidden. Results are shown in Figure 4. Except for linear data sets with sparse graphs, DCDI-G and DCDI-DSF show an overall better performance than UT-IGSP.

**Summary.** For all intervention settings, DCDI has overall the best performance. In Appendix C.5, we show similar results for different types of perfect/imperfect interventions. While the advantage of DCDI-DSF over DCDI-G is marginal, it might be explained by the fact that the densities can be sufficiently well modeled by DCDI-G. In Appendix C.2, we show cases where DCDI-G fails to detect the right causal direction due to its lack of capacity, whereas DCDI-DSF systematically succeeds. In Appendix C.4, we present an ablation study confirming the advantage of neural networks against linear models and the ability of our score to leverage interventional data.

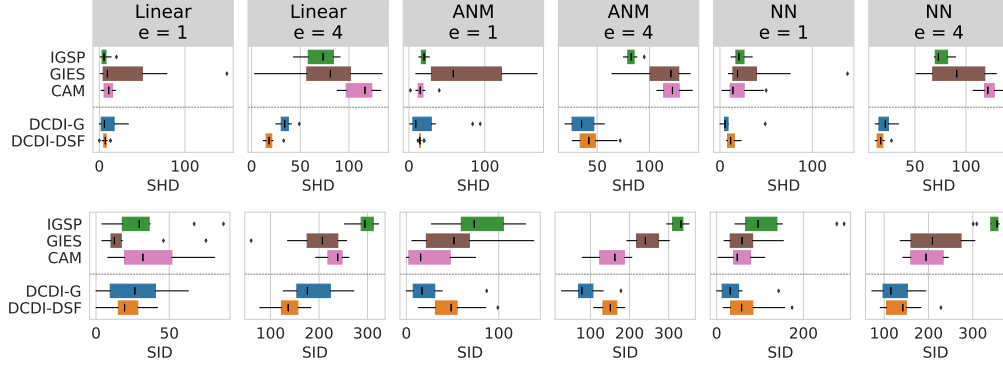

Figure 3: **Imperfect interventions.** SHD and SID for 20-node graphs

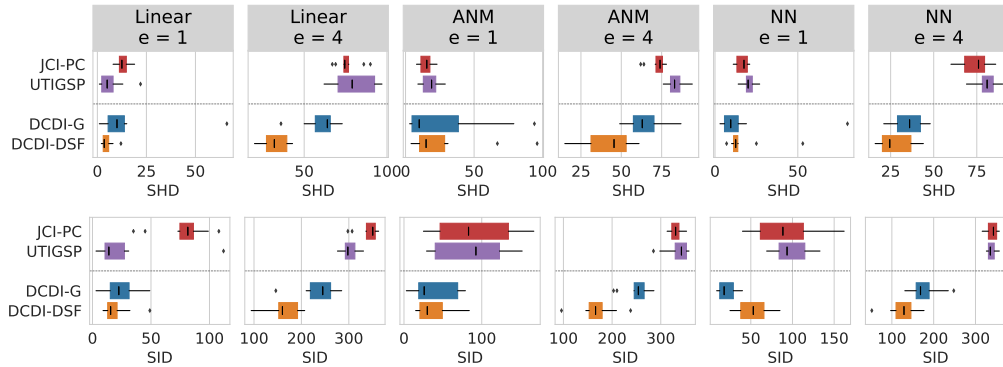

Figure 4: **Unknown interventions.** SHD and SID for 20-node graphs

## 4.2 Scalability experiments

So far the experiments focused on moderate size data sets, both in terms of number of variables (10 or 20) and number of examples ($\approx 10^4$). In Appendix C.3, we compare the running times of DCDI to those of other methods on graphs of up to 100 nodes and on data sets of up to 1 million examples.

The augmented Lagrangian procedure on which DCDI relies requires the computation of the matrix exponential at each gradient step, which costs $O(d^3)$. We found this does not prevent DCDI from being applied to 100 nodes graphs. Several constraint-based methods use kernel-based conditional independence tests [50, 12], which scale poorly with the number of examples. For example, KCI-test scales in $O(n^3)$ [43] and HSIC in $O(n^2)$ [51]. On the other hand, DCDI is not greatly affected by the sample size since it relies on stochastic gradient descent which is known to scale well with the data set size [3]. Our comparison shows that, among all considered methods, DCDI is the only one supporting nonlinear relationships that can scale to as much as one million examples. We believe that this can open the way to new applications of causal discovery where data is abundant.

## 5 Conclusion

We proposed a general continuous-constrained method for causal discovery which can leverage various types of interventional data as well as expressive neural architectures, such as normalizing flows. This approach is rooted in a sound theoretical framework and is competitive with other state-of-the-art algorithms on real and simulated data sets, both in terms of graph recovery and scalability. This work opens interesting opportunities for future research. One direction is to extend DCDI to time-series data, where non-stationarities can be modeled as unknown interventions [37]. Another exciting direction is to learn representations of variables across multiple systems that could serve as prior knowledge for causal discovery in low data settings.

## Broader impact

Causal structure learning algorithms are general tools that address two high-level tasks: *understanding* and *acting*. That is, they can help a user understand a complex system and, once such an understanding is achieved, they can help in recommending actions. We envision positive impacts of our work in fields such as scientific investigation (e.g., interpreting and anticipating the outcome of experiments), policy making for decision-makers (e.g., identifying actions that could stimulate economic growth), and improving policies in autonomous agents (e.g., learning causal relationships in the world via interaction). As a concrete example, consider the case of gene knockouts/knockdowns experiments in the field of genomics, which aim to understand how specific genes and diseases interact [55]. Learning causal models using interventions performed in this setting could help gain precious insight into gene pathways, which may catalyze the development of better pharmaceutic targets and broaden our understanding of complex diseases such as cancer. Of course, applications are likely to extend beyond these examples which seem natural from our current position.

Like any methodological contribution, our work is not immune to undesirable applications that could have negative impacts. For instance, it would be possible, yet unethical for a policy-maker to use our algorithm to understand how specific human-rights violations can reduce crime and recommend their enforcement. The burden of using our work within ethical and benevolent boundaries would rely on the user. Furthermore, even when used in a positive application, our method could have unintended consequences if used without understanding its assumptions.

In order to use our method correctly, it is crucial to understand the assumptions that it makes about the data. When such assumptions are not met, the results may still be valid, but should be used as a support to decision rather than be considered as the absolute truth. These assumptions are:

- Causal sufficiency: there are no hidden confounding variables
- The samples for a given interventional distribution are independent and identically distributed
- The causal relationships form an acyclic graph (no feedback loops)
- Our theoretical results are valid in the infinite-data regime

We encourage users to be mindful of this and to carefully analyze their results before making decisions that could have a significant downstream impact.

## Acknowledgments

This research was partially supported by the Canada CIFAR AI Chair Program, by an IVADO excellence PhD scholarship and by a Google Focused Research award. The experiments were in part enabled by computational resources provided by Element AI, Calcul Quebec, Compute Canada. The authors would like to thank Nicolas Chapados, Rémi Lepriol, Damien Scieur and Assya Trofimov for their useful comments on the writing, Jose Gallego and Brady Neal for reviewing the proofs of Theorem 1 & 2, and Grace Abuhamad for useful comments on the statement of broader impact. Simon Lacoste-Julien is a CIFAR Associate Fellow in the Learning in Machines & Brains program.

## Footnotes

[1]This turns into the BIC score when the expectation is estimated with $n$ samples, the model has one parameter per edge (like in linear models) and $\lambda = \frac{\log n}{2n}$ [36, Section 7.2.2].

[2]In practice, we observe that $\sigma(\Lambda)$ tends to become deterministic as we optimize.

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
