[Supplementary Material]

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

[3]Note that Proposition 4 holds even for distributions with densities which are not strictly positive.

[4]Yang et al. [48] defines $\mathcal{M}_{\mathcal{I}}(\mathcal{G})$ slightly differently, but show their definition to be equivalent to the one used here. See Lemma A.1 in Yang et al. [48]

[5]The linearity of the Lebesgue integral is typically stated for Lebesgue integrable functions $f$ and $g$, i.e. $\int |f|, \int |g| < \infty$. See for example Billingsley [2, Theorem 16.1]. However, it can be extended to cases where $f$ and $g$ are not integrable, as long as $\int f$ and $\int g$ are well-defined and are not infinities of opposite sign (which would yield the undefined expression $\infty - \infty$). The proof is a simple adaptation of Theorem 16.1 which makes use of Theorem 15.1 in Billingsley [2].

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

# Appendix

## Table of Contents

## Contents

## A  Theory

### A.1  Theoretical Foundations for Causal Discovery with Imperfect Interventions

Before showing results about our regularized maximum likelihood score from Section 3.1, we start by briefly presenting useful definitions and results from Yang et al. [48]. We refer the reader to the original paper for a more comprehensive introduction to these notions, examples, and proofs. Throughout the appendix, we assume that the reader is comfortable with the concept of d-separation and immorality in directed graphs. These notions are presented in any standard book on probabilistic graphical models, e.g. Koller and Friedman [25]. Recall that $\mathcal{I} := (I_1^*, ..., I_K)$ and that we always assume $I_1 := \emptyset$. Following the approach of Yang et al. [48] and to simplify the presentation, we consider only densities which are strictly positive everywhere throught this appendix. We also note that while we present proofs for the cases where the distributions have densities with respect to the Lebesgue measure, all our results also hold for discrete distributions by simply replacing the Lebesgue measure with the counting measure in the integrals. We use the notation $i \to j \in \mathcal{G}$ to indicate that the edge $(i, j)$ is in the edge set of $\mathcal{G}$. Given disjoint $A, B, C \subset V$, when $C$ d-separates $A$ from $B$ in graph $\mathcal{G}$, we write $A \perp\!\!\!\perp_{\mathcal{G}} B \mid C$ and when random variables $X_A$ and $X_B$ are independent given $X_C$ in distribution $f$, we write $X_A \perp\!\!\!\perp_f X_B \mid X_C$.

**Definition 3** *For a DAG $\mathcal{G}$, let $\mathcal{M}(\mathcal{G})$ be the set of strictly positive densities $f : \mathbb{R}^d \to \mathbb{R}$ such that*

$$f(x_1, \cdots, x_d) = \prod_j f_j(x_j \mid x_{\pi_j^{\mathcal{G}}}), \tag{13}$$

where $\int_{\mathbb{R}} f_j(x_j \mid x_{\pi_j^{\mathcal{G}}}) dm(x_j) = 1$ for all $x_{\pi_j^{\mathcal{G}}} \in \mathbb{R}^{|\pi_j^{\mathcal{G}}|}$ and all $j \in [d]$, where $m$ is the Lebesgue measure on $\mathbb{R}$.

Next proposition is adapted from Lauritzen [28, Theorem 3.27]. It relates the factorization of (13) to d-separation statements.

**Proposition 4** *For a DAG $\mathcal{G}$ and a strictly positive density $f$,[3] we have $f \in \mathcal{M}(\mathcal{G})$ if and only if for any disjoint sets $A, B, C \subset V$ we have*

$$A \perp\!\!\!\perp_{\mathcal{G}} B \mid C \implies X_A \perp\!\!\!\perp_f X_B \mid X_C.$$

**Definition 5** *For a DAG $\mathcal{G}$ and an interventional family $\mathcal{I}$, let*

$$\mathcal{M}_{\mathcal{I}}(\mathcal{G}) := \{(f^{(k)})_{k \in [K]} \mid \forall k \in [K], \ f^{(k)} \in \mathcal{M}(\mathcal{G}) \text{ and } \forall j \notin I_k, f_j^{(k)}(x_j \mid x_{\pi_j^{\mathcal{G}}}) = f_j^{(1)}(x_j \mid x_{\pi_j^{\mathcal{G}}})\}.$$

Definition 5 defines a set $\mathcal{M}_{\mathcal{I}}(\mathcal{G})$ which contains all the sets of distributions $(f^{(k)})_{k \in [K]}$ which are coherent with the definition of interventions provided at Equation (2).[4] Note that the assumption of causal sufficiency is implicit to this definition of interventions. Analogously to the observational case, two different DAGs $\mathcal{G}_1$ and $\mathcal{G}_2$ can induce the same interventional distributions.

**Definition 6 ($\mathcal{I}$-Markov Equivalence Class)** *Two DAGs $\mathcal{G}_1$ and $\mathcal{G}_2$ are $\mathcal{I}$-Markov equivalent iff $\mathcal{M}_{\mathcal{I}}(\mathcal{G}_1) = \mathcal{M}_{\mathcal{I}}(\mathcal{G}_2)$. We denote by $\mathcal{I}$-$MEC(\mathcal{G}_1)$ the set of all DAGs which are $\mathcal{I}$-Markov equivalent to $\mathcal{G}_1$, this is the $\mathcal{I}$-Markov equivalence class of $\mathcal{G}_1$.*

We now define an augmented graph containing exactly one node for each intervention $k$.

**Definition 7** *Given a DAG $\mathcal{G}$ and an interventional family $\mathcal{I}$, the associated $\mathcal{I}$-DAG, denoted by $\mathcal{G}^{\mathcal{I}}$, is the graph $\mathcal{G}$ augmented with nodes $\zeta_k$ and edges $\zeta_k \to i$ for all $k \in [K] \setminus \{1\}$ and all $i \in I_k$.*

In the observational case, we say that a distribution $f$ has the Markov property w.r.t. a graph $\mathcal{G}$ if whenever some d-separation holds in the graph, the corresponding conditional independence holds in $f$. We now define the $\mathcal{I}$-Markov property, which generalizes this idea to interventions. This property is important since it holds in causal graphical models, as Proposition 9 states.

**Definition 8 ($\mathcal{I}$-Markov property)** *Let $\mathcal{I}$ be interventional family such that $I_1 := \emptyset$ and $(f^{(k)})_{k \in [K]}$ be a family of strictly positive densities over $X$. We say that $(f^{(k)})_{k \in [K]}$ satisfies the $\mathcal{I}$-Markov property w.r.t. the $\mathcal{I}$-DAG $\mathcal{G}^{\mathcal{I}}$ iff*

1. *For any disjoint $A, B, C \subset V$, $A \perp\!\!\!\perp_{\mathcal{G}} B|C$ implies $X_A \perp\!\!\!\perp_{f^{(k)}} X_B|X_C$ for all $k \in [K]$.*

2. *For any disjoint $A, C \subset V$ and $k \in [K] \setminus \{1\}$,*
   *$A \perp\!\!\!\perp_{\mathcal{G}^{\mathcal{I}}} \zeta_k \mid C \cup \zeta_{-k}$ implies $f^{(k)}(X_A|X_C) = f^{(1)}(X_A|X_C)$, where $\zeta_{-k} := \zeta_{[K] \setminus \{1,k\}}$.*

The next proposition relates the definition of interventions with the $\mathcal{I}$-Markov property that we just defined.

**Proposition 9** *(Yang et al. [48]) Suppose the interventional family $\mathcal{I}$ is such that $I_1 := \emptyset$. Then $(f^{(k)})_{k \in [K]} \in \mathcal{M}_{\mathcal{I}}(\mathcal{G})$ iff $(f^{(k)})_{k \in [K]}$ is $\mathcal{I}$-Markov to $\mathcal{G}^{\mathcal{I}}$.*

The next theorem gives a graphical characterization of $\mathcal{I}$-Markov equivalence classes, which will be crucial in the proof of Theorem 1.

**Theorem 10** *(Yang et al. [48]) Suppose the interventional family $\mathcal{I}$ is such that $I_1 := \emptyset$. Two DAGs $\mathcal{G}_1$ and $\mathcal{G}_2$ are $\mathcal{I}$-Markov equivalent iff their $\mathcal{I}$-DAGs $\mathcal{G}_1^{\mathcal{I}}$ and $\mathcal{G}_2^{\mathcal{I}}$ share the same skeleton and immoralities.*

See Figure 5 for a simple illustration of this concept. We now present a very simple corollary which gives a situation where the $\mathcal{I}$-Markov equivalence class contains a unique graph.

Figure 5: Different $\mathcal{I}$-DAGs with a single intervention. The first graph is alone in its $\mathcal{I}$-Markov equivalence class since reversing the $1 \to 2$ edge would break the immorality $1 \to 2 \leftarrow \zeta$. The second graph is also alone in its equivalence class since reversing $1 \to 2$ would create a new immorality $\zeta \to 1 \leftarrow 2$. The third DAG is not alone in its equivalence class since reversing $1 \to 2$ would preserve the skeleton without adding or removing an immorality. It should become apparent that adding more interventions will likely reduce the size of the $\mathcal{I}$-Markov equivalence class by introducing more immoralities.

**Corollary 11** *Let $\mathcal{G}$ be a DAG and let $\mathcal{I} = (\emptyset, \{1\}, \cdots, \{d\})$. Then $\mathcal{G}$ is alone in its $\mathcal{I}$-Markov equivalence class.*

*Proof.* By Theorem 10, all $\mathcal{I}$-Markov equivalent graphs will share its skeleton with $\mathcal{G}$, so we consider only graphs obtained by reversing edges in $\mathcal{G}$.

Consider any edge $i \to j$ in $\mathcal{G}$. We note that $i \to j \leftarrow \zeta_{j+1}$ forms an immorality in the $\mathcal{I}$-DAG $\mathcal{G}^{\mathcal{I}}$. Reversing $i \to j$ would break this immorality which would imply that the resulting DAG is not $\mathcal{I}$-Markov equivalent to $\mathcal{G}$, by Theorem 10. Hence, $\mathcal{G}$ is alone in its equivalence class.∎

### A.2 Proof of Theorem 1

We are now ready to present the main result of this section. We recall the score function introduced in Section 3.1:

$$\mathcal{S}_{\mathcal{I}^*}(\mathcal{G}) := \sup_{\phi} \sum_{k=1}^{K} \mathbb{E}_{X \sim p^{(k)}} \log f^{(k)}(X; M^{\mathcal{G}}, R^{\mathcal{I}^*}, \phi) - \lambda |\mathcal{G}| \,, \tag{14}$$

where

$$f^{(k)}(x; M^{\mathcal{G}}, R^{\mathcal{I}}, \phi) := \prod_{j=1}^{d} \tilde{f}(x_j; \mathrm{NN}(M_j^{\mathcal{G}} \odot x; \phi_j^{(1)}))^{1 - R_{kj}^{\mathcal{I}}} \tilde{f}(x_j; \mathrm{NN}(M_j^{\mathcal{G}} \odot x; \phi_j^{(k)}))^{R_{kj}^{\mathcal{I}}} \,. \tag{15}$$

Recall that $(p^{(k)})_{k \in [K]}$ are the ground truth interventional distributions with ground truth graph $\mathcal{G}^*$ and ground truth interventional family $\mathcal{I}^*$. We will sometimes use the notation $f_{\mathcal{G}\mathcal{I}\phi}^{(k)}(x)$ to refer to $f^{(k)}(x; M^{\mathcal{G}}, R^{\mathcal{I}}, \phi)$. We define $\mathcal{F}_{\mathcal{I}}(\mathcal{G})$ to be the set of all $(f^{(k)})_{k \in [K]}$ which are expressible by the model specified in Equation (15). More precisely,

$$\mathcal{F}_{\mathcal{I}}(\mathcal{G}) := \{(f^{(k)})_{k \in [K]} \mid \exists \phi \text{ s.t. } \forall k \in [K] \ f^{(k)} = f_{\mathcal{G}\mathcal{I}\phi}^{(k)}\} \,. \tag{16}$$

Theorem 1 relies on four assumptions. The first one requires that the model is expressive enough to represent the ground truth distributions exactly.

**Assumption 1 (Sufficient capacity)** *The ground truth interventional distributions $P^{(k)}$ all have a density $p^{(k)}$ w.r.t. the Lebesgue measure on $\mathbb{R}^n$ such that $(p^{(k)})_{k \in [K]} \in \mathcal{F}_{\mathcal{I}^*}(\mathcal{G}^*)$, i.e. the model specified in Equation (15) is expressive enough to represent the ground truth distributions.*

The second assumption is a generalization of faithfulness to interventions.

**Assumption 2 ($\mathcal{I}^*$-Faithfulness)**

    *1. For any disjoint $A, B, C \subset V$,*

$$A \not\perp_{\mathcal{G}^*} B | C \ \text{ implies } \ X_A \not\perp_{p^{(1)}} X_B | X_C \,.$$

2. *For any disjoint $A, C \subset V$ and $k \in [K]$,*

$$A \not\perp_{\mathcal{G}^* \mathcal{I}^*} \zeta_k \mid C \cup \zeta_{-k} \text{ implies } p^{(k)}\left(X_A | X_C\right) \neq p^{(1)}\left(X_A | X_C\right).$$

The first condition of Assumption 2 is exactly the standard faithfulness assumption for the ground truth observational distribution. The second condition is simply the converse of the second condition in the $\mathcal{I}$-Markov property (Definition 8) and can be understood as avoiding pathological interventions to make sure that every variables that can be potentially affected by the intervention are indeed affected. The simplest case is when $I_k := \{j\}$, $A := \{j\}$ and $C := \pi_j^{\mathcal{G}^*}$. In this case the condition requires that the intervention actually change something. Another simple case is when $C := \emptyset$. In this case, the condition requires that all descendants are affected, in the sense that their marginals change.

As we just saw, a trivial violation of $\mathcal{I}^*$-faithfulness would be when the intervention is not changing anything, not even the targeted conditional. We now present a non-trivial violation of $\mathcal{I}^*$-faithfulness.

**Example 12 ($\mathcal{I}^*$-Faithfulness violation)** *Suppose $\mathcal{G}^*$ is $X_1 \to X_2$ where both variables are binary. Assume $p^{(1)}(X_1 = 1) = \frac{1}{2}$, $p^{(1)}(X_2 = 1 \mid X_1 = 0) = \frac{1}{4}$ and $p^{(1)}(X_2 = 1 \mid X_1 = 1) = \frac{3}{4}$. From this, we can compute $p^{(1)}(X_2 = 1) = \frac{1}{2}$. Consider the intervention targeting only $X_2$ which changes its conditional to $p^{(2)}(X_2 = 1 \mid X_1 = 0) = \frac{3}{4}$ and $p^{(2)}(X_2 = 1 \mid X_1 = 1) = \frac{1}{4}$. So the interventional family is $\mathcal{I}^* = (\emptyset, \{2\})$. A simple computation shows the new marginal on $X_2$ has not changed, i.e. $p^{(2)}(X_2) = p^{(1)}(X_2)$. This is a violation of $\mathcal{I}^*$-faithfulness since clearly $X_2$ is not d-separated from the interventional node $\zeta_2$ in $\mathcal{G}^{*\mathcal{I}^*}$.*

The third assumption is a technicality to simplify the presentation of the proofs and to follow the presentation of Yang et al. [48]: we require the density model to be strictly positive.

**Assumption 3 (Strict positivity)** *For all $k \in [K]$, the model density $f^{(k)}(x; M^{\mathcal{G}}, R^{\mathcal{I}}, \phi)$ is strictly positive for all $\phi$, DAG $\mathcal{G}$ and interventional family $\mathcal{I}$.*

Note that Assumption 3 is satisfied for example when for all $\theta$ in the image of NN, the density $\tilde{f}(\cdot; \theta)$ is strictly positive. This happens when using a Gaussian density with variance strictly positive or a deep sigmoidal flow.

From Equation (16) and Assumption 3, it should be clear that $\mathcal{F}_{\mathcal{I}}(\mathcal{G}) \subset \mathcal{M}_{\mathcal{I}}(\mathcal{G})$ (recall $\mathcal{M}_{\mathcal{I}}(\mathcal{G})$ contains only strictly positive densities). Thus, from Proposition 9 we see that the $\mathcal{I}$-Markov property holds for all $(f^{(k)})_{k \in [K]} \in \mathcal{F}_{\mathcal{I}}(\mathcal{G})$. This fact will be useful in the proof of Theorem 1.

The fourth assumption is purely technical. It requires the differential entropy of the densities $p^{(k)}$ to be finite, which, as we will see in Lemma 13, ensures that the score of the ground truth graph $\mathcal{S}_{\mathcal{I}^*}(\mathcal{G}^*)$ is finite. This will be important to ensure that the score of any other graphs can be compared to it. In particular, this is avoiding the hypothetical situation where $\mathcal{S}_{\mathcal{I}^*}(\mathcal{G}^*)$ and $\mathcal{S}_{\mathcal{I}^*}(\mathcal{G})$ are both equal to infinity, which means they cannot be easily compared without defining a specific limiting process.

**Assumption 4 (Finite differential entropies)** *For all $k \in [K]$,*

$$|\mathbb{E}_{p^{(k)}} \log p^{(k)}(X)| < \infty.$$

**Lemma 13 (Finite scores)** *Under Assumptions 1 & 4, $|\mathcal{S}_{\mathcal{I}^*}(\mathcal{G}^*)| < \infty$.*

*Proof.* Consider the Kullback-Leibler divergence between $p^{(k)}$ and $f^{(k)}_{\mathcal{G}^* \mathcal{I}^* \phi}$ for an arbitrary $\phi$.

$$0 \leq D_{KL}(p^{(k)} || f^{(k)}_{\mathcal{G}^* \mathcal{I}^* \phi}) = \mathbb{E}_{p^{(k)}} \log p^{(k)}(X) - \mathbb{E}_{p^{(k)}} \log f^{(k)}_{\mathcal{G}^* \mathcal{I}^* \phi}(X), \quad (17)$$

where we applied the linearity of the expectation (which holds because $|\mathbb{E}_{p^{(k)}} \log p^{(k)}(X)| < \infty$). We thus have that

$$\mathbb{E}_{p^{(k)}} \log f^{(k)}_{\mathcal{G}^* \mathcal{I}^* \phi}(X) \leq \mathbb{E}_{p^{(k)}} \log p^{(k)}(X) < \infty. \quad (18)$$

Thus, $\sup_{\phi} \mathbb{E}_{p^{(k)}} \log f^{(k)}_{\mathcal{G}^* \mathcal{I}^* \phi}(X) < \infty$, which implies $\mathcal{S}_{\mathcal{I}^*}(\mathcal{G}^*) < \infty$.

By the assumption of sufficient capacity, there exists some $\phi^*$ such that $f_{\mathcal{G}^*\phi^*}^{(k)} = p^{(k)}$ for all $k$, hence $\sup_\phi \sum_{k=1}^K \mathbb{E}_{p^{(k)}} \log f_{\mathcal{G}^*\mathcal{I}^*\phi}^{(k)}(X) \geq \sum_{k=1}^K \mathbb{E}_{p^{(k)}} \log f_{\mathcal{G}^*\phi^*}^{(k)}(X) = \sum_{k=1}^K \mathbb{E}_{p^{(k)}} \log p^{(k)}(X) > -\infty$. This implies that $\mathcal{S}_{\mathcal{I}^*}(\mathcal{G}^*) > -\infty$. ∎

The next lemma shows that the difference $\mathcal{S}_{\mathcal{I}^*}(\mathcal{G}^*) - \mathcal{S}_{\mathcal{I}^*}(\mathcal{G})$ can be rewritten as a minimization of a sum of KL divergences plus the difference in regularizing terms.

**Lemma 14 (Rewriting of score differences)** *Under Assumption 1 & 4, we have*

$$\mathcal{S}_{\mathcal{I}^*}(\mathcal{G}^*) - \mathcal{S}_{\mathcal{I}^*}(\mathcal{G}) = \inf_\phi \sum_{k\in[K]} D_{KL}(p^{(k)}||f_{\mathcal{G}\mathcal{I}^*\phi}^{(k)}) + \lambda(|\mathcal{G}| - |\mathcal{G}^*|)\,. \tag{19}$$

*Proof.* By Lemma 13, we have that $|\mathcal{S}_{\mathcal{I}^*}(\mathcal{G}^*)| < \infty$, which ensures the difference $\mathcal{S}_{\mathcal{I}^*}(\mathcal{G}^*) - \mathcal{S}_{\mathcal{I}^*}(\mathcal{G})$ is well defined.

$$\mathcal{S}_{\mathcal{I}^*}(\mathcal{G}^*) - \mathcal{S}_{\mathcal{I}^*}(\mathcal{G}) \tag{20}$$

$$= \mathcal{S}_{\mathcal{I}^*}(\mathcal{G}^*) - \sum_{k\in[K]} \mathbb{E}_{p^{(k)}} \log p^{(k)}(X) - \mathcal{S}_{\mathcal{I}^*}(\mathcal{G}) + \sum_{k\in[K]} \mathbb{E}_{p^{(k)}} \log p^{(k)}(X) \tag{21}$$

$$= \sup_\phi \sum_{k\in[K]} \mathbb{E}_{p^{(k)}} \log f_{\mathcal{G}^*\mathcal{I}^*\phi}^{(k)}(X) - \sum_{k\in[K]} \mathbb{E}_{p^{(k)}} \log p^{(k)}(X)$$

$$- \sup_\phi \sum_{k\in[K]} \mathbb{E}_{p^{(k)}} \log f_{\mathcal{G}\mathcal{I}^*\phi}^{(k)}(X) + \sum_{k\in[K]} \mathbb{E}_{p^{(k)}} \log p^{(k)}(X)$$

$$+ \lambda(|\mathcal{G}| - |\mathcal{G}^*|) \tag{22}$$

$$= \inf_\phi - \sum_{k\in[K]} \mathbb{E}_{p^{(k)}} \log f_{\mathcal{G}\mathcal{I}^*\phi}^{(k)}(X) + \sum_{k\in[K]} \mathbb{E}_{p^{(k)}} \log p^{(k)}(X)$$

$$- \inf_\phi - \sum_{k\in[K]} \mathbb{E}_{p^{(k)}} \log f_{\mathcal{G}^*\mathcal{I}^*\phi}^{(k)}(X) - \sum_{k\in[K]} \mathbb{E}_{p^{(k)}} \log p^{(k)}(X)$$

$$+ \lambda(|\mathcal{G}| - |\mathcal{G}^*|) \tag{23}$$

$$= \inf_\phi \sum_{k\in[K]} D_{KL}(p^{(k)}||f_{\mathcal{G}\mathcal{I}^*\phi}^{(k)}) - \inf_\phi \sum_{k\in[K]} D_{KL}(p^{(k)}||f_{\mathcal{G}^*\mathcal{I}^*\phi}^{(k)})$$

$$+ \lambda(|\mathcal{G}| - |\mathcal{G}^*|) \tag{24}$$

The first equality holds since by Assumption 4 the differential entropy of $p^{(k)}$ is finite for all $k$. In (24), we use the linearity of the expectation, which holds because the entropy term is finite. By Assumption 1, $(p^{(k)})_{k\in[K]} \in \mathcal{F}_{\mathcal{I}^*}(\mathcal{G}^*)$ which implies that $\inf_\phi \sum_{k\in[K]} D_{KL}(p^{(k)}||f_{\mathcal{G}^*\mathcal{I}^*\phi}^{(k)}) = 0$. ∎

We will now prove three technical lemmas (Lemma 15, 16 & 18). Their proof can be safely skipped during a first reading.

Lemma 15 is adapted from Koller and Friedman [25, Theorem 8.7] to handle cases where infinite differential entropies might arise.

**Lemma 15** *Let $\mathcal{G}$ be a DAG. If $p \notin \mathcal{M}(\mathcal{G})$ and $p(x) > 0$ for all $x \in \mathbb{R}^d$, then*

$$\inf_{f\in\mathcal{M}(\mathcal{G})} D_{KL}(p||f) > 0\,.$$

*Proof.* We consider a new density function defined as

$$\hat{f}(x) := \prod_{j=1}^d p(x_j \mid x_{\pi_j^\mathcal{G}})\,, \tag{25}$$

where

$$p(x_j|x_{\pi_j^\mathcal{G}}) := \frac{p(x_j, x_{\pi_j^\mathcal{G}})}{p(x_{\pi_j^\mathcal{G}})}\,, \tag{26}$$

i.e. it is the conditional density. This should not be conflated with $p(x_j|x_{\pi_j^{\mathcal{G}*}})$. It should be clear from (25) and the fact that $p$ is strictly positive that $\hat{f} \in \mathcal{M}(\mathcal{G})$ hence $p \neq \hat{f}$. We will show that $\hat{f} \in \arg\min_{f \in \mathcal{M}(\mathcal{G})} D_{KL}(p||f)$.

Pick an arbitrary $f \in \mathcal{M}(\mathcal{G})$. We first show that $\mathbb{E}_p \log \frac{\hat{f}(X)}{f(X)}$ can be written as a sum of KL divergences.

$$\mathbb{E}_p \log \frac{\hat{f}(X)}{f(X)} = \mathbb{E}_p \sum_{j=1}^{d} \log \frac{p(X_j|X_{\pi_j^{\mathcal{G}}})}{f(X_j|X_{\pi_j^{\mathcal{G}}})} \tag{27}$$

$$= \sum_{j=1}^{d} \mathbb{E}_p \log \frac{p(X_j|X_{\pi_j^{\mathcal{G}}})}{f(X_j|X_{\pi_j^{\mathcal{G}}})} \tag{28}$$

In Equation (28), we apply the linearity of the Lebesgue integral, which holds as long as we are not summing infinities of opposite signs (in which case the sum is undefined).[5] We now show that it is not the case since each term is an expectation of a KL divergence, which is in $[0, +\infty]$:

$$\mathbb{E}_p \log \frac{p(X_j|X_{\pi_j^{\mathcal{G}}})}{f(X_j|X_{\pi_j^{\mathcal{G}}})} = \int p(x_{\pi_j^{\mathcal{G}}}) \int p(x_j \mid x_{\pi_j^{\mathcal{G}}}) \log \frac{p(x_j|x_{\pi_j^{\mathcal{G}}})}{f(x_j|x_{\pi_j^{\mathcal{G}}})} dx_j dx_{\pi_j^{\mathcal{G}}} \tag{29}$$

$$= \int p(x_{\pi_j^{\mathcal{G}}}) D_{KL}(p(\cdot_j \mid x_{\pi_j^{\mathcal{G}}})||f(\cdot_j \mid x_{\pi_j^{\mathcal{G}}}))) . \tag{30}$$

This implies that $\mathbb{E}_p \log \frac{\hat{f}(X)}{f(X)} \in [0, +\infty]$. We can now show that $\hat{f} \in \arg\min_{f \in \mathcal{M}(\mathcal{G})} D_{KL}(p||f)$:

$$D_{KL}(p||f) = \mathbb{E}_p \log \frac{p(X)}{\hat{f}(X)} \frac{\hat{f}(X)}{f(X)} \tag{31}$$

$$= \mathbb{E}_p \log \frac{p(X)}{\hat{f}(X)} + \mathbb{E}_p \log \frac{\hat{f}(X)}{f(X)} \tag{32}$$

$$= D_{KL}(p||\hat{f}) + \mathbb{E}_p \log \frac{\hat{f}(X)}{f(X)} \tag{33}$$

$$\geq D_{KL}(p||\hat{f}) > 0 . \tag{34}$$

Equation (32) holds as long as we do not have $\infty - \infty$. It is not the case here since (i) the first term is a KL divergence, so it is in $[0, +\infty]$, and (ii) the second term was already shown to be in $[0, +\infty]$. The very last inequality holds because $p \neq \hat{f}$.

We conclude by noting that $\inf_{f \in \mathcal{M}(\mathcal{G})} D_{KL}(p||f) = D_{KL}(p||\hat{f}) > 0$. ∎

The following lemma will make use of the following definition:

$$\mathcal{Z}(j, A) := \{(f^{(1)}, f^{(2)}) \mid f^{(1)}(x_j \mid x_A) = f^{(2)}(x_j \mid x_A) \text{ and } f^{(1)}, f^{(2)} > 0\} . \tag{35}$$

**Lemma 16** *Let $j \in V$ and $A \subset V \setminus \{j\}$. If $(p^{(1)}, p^{(2)}) \notin \mathcal{Z}(j, A)$ and both $p^{(1)}$ and $p^{(2)}$ are strictly positive, then*

$$\inf_{(f^{(1)}, f^{(2)}) \in \mathcal{Z}(j, A)} D_{KL}(p^{(1)}||f^{(1)}) + D_{KL}(p^{(2)}||f^{(2)}) > 0 .$$

*Proof.* The proof is very similar in spirit to the proof of Lemma 15.

We define new density functions:

$$p^{\text{mid}}(x) := \frac{p^{(1)}(x) + p^{(2)}(x)}{2} \tag{36}$$

$$\hat{f}^{(k)}(x) := p^{(k)}(x_A)p^{\text{mid}}(x_j \mid x_A)p^{(k)}(x_{V \setminus A \setminus j} \mid x_{A \cup j}) \ \forall k \in \{1, 2\}. \tag{37}$$

We note that $p^{\text{mid}}$, $\hat{f}^{(1)}$ and $\hat{f}^{(2)}$ are strictly positive since $p^{(1)}$ and $p^{(2)}$ are strictly positive. By construction, we have $\hat{f}^{(1)}(x_j|x_A) = \hat{f}^{(2)}(x_j|x_A)$, and thus $(\hat{f}^{(1)}, \hat{f}^{(2)}) \in \mathcal{Z}(i, A)$. This means that $\hat{f}^{(1)} \neq p^{(1)}$ or $\hat{f}^{(2)} \neq p^{(2)}$.

Pick an arbitrary $(f^{(1)}, f^{(2)}) \in \mathcal{Z}(j, A)$. We start by showing that the integral $\int p^{(1)}(x) \log \frac{\hat{f}^{(1)}(x)}{f^{(1)}(x)} + p^{(2)}(x) \log \frac{\hat{f}^{(2)}(x)}{f^{(2)}(x)} dx$ is in $[0, +\infty]$.

$$\int p^{(1)}(x) \log \frac{\hat{f}^{(1)}(x)}{f^{(1)}(x)} + p^{(2)}(x) \log \frac{\hat{f}^{(2)}(x)}{f^{(2)}(x)} dx \tag{38}$$

$$= \int p^{(1)}(x) \left[ \log \frac{p^{(1)}(x_A)}{f^{(1)}(x_A)} + \log \frac{p^{\text{mid}}(x_j \mid x_A)}{f^{(1)}(x_j \mid x_A)} + \log \frac{p^{(1)}(x_{V \setminus A \setminus j} \mid x_{A \cup j})}{f^{(1)}(x_{V \setminus A \setminus j} \mid x_{A \cup j})} \right]$$

$$+ p^{(2)}(x) \left[ \log \frac{p^{(2)}(x_A)}{f^{(2)}(x_A)} + \log \frac{p^{\text{mid}}(x_j \mid x_A)}{f^{(1)}(x_j \mid x_A)} + \log \frac{p^{(2)}(x_{V \setminus A \setminus j} \mid x_{A \cup j})}{f^{(2)}(x_{V \setminus A \setminus j} \mid x_{A \cup j})} \right] dx \tag{39}$$

$$= D_{KL}(p^{(1)}(\cdot_A)||f^{(1)}(\cdot_A)) + \mathbb{E}_{p^{(1)}} D_{KL}(p^{(1)}(\cdot_{V \setminus A \setminus j} \mid X_{A \cup j})||f^{(1)}(\cdot_{V \setminus A \setminus j} \mid X_{A \cup j}))$$

$$+ D_{KL}(p^{(2)}(\cdot_A)||f^{(2)}(\cdot_A)) + \mathbb{E}_{p^{(2)}} D_{KL}(p^{(2)}(\cdot_{V \setminus A \setminus j} \mid X_{A \cup j})||f^{(2)}(\cdot_{V \setminus A \setminus j} \mid X_{A \cup j}))$$

$$+ \underbrace{2 \int \frac{p^{(1)}(x) + p^{(2)}(x)}{2} \log \frac{p^{\text{mid}}(x_j \mid x_A)}{f^{(1)}(x_j \mid x_A)} dx}_{=\mathbb{E}_{p^{\text{mid}}} D_{KL}(p^{\text{mid}}(\cdot_j|x_A)||f^{(1)}(\cdot_j|x_A))}. \tag{40}$$

In (39), we used the fact that $f^{(1)}(x_j \mid x_A) = f^{(2)}(x_j \mid x_A)$. In (40), we use the linearity of the integral (which can be safely apply because each resulting "piece" is in $[0, +\infty]$). Since each term in (40) is in $[0, +\infty]$, their sum is in $[0, +\infty]$ as well.

We can now look at the sum of KL-divergences we are interested in.

$$D_{KL}(p^{(1)}||f^{(1)}) + D_{KL}(p^{(2)}||f^{(2)})$$

$$= \int p^{(1)}(x) \log \frac{p^{(1)}}{f^{(1)}} dx + \int p^{(2)}(x) \log \frac{p^{(2)}}{f^{(2)}} dx \tag{41}$$

$$= \int p^{(1)}(x) \log \frac{p^{(1)}}{f^{(1)}} + p^{(2)}(x) \log \frac{p^{(2)}}{f^{(2)}} dx \tag{42}$$

$$= \int p^{(1)}(x) \log \frac{p^{(1)}(x)}{\hat{f}^{(1)}(x)} + p^{(1)}(x) \log \frac{\hat{f}^{(1)}(x)}{f^{(1)}(x)} + p^{(2)}(x) \log \frac{p^{(2)}(x)}{\hat{f}^{(2)}(x)} + p^{(2)}(x) \log \frac{\hat{f}^{(2)}(x)}{f^{(2)}(x)} dx \tag{43}$$

$$= D_{KL}(p^{(1)}||\hat{f}^{(1)}) + D_{KL}(p^{(2)}||\hat{f}^{(2)}) + \int p^{(1)}(x) \log \frac{\hat{f}^{(1)}(x)}{f^{(1)}(x)} + p^{(2)}(x) \log \frac{\hat{f}^{(2)}(x)}{f^{(2)}(x)} dx \tag{44}$$

$$\geq D_{KL}(p^{(1)}||\hat{f}^{(1)}) + D_{KL}(p^{(2)}||\hat{f}^{(2)}) > 0. \tag{45}$$

In (42), we use the linearity of the integral (which can be safely applied given the initial integrals were in $[0, +\infty]$). In (44), we again use the linearity of the integral (which is, again, possible because each resulting piece are in $[0, +\infty]$). In (45), we use the fact that $\int p^{(1)}(x) \log \frac{\hat{f}^{(1)}(x)}{f^{(1)}(x)} + p^{(2)}(x) \log \frac{\hat{f}^{(2)}(x)}{f^{(2)}(x)} dx \in [0, +\infty]$ to get the $\geq$ while the strict inequality holds because either $\hat{f}^{(1)} \neq p^{(1)}$ or $\hat{f}^{(k)} \neq p^{(k)}$.

This implies that

$$\inf_{(f^{(1)}, f^{(2)}) \in \mathcal{Z}(j,A)} D_{KL}(p^{(1)}||f^{(1)}) + D_{KL}(p^{(2)}||f^{(2)}) = D_{KL}(p^{(1)}||\hat{f}^{(1)}) + D_{KL}(p^{(2)}||\hat{f}^{(2)}) > 0. \blacksquare$$

The following definition will be useful for the next lemma.

**Definition 17** *Given a DAG $\mathcal{G}$ with node set $V$ and two nodes $i, j \in V$, we define the following sets:*

$$T_{ij}^{\mathcal{G}} := \{\ell \in V \mid \text{the immorality } i \to \ell \leftarrow j \text{ is in } \mathcal{G}\} \tag{46}$$

$$L_{ij}^{\mathcal{G}} := \mathbf{DE}_{\mathcal{G}}(T_{ij}^{\mathcal{G}}) \cup \{i, j\}, \tag{47}$$

*where $\mathbf{DE}_{\mathcal{G}}(S)$ is the set of descendants of $S$ in $\mathcal{G}$, including $S$ itself.*

**Lemma 18** *Let $\mathcal{G}$ be a DAG with node set $V$. When $i \to j \notin \mathcal{G}$ and $i \leftarrow j \notin \mathcal{G}$ we have*

$$i \perp\!\!\!\perp_{\mathcal{G}} j \mid V \setminus L_{ij}^{\mathcal{G}}. \tag{48}$$

*Proof:* By contradiction. Suppose there is a path from $(i = a_0, a_1, ..., a_p = j)$ with $p > 1$ which is not d-blocked by $V \setminus L_{ij}^{\mathcal{G}}$ in $\mathcal{G}$. We first consider the case where the path contains no colliders.

If the path contains no colliders, then $a_0 \leftarrow a_1$ or $a_{p-1} \to a_p$. Moreover, since the path is not d-blocked and both $a_1$ and $a_{p-1}$ are not colliders, $a_1, a_{p-1} \in L_{ij}^{\mathcal{G}}$. But this implies that there is a directed path from $i = a_0$ to $a_1$ and a directed path from $j = a_p$ to $a_{p-1}$. This creates a directed cycle: either $a_0 \to \cdots \to a_1 \to a_0$ or $a_p \to \cdots \to a_{p-1} \to a_p$. This is a contradiction since $\mathcal{G}$ is acyclic.

Suppose there is a collider $a_k$, i.e. $a_{k-1} \to a_k \leftarrow a_{k+1}$. Since the path is not d-blocked, there must exists a node $z \in \mathbf{DE}_{\mathcal{G}}(a_k) \cup \{a_k\}$ such that $z \notin L_{ij}^{\mathcal{G}}$. If $i = a_{k-1}$ and $j = a_{k+1}$, then clearly $z \in L_{ij}^{\mathcal{G}}$, which is a contradiction. Otherwise, $i \neq a_{k-1}$ or $j \neq a_{k+1}$. Without loss of generality, assume $i \neq a_{k-1}$. Clearly, $a_{k-1}$ is not a collider and since the path is not d-blocked, $a_{k-1} \in L_{ij}^{\mathcal{G}}$. But by definition, $L_{ij}^{\mathcal{G}}$ also contains all the descendants of $a_{k-1}$ including $z$. Again, this is a contradiction with $z \notin L_{ij}^{\mathcal{G}}$. ∎

We recall Theorem 1 from Section 3.1 and present its proof.

**Theorem 1 (Identification via score maximization)** *Suppose the interventional family $\mathcal{I}^*$ is such that $I_1^* := \emptyset$. Let $\mathcal{G}^*$ be the ground truth DAG and $\hat{\mathcal{G}} \in \arg\max_{\mathcal{G} \in DAG} \mathcal{S}_{\mathcal{I}^*}(\mathcal{G})$. Assume that the density model has enough capacity to represent the ground truth distributions, that $\mathcal{I}^*$-faithfulness holds, that the density model is strictly positive and that the ground truth densities $p^{(k)}$ have finite differential entropy, respectively Assumptions 1, 2, 3 & 4. Then for $\lambda > 0$ small enough, we have that $\hat{\mathcal{G}}$ is $\mathcal{I}^*$-Markov equivalent to $\mathcal{G}^*$.*

*Proof.* It is sufficient to prove that, for all $\mathcal{G} \notin \mathcal{I}^*\text{-MEC}(\mathcal{G}^*)$, $\mathcal{S}_{\mathcal{I}^*}(\mathcal{G}^*) > \mathcal{S}_{\mathcal{I}^*}(\mathcal{G})$. We use Theorem 10 which states that $\hat{\mathcal{G}}$ is not $\mathcal{I}^*$-Markov equivalent to $\mathcal{G}^*$ if and only if $\hat{\mathcal{G}}^{\mathcal{I}^*}$ does not share its skeleton or its immoralities with $\mathcal{G}^{*\mathcal{I}^*}$. The proof is organized in six cases. Cases 1-2 treat when $\mathcal{G}$ and $\mathcal{G}^*$ do not share the same skeleton, cases 3 & 4 when their immoralities differ and cases 5 & 6 when their immoralities implying interventional nodes $\zeta_k$ differ. In almost every cases, the idea is the same:

1. Use Lemma 18 to find a d-separation which holds in $\mathcal{G}^{\mathcal{I}^*}$ and show it does not hold in $\mathcal{G}^{*\mathcal{I}^*}$;

2. Use the fact that $\mathcal{F}_{\mathcal{I}}(\mathcal{G}) \subset \mathcal{M}_{\mathcal{I}}(\mathcal{G})$ (by strict positivity), Proposition 9 and the $\mathcal{I}^*$-faithfulness assumption to obtain an invariance which holds for all $(f^{(k)})_{k \in [K]} \in \mathcal{F}_{\mathcal{I}^*}(\mathcal{G})$ but not in $(p^{(k)})_{k \in [K]}$;

3. Use the fact that the invariance forces $\inf_\phi \sum_{k \in [K]} D_{KL}(p^{(k)} || f^{(k)}_{\mathcal{G}\mathcal{I}^* \phi})$ to be greater than zero (by Lemma 15 or 16) and;

4. Conclude that $\mathcal{S}_{\mathcal{I}^*}(\mathcal{G}^*) > \mathcal{S}_{\mathcal{I}^*}(\mathcal{G})$ via Lemma 14.

In this proof, we are exclusively referring to $\mathcal{I}^*$. Thus for notational convenience, we set $\mathcal{I} := \mathcal{I}^*$.

**Case 1:** We consider the graphs $\mathcal{G}$ such that there exists $i \to j \in \mathcal{G}^*$ but $i \to j \notin \mathcal{G}$ and $i \leftarrow j \notin \mathcal{G}$. Let $\mathbb{G}$ be the set of all such $\mathcal{G}$. By Lemma 18, $i \perp\!\!\!\perp_{\mathcal{G}} j \mid V \setminus L_{ij}^{\mathcal{G}}$ but clearly $i \not\perp\!\!\!\perp_{\mathcal{G}^*} j \mid V \setminus L_{ij}^{\mathcal{G}}$. Hence, by $\mathcal{I}$-faithfulness (Assumption 2) we have $X_i \not\perp\!\!\!\perp_{p^{(1)}} X_j | X_{V \setminus L_{ij}^{\mathcal{G}}}$. It implies that $p^{(1)} \notin \mathcal{M}(\mathcal{G})$, by Proposition 4.

For notation convenience, let us define

$$\eta(\mathcal{G}) := \inf_{\phi} \sum_{k \in [K]} D_{KL}(p^{(k)} || f^{(k)}_{\mathcal{GI}\phi}) \,. \tag{49}$$

Note that

$$\eta(\mathcal{G}) \geq \inf_{\phi} D_{KL}(p^{(1)} || f^{(1)}_{\mathcal{GI}\phi}) \geq \inf_{f \in \mathcal{M}(\mathcal{G})} D_{KL}(p^{(1)} || f) > 0 \,, \tag{50}$$

where the first inequality holds by non-negativity of the KL divergence, the second holds because, for all $\phi$, $f^{(1)}_{\mathcal{GI}\phi} \in \mathcal{M}(\mathcal{G})$ and the third holds by Lemma 15 (which applies here because $p^{(1)} \notin \mathcal{M}(\mathcal{G})$). Using Lemma 14, we can write

$$\mathcal{S}_{\mathcal{I}}(\mathcal{G}^*) - \mathcal{S}_{\mathcal{I}}(\mathcal{G}) = \eta(\mathcal{G}) + \lambda(|\mathcal{G}| - |\mathcal{G}^*|) \,. \tag{51}$$

If $|\mathcal{G}| \geq |\mathcal{G}^*|$ then clearly $\mathcal{S}_{\mathcal{I}}(\mathcal{G}^*) - \mathcal{S}_{\mathcal{I}}(\mathcal{G}) > 0$. Let $\mathbb{G}^+ := \{\mathcal{G} \in \mathbb{G} \mid |\mathcal{G}| < |\mathcal{G}^*|\}$. To make sure we have $\mathcal{S}_{\mathcal{I}}(\mathcal{G}^*) - \mathcal{S}_{\mathcal{I}}(\mathcal{G}) > 0$ for all $\mathcal{G} \in \mathbb{G}^+$, we need to pick $\lambda$ sufficiently small. Choosing $0 < \lambda < \min_{\mathcal{G} \in \mathbb{G}^+} \frac{\eta(\mathcal{G})}{|\mathcal{G}^*| - |\mathcal{G}|}$ is sufficient since (and note that minimum exists because the set $\mathbb{G}^+$ is finite and is strictly positive by (50)):

$$\lambda < \min_{\mathcal{G} \in \mathbb{G}^+} \frac{\eta(\mathcal{G})}{|\mathcal{G}^*| - |\mathcal{G}|} \tag{52}$$

$$\iff \lambda < \frac{\eta(\mathcal{G})}{|\mathcal{G}^*| - |\mathcal{G}|} \ \ \forall \mathcal{G} \in \mathbb{G}^+ \tag{53}$$

$$\iff \lambda(|\mathcal{G}^*| - |\mathcal{G}|) < \eta(\mathcal{G}) \ \ \forall \mathcal{G} \in \mathbb{G}^+ \tag{54}$$

$$\iff 0 < \eta(\mathcal{G}) + \lambda(|\mathcal{G}| - |\mathcal{G}^*|) = \mathcal{S}_{\mathcal{I}}(\mathcal{G}^*) - \mathcal{S}_{\mathcal{I}}(\mathcal{G}) \ \ \forall \mathcal{G} \in \mathbb{G}^+ \,. \tag{55}$$

**Case 2:** We consider the graphs $\mathcal{G}$ such that there exists $i \to j \in \mathcal{G}$ but $i \to j \notin \mathcal{G}^*$ and $i \leftarrow j \notin \mathcal{G}^*$. We can assume $k \to \ell \in \mathcal{G}^*$ implies $k \to \ell \in \mathcal{G}$ or $k \leftarrow \ell \in \mathcal{G}$, since otherwise we are in Case 1. Hence, it means $|\mathcal{G}| > |\mathcal{G}^*|$ which in turn implies that $\mathcal{S}_{\mathcal{I}}(\mathcal{G}^*) > \mathcal{S}_{\mathcal{I}}(\mathcal{G})$.

Cases 1 and 2 completely cover the situations where $\mathcal{G}^{\mathcal{I}}$ and $\mathcal{G}^{*\mathcal{I}}$ do not share the same skeleton. Next, we assume that $\mathcal{G}^{\mathcal{I}}$ and $\mathcal{G}^{*\mathcal{I}}$ do have the same skeleton (which implies that $|\mathcal{G}| = |\mathcal{G}^*|$). The remaining cases treat the differences in immoralities.

**Case 3:** Suppose $\mathcal{G}^*$ contains an immorality $i \to \ell \leftarrow j$ which is not present in $\mathcal{G}$. We first show that $\ell \notin L^{\mathcal{G}}_{ij}$. Suppose the opposite. This means $\ell$ is a descendant of both $i$ and $j$ in $\mathcal{G}$. Since $\mathcal{G}$ and $\mathcal{G}^*$ share skeleton and because $i \to \ell \leftarrow j$ is not an immorality in $\mathcal{G}$, we have that $i \leftarrow \ell \in \mathcal{G}$ or $\ell \to j \in \mathcal{G}$, which in both cases creates a cycle. This is a contradiction.

The path $(i, \ell, j)$ is not d-blocked by $V \setminus L^{\mathcal{G}}_{ij}$ in $\mathcal{G}^*$ since $\ell \in V \setminus L^{\mathcal{G}}_{ij}$. By $\mathcal{I}$-faithfulness (Assumption 2), this means that $X_i \not\perp\!\!\!\perp_{p^{(1)}} X_j \mid X_{V \setminus L^{\mathcal{G}}_{ij}}$. Since $\mathcal{G}^*$ and $\mathcal{G}$ share the same skeleton, we know $i \to j$ and $i \leftarrow j$ are not in $\mathcal{G}$. Using Lemma 18, we have that $i \perp\!\!\!\perp_{\mathcal{G}} j \mid V \setminus L^{\mathcal{G}}_{ij}$. Hence by Proposition 4, $p^{(1)} \notin \mathcal{M}(\mathcal{G})$. Similarly to Case 1, this implies that $\eta(\mathcal{G}) > 0$ which in turn implies that $\mathcal{S}_{\mathcal{I}}(\mathcal{G}^*) - \mathcal{S}_{\mathcal{I}}(\mathcal{G}) > 0$ (using the fact $|\mathcal{G}^*| = |\mathcal{G}|$).

**Case 4:** Suppose $\mathcal{G}$ contains an immorality $i \to \ell \leftarrow j$ which is not present in $\mathcal{G}^*$. Since $\mathcal{G}$ and $\mathcal{G}^*$ share the same skeleton and $\ell \notin V \setminus L^{\mathcal{G}}_{ij}$, we know there is a (potentially undirected) path $(i, \ell, j)$ which is not d-blocked by $V \setminus L^{\mathcal{G}}_{ij}$ in $\mathcal{G}^*$. By $\mathcal{I}$-faithfulness (Assumption 2), we know that $X_i \not\perp\!\!\!\perp_{p^{(1)}} X_j \mid X_{V \setminus L^{\mathcal{G}}_{ij}}$. However by Lemma 18, we have that $i \perp\!\!\!\perp_{\mathcal{G}} j \mid V \setminus L^{\mathcal{G}}_{ij}$, which implies, again by Proposition 4, that $p^{(1)} \notin \mathcal{M}(\mathcal{G})$. Thus, again by the same argument as Case 3, $\mathcal{S}_{\mathcal{I}}(\mathcal{G}^*) - \mathcal{S}_{\mathcal{I}}(\mathcal{G}) > 0$.

So far, all cases did not require interventional nodes $\zeta_k$. Cases 5 and 6 treat the difference in immoralities implying interventional nodes $\zeta_k$. Note that the arguments are analog to cases 3 and 4.

**Case 5:** Suppose that there is an immorality $i \to \ell \leftarrow \zeta_j$ in $\mathcal{G}^{*\mathcal{I}}$ which does not appear in $\mathcal{G}^{\mathcal{I}}$. The path $(i, \ell, \zeta_j)$ is not d-blocked by $\zeta_{-j} \cup V \setminus L^{\mathcal{G}^{\mathcal{I}}}_{i\zeta_j}$ in $\mathcal{G}^{*\mathcal{I}}$ since $\ell \in \zeta_{-j} \cup V \setminus L^{\mathcal{G}^{\mathcal{I}}}_{i\zeta_j}$ (by same argument as presented in Case 3). By $\mathcal{I}$-faithfulness (Assumption 2), this means that

$$p^{(1)}(x_i \mid x_{V \setminus L^{\mathcal{G}^{\mathcal{I}}}_{i\zeta_j}}) \neq p^{(j)}(x_i \mid x_{V \setminus L^{\mathcal{G}^{\mathcal{I}}}_{i\zeta_j}}) \,. \tag{56}$$

Thus, $(p^{(1)}, p^{(j)}) \notin \mathcal{Z}(i, V \setminus L^{\mathcal{G}^{\mathcal{I}}}_{i\zeta_j})$ (defined in Equation (35)).

On the other hand, Lemma 18 implies that $i \perp\!\!\!\perp_{\mathcal{G}^{\mathcal{I}}} \zeta_j \mid \zeta_{-j} \cup V \setminus L^{\mathcal{G}^{\mathcal{I}}}_{i\zeta_j}$. Thus by Proposition 9 and since $\mathcal{F}_{\mathcal{I}}(\mathcal{G}) \subset \mathcal{M}_{\mathcal{I}}(\mathcal{G})$, we have that for all $\phi$,

$$f^{(1)}_{\mathcal{GI}\phi}(x_i \mid x_{V \setminus L^{\mathcal{G}^{\mathcal{I}}}_{i\zeta_j}}) = f^{(j)}_{\mathcal{GI}\phi}(x_i \mid x_{V \setminus L^{\mathcal{G}^{\mathcal{I}}}_{i\zeta_j}}) \text{ i.e. } (f^{(1)}_{\mathcal{GI}\phi}, f^{(j)}_{\mathcal{GI}\phi}) \in \mathcal{Z}(i, V \setminus L^{\mathcal{G}^{\mathcal{I}}}_{i\zeta_j}). \tag{57}$$

This means that $\mathcal{S}_{\mathcal{I}}(\mathcal{G}^*) > \mathcal{S}_{\mathcal{I}}(\mathcal{G})$ since

$$\mathcal{S}_{\mathcal{I}}(\mathcal{G}^*) - \mathcal{S}_{\mathcal{I}}(\mathcal{G}) = \inf_{\phi} \sum_{k \in [K]} D_{KL}(p^{(k)} || f^{(k)}_{\mathcal{GI}\phi}) \tag{58}$$

$$\geq \inf_{\phi} D_{KL}(p^{(1)} || f^{(1)}_{\mathcal{GI}\phi}) + D_{KL}(p^{(j)} || f^{(j)}_{\mathcal{GI}\phi}) \tag{59}$$

$$\geq \inf_{(f^{(1)}, f^{(j)}) \in \mathcal{Z}(i, V \setminus L^{\mathcal{G}^{\mathcal{I}}}_{i\zeta_j})} D_{KL}(p^{(1)} || f^{(1)}) + D_{KL}(p^{(j)} || f^{(j)}) \tag{60}$$

$$> 0. \tag{61}$$

In (60), we use the fact that, for all $\phi$, $(f^{(1)}_{\mathcal{GI}\phi}, f^{(j)}_{\mathcal{GI}\phi}) \in \mathcal{Z}(i, V \setminus L^{\mathcal{G}^{\mathcal{I}}}_{i\zeta_j})$. The very last strict inequality holds by Lemma 16, which applies here because $(p^{(1)}, p^{(j)}) \notin \mathcal{Z}(i, V \setminus L^{\mathcal{G}^{\mathcal{I}}}_{i\zeta_j})$.

**Case 6:** Suppose that there is an immorality $i \to \ell \leftarrow \zeta_j$ in $\mathcal{G}^{\mathcal{I}}$ which does not appear in $\mathcal{G}^{*\mathcal{I}}$. The path $(i, \ell, \zeta_j)$ is not d-blocked by $\zeta_{-j} \cup V \setminus L^{\mathcal{G}^{\mathcal{I}}}_{i\zeta_j}$ in $\mathcal{G}^{*\mathcal{I}}$, since $\ell \notin \zeta_{-j} \cup V \setminus L^{\mathcal{G}^{\mathcal{I}}}_{i\zeta_j}$ and both $\mathcal{I}$-DAGs share the same skeleton. It follows by $\mathcal{I}$-faithfulness (Assumption 2) that

$$p^{(1)}(x_i \mid x_{V \setminus L^{\mathcal{G}^{\mathcal{I}}}_{i\zeta_j}}) \neq p^{(j)}(x_i \mid x_{V \setminus L^{\mathcal{G}^{\mathcal{I}}}_{i\zeta_j}}). \tag{62}$$

On the other hand, Lemma 18 implies that $i \perp\!\!\!\perp_{\mathcal{G}^{\mathcal{I}}} \zeta_j \mid \zeta_{-j} \cup V \setminus L^{\mathcal{G}^{\mathcal{I}}}_{i\zeta_j}$. Again by the $\mathcal{I}$-Markov property (Proposition 9), it means that, for all $\phi$,

$$f^{(1)}_{\mathcal{GI}\phi}(x_i \mid x_{V \setminus L^{\mathcal{G}^{\mathcal{I}}}_{i\zeta_j}}) = f^{(j)}_{\mathcal{GI}\phi}(x_i \mid x_{V \setminus L^{\mathcal{G}^{\mathcal{I}}}_{i\zeta_j}}). \tag{63}$$

By an argument identical to that of Case 5, it follows that $\mathcal{S}_{\mathcal{I}}(\mathcal{G}^*) > \mathcal{S}_{\mathcal{I}}(\mathcal{G})$.

The proof is complete since there is no other way in which $\mathcal{G}^{\mathcal{I}}$ and $\mathcal{G}^{*\mathcal{I}}$ can differ in terms of skeleton and immoralities. ∎

### A.3 Theory for unknown targets

Theorem 1 assumes implicitly that, for each intervention $k$, the ground truth interventional target $I_k^*$ is known. What if we do not have access to this information? We now present an extension of Theorem 1 to unknown targets. In this setting, the interventional family $\mathcal{I}$ is learned similarly to $\mathcal{G}$. We denote the ground truth interventional family by $\mathcal{I}^* := (I_1^*, \cdots, I_K^*)$ and assume that $I_1^* := \emptyset$. We first recall score introduced in Section 3.3:

$$\mathcal{S}(\mathcal{G}, \mathcal{I}) := \sup_{\phi} \sum_{k=1}^{K} \mathbb{E}_{X \sim p^{(k)}} \log f^{(k)}(X; M^{\mathcal{G}}, R^{\mathcal{I}}, \phi) - \lambda|\mathcal{G}| - \lambda_R|\mathcal{I}|, \tag{64}$$

where $f^{(k)}(X; M^{\mathcal{G}}, R^{\mathcal{I}}, \phi)$ was defined in (15) and $|\mathcal{I}| = \sum_{k=1}^{K} |I_k|$. Notice that the assumption that $I_1^* = \emptyset$ is integrated in the joint density of (15) with $k = 1$ (the row vector $R^{\mathcal{I}}_{1:}$ has no effect). The only difference between $\mathcal{S}_{\mathcal{I}^*}(\mathcal{G})$ and $\mathcal{S}(\mathcal{G}, \mathcal{I})$ is that, in the latter, $\mathcal{I}$ is considered a variable and the extra regularizing term $-\lambda_R|\mathcal{I}|$.

The result of this section relies on the exact same assumptions as those of Theorem 1, namely Assumptions 1, 2, 3 & 4.

The next Lemma is an adaptation of Lemma 14 to this new setting.

**Lemma 19 (Rewriting of score differences)** *Under Assumption 1 & 4, we have*

$$\mathcal{S}(\mathcal{G}^*, \mathcal{I}^*) - \mathcal{S}(\mathcal{G}, \mathcal{I}) = \inf_{\phi} \sum_{k \in [K]} D_{KL}(p^{(k)} || f_{\mathcal{G}\mathcal{I}\phi}^{(k)}) + \lambda(|\mathcal{G}| - |\mathcal{G}^*|) + \lambda_R(|\mathcal{I}| - |\mathcal{I}^*|). \tag{65}$$

*Proof.* We note that $|\mathcal{S}(\mathcal{G}^*, \mathcal{I}^*)| = |\mathcal{S}_{\mathcal{I}^*}(\mathcal{G}^*) - \lambda_R|\mathcal{I}^*|| < \infty$, by Lemma 13. This implies that the difference $\mathcal{S}(\mathcal{G}^*, \mathcal{I}^*) - \mathcal{S}(\mathcal{G}, \mathcal{I})$ is always well defined.

The rest of the proof is identical to Lemma 14.∎

We are now ready to state and prove our identifiability result for unknown targets.

**Theorem 2 (Unknown targets identification)** *Suppose $\mathcal{I}^*$ is such that $I_1^* := \emptyset$. Let $\mathcal{G}^*$ be the ground truth DAG and $(\hat{\mathcal{G}}, \hat{\mathcal{I}}) \in \arg\max_{\mathcal{G} \in DAG, \mathcal{I}} \mathcal{S}(\mathcal{G}, \mathcal{I})$. Under the same assumptions as Theorem 1 and for $\lambda, \lambda_R > 0$ small enough, $\hat{\mathcal{G}}$ is $\mathcal{I}^*$-Markov equivalent to $\mathcal{G}^*$ and $\hat{\mathcal{I}} = \mathcal{I}^*$.*

*Proof:* We simply add two cases at the beginning of the proof of Theorem 1 to handle cases where $\mathcal{I} \neq \mathcal{I}^*$ (we will denote them by Case 0.1 and Case 0.2). Similarly to Theorem 1, it is sufficient to prove that, whenever $\mathcal{G} \notin \mathcal{I}^*$-MEC($\mathcal{G}^*$) or $\mathcal{I} \neq \mathcal{I}^*$, we have that $\mathcal{S}(\mathcal{G}^*, \mathcal{I}^*) > \mathcal{S}(\mathcal{G}, \mathcal{I})$. For convenience, let us define

$$\eta(\mathcal{G}, \mathcal{I}) := \inf_{\phi} \sum_{k \in [K]} D_{KL}(p^{(k)} || f_{\mathcal{G}\mathcal{I}\phi}^{(k)}). \tag{66}$$

**Case 0.1:** Let $\mathbb{I}$ be the set of all $\mathcal{I}$ such that there exists $k_0 \in [K]$ and $j \in [d]$ such that $j \in I_{k_0}^*$ but $j \notin I_{k_0}$. Let $\mathcal{I} \in \mathbb{I}$ and let $\mathcal{G}$ be an arbitrary DAG.

Since the edge $\zeta_{k_0} \to j$ is in $\mathcal{G}^{*\mathcal{I}^*}$, we have that $\zeta_{k_0}$ and $j$ are never d-separated. By $\mathcal{I}^*$-faithfulness (Assumption 2), we have that

$$p^{(1)}(x_j | x_{\pi_j^{\mathcal{G}}}) \neq p^{(k_0)}(x_j | x_{\pi_j^{\mathcal{G}}}). \tag{67}$$

Note that this is true for any conditioning set. It means $(p^{(1)}, p^{(k_0)}) \notin \mathcal{Z}(j, \pi_j^{\mathcal{G}})$ (defined in (35)).

Since $j \notin I_k$, we have by definition from (15) that, for all $\phi$,

$$f_{\mathcal{G}\mathcal{I}\phi}^{(1)}(x_j | x_{\pi_j^{\mathcal{G}}}) = f_{\mathcal{G}\mathcal{I}\phi}^{(k_0)}(x_j | x_{\pi_j^{\mathcal{G}}}) \text{ i.e. } (f_{\mathcal{G}\mathcal{I}\phi}^{(1)}, f_{\mathcal{G}\mathcal{I}\phi}^{(k_0)}) \in \mathcal{Z}(j, \pi_j^{\mathcal{G}}). \tag{68}$$

This implies that

$$\eta(\mathcal{G}, \mathcal{I}) \geq \inf_{\phi} D_{KL}(p^{(1)} || f_{\mathcal{G}\mathcal{I}\phi}^{(1)}) + D_{KL}(p^{(k_0)} || f_{\mathcal{G}\mathcal{I}\phi}^{(k_0)}) \tag{69}$$

$$\geq \inf_{(f^{(1)}, f^{(k_0)}) \in \mathcal{Z}(j, \pi_j^{\mathcal{G}})} D_{KL}(p^{(1)} || f^{(1)}) + D_{KL}(p^{(k_0)} || f^{(k_0)}) \tag{70}$$

$$> 0, \tag{71}$$

where (70) holds because, for all $\phi$, $(f_{\mathcal{G}\mathcal{I}\phi}^{(1)}, f_{\mathcal{G}\mathcal{I}\phi}^{(k_0)}) \in \mathcal{Z}(j, \pi_j^{\mathcal{G}})$ and (71) holds by Lemma 16.

If $\min\{|\mathcal{G}| - |\mathcal{G}^*|, |\mathcal{I}| - |\mathcal{I}^*|\} \geq 0$, then clearly $\mathcal{S}(\mathcal{G}^*, \mathcal{I}^*) - \mathcal{S}(\mathcal{G}, \mathcal{I}) > 0$. Let $\mathbb{S} := \{(\mathcal{G}, \mathcal{I}) \in DAG \times \mathbb{I} \mid \min\{|\mathcal{G}| - |\mathcal{G}^*|, |\mathcal{I}| - |\mathcal{I}^*|\} < 0\}$. To make sure we have $\mathcal{S}(\mathcal{G}^*, \mathcal{I}^*) - \mathcal{S}(\mathcal{G}, \mathcal{I}) > 0$ for all $(\mathcal{G}, \mathcal{I}) \in \mathbb{S}$, we need to pick $\lambda$ and $\lambda_R$ sufficiently small. Choosing $\lambda + \lambda_R < \min_{(\mathcal{G}, \mathcal{I}) \in \mathbb{S}} \frac{\eta(\mathcal{G}, \mathcal{I})}{-\min\{|\mathcal{G}| - |\mathcal{G}^*|, |\mathcal{I}| - |\mathcal{I}^*|\}}$ is sufficient since (and note that the minimum exists because the set $\mathbb{S}$ is finite, and is strictly positive by (71)):

$$\lambda + \lambda_R < \min_{(\mathcal{G}, \mathcal{I}) \in \mathbb{S}} \frac{\eta(\mathcal{G}, \mathcal{I})}{-\min\{|\mathcal{G}| - |\mathcal{G}^*|, |\mathcal{I}| - |\mathcal{I}^*|\}} \tag{72}$$

$$\iff \lambda + \lambda_R < \frac{\eta(\mathcal{G}, \mathcal{I})}{-\min\{|\mathcal{G}| - |\mathcal{G}^*|, |\mathcal{I}| - |\mathcal{I}^*|\}} \quad \forall (\mathcal{G}, \mathcal{I}) \in \mathbb{S} \tag{73}$$

$$\iff -(\lambda + \lambda_R)\min\{|\mathcal{G}| - |\mathcal{G}^*|, |\mathcal{I}| - |\mathcal{I}^*|\} < \eta(\mathcal{G}, \mathcal{I}) \quad \forall (\mathcal{G}, \mathcal{I}) \in \mathbb{S} \tag{74}$$

$$\iff 0 < \eta(\mathcal{G}, \mathcal{I}) + (\lambda + \lambda_R)\min\{|\mathcal{G}| - |\mathcal{G}^*|, |\mathcal{I}| - |\mathcal{I}^*|\} \quad \forall (\mathcal{G}, \mathcal{I}) \in \mathbb{S} \tag{75}$$

$$\leq \eta(\mathcal{G}, \mathcal{I}) + \lambda(|\mathcal{G}| - |\mathcal{G}^*|) + \lambda_R(|\mathcal{I}| - |\mathcal{I}^*|) \tag{76}$$

$$= \mathcal{S}(\mathcal{G}^*, \mathcal{I}^*) - \mathcal{S}(\mathcal{G}, \mathcal{I}). \tag{77}$$

From now on, we can assume $I_k^* \subset I_k$ for all $k \in [K]$, since otherwise we are in Case 0.1.

**Case 0.2:** Let $\bar{\bar{\mathbb{I}}} := \{\mathcal{I} \mid [I_k^* \subset I_k \; \forall k] \text{ and } [\exists \, k_0, j \text{ s.t. } j \in I_{k_0} \text{ and } j \notin I_{k_0}^*]\}$. Let $\mathcal{I} \in \bar{\bar{\mathbb{I}}}$ and let $\mathcal{G}$ be a DAG. We can already notice that $|\mathcal{I}| > |\mathcal{I}^*|$.

If $|\mathcal{G}| \geq |\mathcal{G}^*|$, then $\mathcal{S}(\mathcal{G}^*, \mathcal{I}^*) - \mathcal{S}(\mathcal{G}, \mathcal{I}) > 0$ by (65). Let $\bar{\bar{\mathbb{S}}} := \{(\mathcal{G}, \mathcal{I}) \in \text{DAG} \times \bar{\bar{\mathbb{I}}} \mid |\mathcal{G}| < |\mathcal{G}^*|\}$. To make sure $\mathcal{S}(\mathcal{G}^*, \mathcal{I}^*) - \mathcal{S}(\mathcal{G}, \mathcal{I}) > 0$ for all $(\mathcal{G}, \mathcal{I}) \in \bar{\bar{\mathbb{S}}}$, we need to pick $\lambda$ sufficiently small. Choosing $\lambda < \min_{(\mathcal{G}, \mathcal{I}) \in \bar{\bar{\mathbb{S}}}} \frac{\eta(\mathcal{G}, \mathcal{I}) + \lambda_R(|\mathcal{I}| - |\mathcal{I}^*|)}{|\mathcal{G}^*| - |\mathcal{G}|}$ is sufficient since this implies

$$\lambda < \frac{\eta(\mathcal{G}, \mathcal{I}) + \lambda_R(|\mathcal{I}| - |\mathcal{I}^*|)}{|\mathcal{G}^*| - |\mathcal{G}|} \; \forall \, (\mathcal{G}, \mathcal{I}) \in \bar{\bar{\mathbb{S}}} \tag{78}$$

$$\iff \lambda(|\mathcal{G}^*| - |\mathcal{G}|) < \eta(\mathcal{G}, \mathcal{I}) + \lambda_R(|\mathcal{I}| - |\mathcal{I}^*|) \; \forall \, (\mathcal{G}, \mathcal{I}) \in \bar{\bar{\mathbb{S}}} \tag{79}$$

$$\iff 0 < \eta(\mathcal{G}, \mathcal{I}) + \lambda(|\mathcal{G}| - |\mathcal{G}^*|) + \lambda_R(|\mathcal{I}| - |\mathcal{I}^*|) \; \forall \, (\mathcal{G}, \mathcal{I}) \in \bar{\bar{\mathbb{S}}} \tag{80}$$

$$= \mathcal{S}(\mathcal{G}^*, \mathcal{I}^*) - \mathcal{S}(\mathcal{G}, \mathcal{I}). \tag{81}$$

Cases 0.1 & 0.2 cover all situations where $\mathcal{I} \neq \mathcal{I}^*$. This implies that $\hat{\mathcal{I}} = \mathcal{I}^*$. For the rest of the proof, we can assume that $\mathcal{I} = \mathcal{I}^*$. By noting that $\mathcal{S}(\mathcal{G}^*, \mathcal{I}^*) - \mathcal{S}(\mathcal{G}, \mathcal{I}) = \mathcal{S}_{\mathcal{I}^*}(\mathcal{G}^*) - \mathcal{S}_{\mathcal{I}^*}(\mathcal{G})$, we can apply exactly the same steps as in Theorem 1 to show that $\hat{\mathcal{G}} \in \mathcal{I}^*\text{-MEC}(\mathcal{G}^*)$.

We will end up with multiple conditions on $\lambda$ and $\lambda_R$. We now make sure they can all be satisfied simultaneously. Recall the three conditions we derived:

$$\lambda + \lambda_R < \min_{(\mathcal{G}, \mathcal{I}) \in \mathbb{S}} \frac{\eta(\mathcal{G}, \mathcal{I})}{-\min\{|\mathcal{G}| - |\mathcal{G}^*|, |\mathcal{I}| - |\mathcal{I}^*|\}} =: \alpha \tag{82}$$

$$\lambda < \min_{(\mathcal{G}, \mathcal{I}) \in \bar{\bar{\mathbb{S}}}} \frac{\eta(\mathcal{G}, \mathcal{I}) + \lambda_R(|\mathcal{I}| - |\mathcal{I}^*|)}{|\mathcal{G}^*| - |\mathcal{G}|} =: \beta(\lambda_R) \tag{83}$$

$$\lambda < \min_{\mathcal{G} \in \mathbb{G}^+} \frac{\eta(\mathcal{G}, \mathcal{I}^*)}{|\mathcal{G}^*| - |\mathcal{G}|} =: \gamma, \tag{84}$$

where the third condition comes from the steps of Theorem 1. We can simply pick $\lambda_R \in (0, \alpha)$ and $\lambda \in (0, \min\{\alpha - \lambda_R, \beta(\lambda_R), \gamma\})$. ∎

### A.4 Adapting the score to perfect interventions

The score developed in Section 3.1 is designed for general imperfect interventions. Since perfect interventions are just a special case of imperfect ones, this score will work on perfect interventions without problems. However, one can leverage the fact that the interventions are perfect to simplify the score a little bit.

$$\max_{\mathcal{G} \in \text{DAG}} \mathcal{S}_{\mathcal{I}^*}(\mathcal{G}) \tag{85}$$

$$= \max_{\mathcal{G} \in \text{DAG}} \sup_{\phi} \sum_{k=1}^{K} \mathbb{E}_{X \sim p^{(k)}} \log f^{(k)}(x; M^{\mathcal{G}}, R^{\mathcal{I}^*}, \phi) - \lambda|\mathcal{G}| \tag{86}$$

$$= \max_{\mathcal{G} \in \text{DAG}} \sup_{\phi^{(1)}} \left[ \sum_{k=1}^{K} \mathbb{E}_{X \sim p^{(k)}} \log \prod_{j \notin I_k^*} \tilde{f}(x_j; \text{NN}(M_j^{\mathcal{G}} \odot x; \phi_j^{(1)})) \right]$$

$$+ \sup_{\phi^{(2)}, \ldots, \phi^{(K)}} \left[ \sum_{k=2}^{K} \mathbb{E}_{X \sim p^{(k)}} \log \prod_{j \in I_k^*} \tilde{f}(x_j; \text{NN}(M_j^{\mathcal{G}} \odot x; \phi_j^{(k)})) \right] - \lambda|\mathcal{G}| \tag{87}$$

$$= \max_{\mathcal{G} \in \text{DAG}} \sup_{\phi^{(1)}} \left[ \sum_{k=1}^{K} \mathbb{E}_{X \sim p^{(k)}} \log \prod_{j \notin I_k^*} \tilde{f}(x_j; \text{NN}(M_j^{\mathcal{G}} \odot x; \phi_j^{(1)})) \right]$$

$$+ \sup_{\phi^{(2)}, \ldots, \phi^{(K)}} \left[ \sum_{k=2}^{K} \mathbb{E}_{X \sim p^{(k)}} \log \prod_{j \in I_k^*} \tilde{f}(x_j; \text{NN}(0 \odot x; \phi_j^{(k)})) \right] - \lambda|\mathcal{G}|, \tag{88}$$

where in (88) we use the fact that the interventions are perfect. In (88), the second $\sup$ does not depend on $\mathcal{G}$, so it can be ignored without changing the $\arg\max_{\mathcal{G}\in\text{DAG}}$.

Hence, for perfect intervention we use the score

$$\mathcal{S}_{\mathcal{I}^*}^{\text{perf}}(\mathcal{G}) := \sup_{\phi^{(1)}} \left[ \sum_{k=1}^{K} \mathbb{E}_{X\sim p^{(k)}} \log \prod_{j\notin I_k^*} \tilde{f}(x_j; \text{NN}(M_j^{\mathcal{G}} \odot x; \phi_j^{(1)})) \right] - \lambda|\mathcal{G}|. \qquad (89)$$

# B  Additional information

## B.1  Synthetic data sets

In this section, we describe how the different synthetic data sets were generated. For each type of data set, we first sample a DAG following the *Erdős-Rényi* scheme and then we sample the parameters of the different causal mechanisms as stated below (in the bulleted list). For 10-node graphs, single node interventions are performed on every node. For 20-node graphs, interventions target 1 to 2 nodes chosen uniformly at random. Then, $n/(d+1)$ examples are sampled for each interventional setting (if $n$ is not divisible by $d+1$, some intervention setting may have one extra sample in order to have a total of $n$ samples). The data are then normalized: we subtract the mean and divide by the standard deviation. For all data sets, the source nodes are Gaussian with zero mean and variance sampled from $\mathcal{U}[1,2]$. The noise variables $N_j$ are mutually independent and sampled from $\mathcal{N}(0,\sigma_j^2)$ $\forall j$, where $\sigma_j^2 \sim \mathcal{U}[1,2]$.

For perfect intervention, the distribution of intervened nodes is replaced by a marginal $\mathcal{N}(2,1)$. This type of intervention, that produce a mean-shift, is similar to those used in [15, 42]. For imperfect interventions, besides the initial parameters, an extra set of parameters were sampled by perturbing the initial parameters as described below. For nodes without parents, the distribution of intervened nodes is replaced by a marginal $\mathcal{N}(2,1)$. Both for the perfect and imperfect cases, we explore other types of interventions and report the results in Appendix C.5. We now describe the causal mechanisms and the nature of the imperfect intervention for the three different types of data set:

- The *linear* data sets are generated following $X_j := w_j^T X_{\pi_j^{\mathcal{G}}} + 0.4 \cdot N_j$ $\forall j$, where $w_j$ is a vector of $|\pi_j^{\mathcal{G}}|$ coefficients each sampled uniformly from $[-1,-0.25] \cup [0.25,1]$ (to make sure there are no $w$ close to 0). Imperfect interventions are obtained by adding a random vector of $\mathcal{U}([-5,-2] \cup [2,5])$ to $w_j$.

- The *additive noise model* (ANM) data sets are generated following $X_j := f_j(X_{\pi_j^{\mathcal{G}}}) + 0.4 \cdot N_j$ $\forall j$, where the functions $f_j$ are fully connected neural networks with one hidden layer of 10 units and *leaky ReLU* with a negative slope of 0.25 as nonlinearities. The weights of each neural network are randomly initialized from $\mathcal{N}(0,1)$. Imperfect interventions are obtained by adding a random vector of $\mathcal{N}(0,1)$ to the last layer.

- The *nonlinear with non-additive noise* (NN) data sets are generated following $X_j := f_j(X_{\pi_j^{\mathcal{G}}}, N_j)$ $\forall j$, where the functions $f_j$ are fully connected neural networks with one hidden layer of 20 units and *tanh* as nonlinearities. The weights of each neural network are randomly initialized from $\mathcal{N}(0,1)$. Similarly to the additive noise model, imperfect intervention are obtained by adding a random vector of $\mathcal{N}(0,1)$ to the last layer.

## B.2  Deep Sigmoidal Flow: Architectural details

A layer of a Deep Sigmoidal Flow is similar to a fully-connected network with one hidden layer, a single input, and a single output, but is defined slightly differently to ensure that the mapping is invertible and that the Jacobian is tractable. Each layer $l$ is defined as follows:

$$h^{(l)}(x) = \sigma^{-1}(w^T \sigma(a \cdot x + b)), \qquad (90)$$

where $0 < w_i < 1$, $\sum_i w_i = 1$ and $a_i > 0$. In our method, the neural networks $\text{NN}(\cdot; \phi_j^{(k)})$ output the parameters $(w_j, a_j, b_j)$ for each DSF $\tau_j$. To ensure that the determinant of the Jacobian is calculated in a numerically-stable way, we follow the recommendations of [18]. While other flows like the Deep Dense Sigmoidal Flow have more capacity, DSF was sufficient for our use.

## B.3 Optimization

In this section, we show how the augmented Lagrangian is applied, how the gradient is estimated and, finally, we illustrate the learning dynamics by analyzing an example.

Let us recall the score and the optimization problem from Section 3.2:

$$\hat{\mathcal{S}}_{\text{int}}(\Lambda) := \sup_\phi \underset{M \sim \sigma(\Lambda)}{\mathbb{E}} \left[ \sum_{k=1}^K \underset{X \sim p^{(k)}}{\mathbb{E}} \log f^{(k)}(X; M, \phi) - \lambda ||M||_0 \right], \tag{91}$$

$$\sup_\Lambda \hat{\mathcal{S}}_{\text{int}}(\Lambda) \quad \text{s.t.} \quad \operatorname{Tr} e^{\sigma(\Lambda)} - d = 0. \tag{92}$$

We optimize for $\phi$ and $\Lambda$ jointly, which yields the following optimization problem:

$$\sup_{\phi, \Lambda} \underset{M \sim \sigma(\Lambda)}{\mathbb{E}} \left[ \sum_{k=1}^K \underset{X \sim p^{(k)}}{\mathbb{E}} \log f^{(k)}(X; M, \phi) \right] - \lambda ||\sigma(\Lambda)||_1 \quad \text{s.t.} \quad \operatorname{Tr} e^{\sigma(\Lambda)} - d = 0, \tag{93}$$

where we used the fact that $\mathbb{E}_{M \sim \sigma(\Lambda)} ||M||_0 = ||\sigma(\Lambda)||_1$. Let us use the notation:

$$h(\Lambda) := \operatorname{Tr} e^{\sigma(\Lambda)} - d. \tag{94}$$

The augmented Lagrangian transforms the constrained problem into a sequence of unconstrained problems of the form

$$\sup_{\phi, \Lambda} \underset{M \sim \sigma(\Lambda)}{\mathbb{E}} \left[ \sum_{k=1}^K \underset{X \sim p^{(k)}}{\mathbb{E}} \log f^{(k)}(X; M, \phi) \right] - \lambda ||\sigma(\Lambda)||_1 - \gamma_t h(\Lambda) - \frac{\mu_t}{2} h(\Lambda)^2, \tag{95}$$

where $\gamma_t$ and $\mu_t$ are the Lagrangian multiplier and the penalty coefficient of the $t$th unconstrained problem, respectively. In all our experiments, we initialize $\gamma_0 = 0$ and $\mu_0 = 10^{-8}$. Each such problem is approximately solved using a stochastic gradient descent algorithm (RMSprop [44] in our experiments). We consider that a subproblem has converged when (95) evaluated on a held-out data set stops increasing. Let $(\phi_t^*, \Lambda_t^*)$ be the approximate solution to subproblem $t$. Then, $\gamma_t$ and $\mu_t$ are updated according to the following rule:

$$\begin{aligned} \gamma_{t+1} &\leftarrow \gamma_t + \mu_t \cdot h\left(\Lambda_t^*\right) \\ \mu_{t+1} &\leftarrow \begin{cases} \eta \cdot \mu_t, & \text{if } h\left(\Lambda_t^*\right) > \delta \cdot h\left(\Lambda_{t-1}^*\right) \\ \mu_t, & \text{otherwise} \end{cases} \end{aligned} \tag{96}$$

with $\eta = 2$ and $\delta = 0.9$. Each subproblem $t$ is initialized using the previous subproblem's solution $(\phi_{t-1}^*, \Lambda_{t-1}^*)$. The augmented Lagrangian method stops when $h(\Lambda) \leq 10^{-8}$ and the graph formed by adding an edge whenever $\sigma(\Lambda) > 0.5$ is acyclic.

**Gradient estimation.** The gradient of (95) w.r.t. $\phi$ and $\Lambda$ is estimated by

$$\nabla_{\phi, \Lambda} \left[ \frac{1}{|B|} \sum_{i \in B} \log f^{(k_i)}(x^{(i)}; M^{(i)}, \phi) - \lambda_t h(\Lambda) - \frac{\mu_t}{2} h(\Lambda)^2 \right], \tag{97}$$

where $B$ is an index set sampled without replacement, $x^{(i)}$ is an example from the training set and $k_i$ is the index of its corresponding intervention. To compute the gradient of the likelihood part w.r.t. $\Lambda$, we use the Straight-Through Gumbel-Softmax estimator, adapted to sigmoids [29, 20]. This approach was already used in the context of causal discovery without interventional data [32, 21]. The matrix $M^{(i)}$ is given by

$$M^{(i)} := \mathbb{I}(\sigma(\Lambda + L^{(i)}) > 0.5) + \sigma(\Lambda + L^{(i)}) - \text{grad-block}(\sigma(\Lambda + L^{(i)})), \tag{98}$$

where $L^{(i)}$ is a $d \times d$ matrix filled with independent Logistic samples, $\mathbb{I}$ is the indicator function applied element-wise and the function *grad-block* is such that grad-block$(z) = z$ and $\nabla_z$grad-block$(z) = 0$. This implies that each entry of $M^{(i)}$ evaluates to a discrete Bernoulli sample with probability given by $\sigma(\Lambda)$ while the gradient w.r.t. $\Lambda$ is computed using the soft Gumbel-Softmax sample. This yields a biased estimation of the actual gradient of objective (95), but its variance is low compared to the

popular unbiased REINFORCE estimator (a Monte Carlo estimator relying on the log-trick) [39, 29]. A temperature term can be added inside the sigmoid, but we found that a temperature of one gave good results.

In addition to this, we experimented with a different relaxation for the discrete variable $M$. We tried treating $M$ directly as a learnable parameter constrained in $[0, 1]$ via gradient projection. However, this approach yielded significantly worse results. We believe that the fact $M$ is continuous in this setting is problematic, since as an entry of $M$ gets closer and closer to zero, the weights of the first neural network layer can compensate, without affecting the likelihood whatsoever. This cannot happen when using the Straight-Through Gumbel-Softmax estimator because the neural network weights are only exposed to discrete $M$.

Figure 6: **Top:** Learning curves during training. *NLL* and *NLL on validation* are respectively the (pseudo) negative log-likelihood (NLL) on training and validation sets. *AL minus NLL* can be thought of as the acyclicity constraint violation plus the edge sparsity regularizer. *AL* and *AL on validation set* are the augmented Lagrangian objectives on training and validation set, respectively. **Middle and bottom:** Entries of the matrix $\sigma(\Lambda)$ w.r.t. to the number of iterations (green edges = edge present in the ground truth DAG, red edges = edge not present). The adjacency matrix to the left correspond to the ground truth DAG. The other matrices correspond to $\sigma(\Lambda)$ at 20 000, 30 000 and 62 000 iterations.

**Learning dynamics.** We present in Figure 6 the learning curves (top) and the matrix $\sigma(\Lambda)$ (middle and bottom) as DCDI-DSF is trained on a linear data set with perfect intervention sampled from a sparse 10-node graph (the same phenomenon was observed in a wide range of settings). In the graph at the top, we show the augmented Lagrangian and the (pseudo) negative log-likelihood (NLL) on train and validation set. To be exact, the NLL corresponds to a negative log-likelihood only once acyclicity is achieved. In the graph representing $\sigma(\Lambda)$ (middle), each curve represents a $\sigma(\alpha_{ij})$: green edges are edges present in the ground truth DAG and red edges are edges not present. The same information is presented in matrix form for a few specific iterations and can be easily compared to the adjacency matrix of the ground truth DAG (white = presence of an edge, blue = absence). Recall that when a $\sigma(\alpha_{ij})$ is equal (or close to) 0, it means that the entry $ij$ of the mask $M$ will also be 0. This is equivalent to say that the edge is not present in the learned DAG.

In this section, we review some important steps of the learning dynamics. At first, the NLL on the training and validation sets decrease sharply as the model fits the data. Around iteration 5000, the decrease slows down and the weights of the constraint (namely $\gamma$ and $\mu$) are increased. This puts pressure on the entries $\sigma(\alpha_{ij})$ to decrease. At iteration 20 000, many $\sigma(\alpha_{ij})$ that correspond to red edges have diminished close to 0, meaning that edges are correctly removed. It is noteworthy to mention that the matrix at this stage is close to being symmetric: the algorithm did not yet choose an orientation for the different edges. While this learned graph still has false-positive edges, the skeleton is reminiscent of a Markov Equivalence Class. As the training progresses, the weights of the constraint are greatly increased passed the 20 000th iteration leading to the removal of additional edges (leading also to an NLL increase). Around iteration 62 000 (the second vertical line), the stopping criterion is met: the acyclicity constraint is below the threshold (i.e. $h(\Lambda) \leq 10^{-8}$), the learned DAG is acyclic and the augmented Lagrangian on the validation set is not improving anymore. Edges with a $\sigma(\alpha_{ij})$ higher than 0.5 are set to 1 and others set to 0. The learned DAG has a SHD of 1 since it has a reversed edge compared to the ground truth DAG.

Finally, we illustrate the learning of interventional targets in the (perfect) unknown intervention setting by comparing an example of $\sigma(\beta_{kj})$, the learned targets, with the ground truth targets in Figure 7. Results are from DCDI-G on 10-node graph with higher connectivity. Each column corresponds to an interventional target $I_k$ and each row corresponds to a node. In the right matrix, a dark grey square in position $ij$ means that the node $i$ was intervened on in the interventional setting $I_j$. Each entry of the left matrix corresponds to the value of $\sigma(\beta_{kj})$. The binary matrix $R$ (from Equation 15) is sampled following these entries.

Figure 7: Learned targets $\sigma(\beta_{kj})$ compared to the ground truth targets.

## B.4 Baseline methods

In this section, we provide additional details on the baseline methods and cite the implementations that were used. GIES has been designed for the perfect interventions setting. It assumes linear relations with Gaussian noise and outputs an $\mathcal{I}$-Markov equivalence classes. In order to obtain the SHD and SID, we compare a DAG randomly sampled from the returned $\mathcal{I}$-Markov equivalence classes to the ground truth DAG. CAM has been modified to support perfect interventions. In particular, we used the loss that was already present in the code (similarly to the loss proposed for DCDI in the perfect intervention setting). Also, the preliminary neighbor search (PNS) and pruning processes were modified to not take into account data where variables are intervened on. Note that, while these two methods yield competitive results in the imperfect intervention setting, they were designed for perfect interventions: the targeted conditional are not fitted by an additional model (in contrast to our proposed score), they are simply removed from the score. Finally, JCI-PC is JCI used with the PC method [31]. The graph to learn is augmented with context variables (one per system variable in

our case). This modified version of PC can deal with unknown interventions. For the conditional independence test, we only used the gaussian CI test since using KCI-test was too slow for this algorithm.

For GIES, we used the implementation from the R package `pcalg`. For CAM, we modified the implementation from the R package `pcalg`. For IGSP and UT-IGSP, we used the implementation from `https://github.com/uhlerlab/causaldag`. The cutoff values used for `alpha-inv` was always the same as `alpha`. For JCI-PC, we modified the implementation from the R package `pcalg` using code from the JCI repository: `https://github.com/caus-am/jci/tree/master/jci`. The normalizing flows that we used for DCDI-DSF were adapted from the DSF implementation provided by its author [18]. We also used several tools from the Causal Discovery Toolbox (`https://github.com/FenTechSolutions/CausalDiscoveryToolbox`) [22] to interface R with Python and to compute the SHD and SID metrics.

## B.5 Default hyperparameters and hyperparameter search

For all score-based methods, we performed a hyperparameter search. The models were trained on $80\%$ examples and evaluated on the $20\%$ remaining examples. The hyperparameter combination chosen was the one that induced the lowest negative log-likelihood on the held-out examples. For DCDI, a grid search was performed over 10 values of the regularization coefficient (see Table 1) for known interventions (10 hyperparameter combinations in total) and, in the unknown intervention case, 3 values for the regularization coefficient of the learned targets $\lambda_R$ were also explored (30 hyperparameter combinations in total). For GIES and CAM, 50 hyperparameter combinations were considered using a random search following the sampling scheme of Table 1.

For IGSP, UT-IGSP and JCI-PC, we could not do a similar hyperparameter search since there is no score available to rank hyperparameter combinations. Thus, all examples were used to fit the model. Despite this, for IGSP and UT-IGSP, we explored a range of cutoff values around $10^{-5}$ (the value used for all the experiments in [42]): $\alpha = \{2e-1, 1e-1, 1e-2, 1e-3, 1e-5, 1e-7, 1e-9\}$. In the main text and figures, we report results with $\alpha = 1e-3$, which yielded low SHD and SID. For JCI-PC, we tested the following range of cutoff values: $\alpha = \{2e-1, 1e-1, 1e-2, 1e-3\}$ and report results with $\alpha = 1e-3$. Note that in a realistic setting, we do not have access to the ground truth graphs to choose a good cutoff value.

Table 1: Hyperparameter search spaces for each algorithm

|  | Hyperparameter space |
|---|---|
| DCDI | $\log_{10}(\lambda) \sim \mathcal{U}\{-7, -6, -5, -4, -3, -2, -1, 0, 1, 2\}$ <br> $\log_{10}(\lambda_R) \sim \mathcal{U}\{-4, -3, -2\}$ (only for unknown interventions) |
| CAM | $\log_{10}(\text{pruning cutoff}) \sim \mathcal{U}[-7, 0]$ |
| GIES | $\log_{10}(\text{regularizer coefficient}) \sim \mathcal{U}[-4, 4]$ |

Except for the normalizing flows of DCDI-DSF, DCDI-G and DCDI-DSF used exactly the same default hyperparameters that are summarized in Table 2. Some of these hyperparameters ($\mu_0, \gamma_0$), which are related to the optimization process are presented in Appendix B.3. These hyperparameters were used for almost all experiments, except for the real-world data set and the two-node graphs with complex densities, where overfitting was observed. Smaller architectures were tested until no major overfitting was observed. The default hyperparameters were chosen using small-scale experiments on perfect-known interventions data sets in order to have a small SHD. Since we observed that DCDI is not highly sensible to changes in hyperparameter values, only the regularization factors were part of a more thorough hyperparameter search. The neural networks were initialized following the Xavier initialization [14]. The neural network activation functions were leaky-ReLU. RMSprop was used as the optimizer [44] with minibatches of size 64.

Table 2: Default Hyperparameter for DCDI-G and DCDI-DSF

| DCDI hyperparameters |
|---|
| $\mu_0$: $10^{-8}$, $\gamma_0$: 0, $\eta$: 2, $\delta$: 0.9 |
| Augmented Lagrangian constraint threshold: $10^{-8}$ |
| learning rate: $10^{-3}$ |
| # hidden units: 16 |
| # hidden layers: 2 |
| # flow hidden units: 16 (only for DCDI-DSF) |
| # flow hidden layers: 2 (only for DCDI-DSF) |

# C    Additional experiments

## C.1    Real-world data set

We tested the methods that support perfect intervention on the flow cytometry data set of Sachs et al. [40]. The measurements are the level of expression of phosphoproteins and phospholipids in human cells. Interventions were performed by using reagents to activate or inhibit the measured proteins. As in Wang et al. [47], we use a subset of the data set, excluding experimental conditions where the perturbations were not directly done on a measured protein. This subset comprises 5 846 measurements: 1 755 measurements are considered observational, while the other 4 091 measurements are from five different single node interventions (with the following proteins as targets: Akt, PKC, PIP2, Mek, PIP3). The consensus graph from Sachs et al. [40] that we use as the ground truth DAG contains 11 nodes and 17 edges. While the flow cytometry data set is standard in the causal structure learning literature, some concerns have been raised. The "consensus" network proposed by [40] has been challenged by some experts [30]. Also, several assumptions of the different models may not be respected in this real-world data set (for more details, see [30]): i) the causal sufficiency assumption may not hold, ii) the interventions may not be as specific as stated, and iii) the ground truth network is possibly not a DAG since feedback loops are common in cellular signaling networks.

Table 3: Results for the flow cytometry data sets

| Method | SHD | SID | tp | fn | fp | rev | $F_1$ score |
|---|---|---|---|---|---|---|---|
| IGSP | 18 | 54 | 4 | 6 | 5 | 7 | 0.42 |
| GIES | 38 | 34 | 10 | 0 | 41 | 7 | 0.33 |
| CAM | 35 | 20 | 12 | 1 | 30 | 4 | 0.51 |
| DCDI-G | 36 | 43 | 6 | 2 | 25 | 9 | 0.31 |
| DCDI-DSF | 33 | 47 | 6 | 2 | 22 | 9 | 0.33 |

In Table 3 we report SHD and SID for all methods, along with the number of true positive (tp), false-negative (fn), false positive (fp), reversed (rev) edges, and the $F_1$ score. There are no measures of central tendencies, since there is only one graph. The modified version of CAM has overall the best performance: the highest $F_1$ score and a low SID. IGSP has a low SHD, but a high SID, which can be explained by the relatively high number of false negative. DCDI-G and DCDI-DSF have SHDs comparable to GIES and CAM, but higher than IGSP. In terms of SID, they outperform IGSP, but not GIES and CAM. Finally, the DCDI models have $F_1$ scores similar to that of GIES. Hence, we conclude that DCDI performs comparably to the state of the art on this data set, while none of the methods show great performance across the board.

**Hyperparameters.**    We report the hyperparameters used for Table 3. IGSP used the KCI-test with a cutoff value of $10^{-3}$. Hyperparameters for CAM and GIES were chosen following the hyperparameter search described in Appendix B.5. For DCDI, since overfitting was observed, we included some hyperparameters related to the architecture in the hyperparameter grid search (number of hidden units: $\{4, 8\}$, number of hidden layers: $\{1, 2\}$ and only for DSF, number of flow hidden units: $\{4, 8\}$, number of flow layers: $\{1, 2\}$), and used the scheme described in Appendix B.5 for choosing the regularization coefficient.

## C.2 Learning causal direction from complex distributions

To show that insufficient capacity can hinder learning the right causal direction, we used toy data sets with simple 2-node graphs under perfect and imperfect interventions. We show, in Figure 8 and 9, the joint densities respectively learned by DCDI-DSF and DCDI-G. We tested two different data sets: X and DNA, which corresponds to the left and right column, respectively. In both data sets, we experimented with perfect and imperfect interventions, on both the cause and the effect, i.e. $\mathcal{I} = (\emptyset, \{1\}, \{2\})$. In both figures, the top row corresponds to the learned densities when no intervention are performed. The bottom row corresponds to the learned densities under an imperfect intervention on the effect variable (changing the conditional).

Figure 8: Joint density learned by DCDI-DSF. White dots are data points and the color represents the learned density. The x-axis is cause and the y-axis is the effect. First row is observational while second row is with an imperfect intervention on the effect.

Figure 9: Joint density learned by DCDI-G. White dots are data points and the color represents the learned density. The x-axis is cause and the y-axis is the effect. First row is observational while second row is with an imperfect intervention on the effect.

For the X data set, both under perfect and imperfect interventions, the incapacity of DCDI-G to model this complex distribution properly makes it conclude (falsely) that there is no dependency between the two variables (the $\mu$ outputted by DCDI-G is constant). Conversely, for the DNA data set with perfect interventions, it does infer the dependencies between the two variables and learn the correct causal direction, although the distribution is modeled poorly. Notice that, for the DNA data set with imperfect interventions, the lack of capacity of DCDI-G has pushed it to learn the same density with and without interventions (compare the two densities in the second column of Figure 9; the learned density functions remain mostly unchanged from top to bottom). This prevented DCDI-G from learning the correct causal direction, while DCDI-DSF had no problem. We believe that if the

imperfect interventions were more radical, DCDI-G could have recovered the correct direction even though it lacks capacity. In all cases, DCDI-DSF can easily model these functions and systematically infers the right causal direction.

While the proposed data sets are synthetic, similar multimodal distributions could be observed in real-world data sets due to latent variables that are parent of only one node (i.e., that are not confounders). A hidden variable that act as a selector between two different mechanisms could induce distributions similar to those in Figures 8 and 9. In fact, this idea was used to produce the synthetic data sets, i.e., a latent variable $z \in \{0, 1\}$ was sampled and, according to its value, examples were generated following one of two mechanisms. The X dataset (second column in the figures) was generated by two linear mechanisms in the following way:

$$ y := \begin{cases} wx + N & z = 0 \\ -wx + N & z = 1, \end{cases} $$

where $N$ is a Gaussian noise and $w$ was randomly sampled from $[-1, -0.25] \cup [0.25, 1]$.

### C.3 Scalability experiments

Figure 10 presents two experiments which study the scalability of various methods in terms of number of examples (left) and number of variables (right). In these experiments, the runtime was restricted to 12 hours while the RAM memory was restricted to 16GB. All experiments considered perfect interventions. Experiments from Figure 10 were run with fixed hyperparameters. **DCDI.** Same as Table 2 except $\mu_0 = 10^{-2}$, # hidden units $= 8$ and $\lambda = 10^{-1}$. **CAM.** Pruning cutoff $= 10^{-3}$. Preliminary neighborhood selection was performed in the large graph experiments (otherwise CAM cannot run on 50 nodes in less than 12 hours). **GIES.** Regularizing parameter $= 1$. **IGSP.** The suffixes -G and -K refers to the partial correlation test and the KCI-test, respectively. The $\alpha$ parameter is set to $10^{-3}$.

**Number of examples.** DCDI was the only algorithm supporting nonlinear relationships that could run on as much as 1 million examples without running out of time or memory. We believe different trade-offs between SHD and SID could be achieved with different hyperparameters, especially for GIES and CAM which achieved very good SID but poor SHD.

**Number of variables.** We see that using a GPU starts to pay off for graphs of 50 nodes or more. For 10-50 nodes data sets, DCDI-GPU outperforms the other methods in terms of SHD and SID, while maintaining a runtime similar to CAM. For the hundred-node data sets, the runtime of DCDI increases significantly with a SHD/SID performance comparable to the much faster GIES. We believe the weaker performance of DCDI in the hundred-node setting is due to the fact that the conditionals are high dimensional functions which are prone to overfitting. Also, we believe this runtime could be significantly reduced by limiting the number of parents via preliminary neighborhood selection similar to CAM [4]. This would have the effect of reducing the cost of computing the gradient of w.r.t. to the neural network parameters. These adaptions to higher dimensionality are left as future work.

### C.4 Ablation study

In this section, by doing ablation studies, we show that i) that interventions are beneficial to our method to recover the DAG, ii) that the proposed losses yield better results than a standard loss ignoring information about interventions, and iii) that the use of high capacity model is relevant for nonlinear data sets.

**Effect of number of interventions.** In a small scale experiment, we show in Figure 11 the effect of the number of interventions on the performance of DCDI-G. The SHD and SID of DCDI-G and DCD are shown over ten linear data sets (20-node graph with sparse connectivity) with $\{0, 5, 10, 15, 20\}$ perfect interventions. The baseline DCD is equivalent to DCDI-G, but it uses a loss that doesn't take into account the interventions. It can first be noticed that, as the number of interventions increases, the performance of DCDI-G increases. This increase is particularly noticeable from the purely interventional data to data with 5 interventions. While DCD's performance also increases in terms of SHD, it seems to have no clear gain in terms of SID. Also, DCDI-G with interventional data is always better than DCD showing that the proposed loss for perfect interventions is pertinent. Note

Figure 10: We report the runtime (in hours), SHD and SID of multiple methods in multiple settings. The horizontal dashed lines at 12 hours represents the time limit imposed. When a curve reaches this dashed line, it means that the method could not finish within 12 hours. We write $\geq 16G$ when the RAM memory needed by the algorithm exceeded 16GB. All data sets have 10 interventional targets containing $0.1d$ targets. We considered perfect interventions. **Left:** Different data set sizes. Ten nodes ANM data with connectivity $e = 1$. **Right:** Different number of variables. NN data set with connectivity $e = 4$ and $10^4$ samples. Each curve is an average over 5 different datasets while the error bars are %95 confidence intervals computed via bootstrap.

that the first two boxes are the same since DCDI-G on observational data is equivalent to DCD (the experiment was done only once).

**Relevance of DCDI score to leverage interventional data.** In a larger scale experiment, with the same data sets used in the main text (Section 4), we compare DCDI-G and DCDI-DSF to DCD and DCD-no-interv for perfect/known, imperfect/known and perfect/unknown interventions (shown respectively in Appendix C.4.1, C.4.2, and C.4.3). The values reported are the mean and the standard deviation of SHD and SID over ten data sets of each condition. DCD-no-interv is DCDI-G applied to purely observational data. These purely observational data sets were generated from the same CGM as the other data set containing interventions and had the same total sample size. For SHD, the advantage of DCDI over DCD and DCD-no-interv is clear over all conditions. For SID, DCDI has no advantage for sparse graphs, but is usually better for graphs with higher connectivity. As in the first small scale experiment, the beneficial effect of interventions is clear. Also, these results show that the proposed losses for the different type of interventions are pertinent.

**Relevance of neural network models.** As a sanity check of our proposed method, we trained DCDI-G without hidden layers, i.e. a linear model. In Table 4, 5 and 6, we report the mean and standard deviation of SHD and SID over ten 20-node graphs for DCDI-linear and compare it to results obtained for DCDI-G and DCDI-DSF (both using hidden layers). As expected, this linear version of DCDI has competitive results for the linear data set, but poorer results on nonlinear data sets, showing the interest of using high capacity models.

Figure 11: SHD and SID for DCDI-G and DCD on data sets with a different number of interventional settings.

Table 4: Results for the linear data set with perfect intervention

| Method | 20 nodes, $e = 1$ | | 20 nodes, $e = 4$ | |
|---|---|---|---|---|
| | SHD | SID | SHD | SID |
| DCDI-linear | $5.9_{\pm 7.6}$ | $7.1_{\pm 6.9}$ | $16.0_{\pm 6.7}$ | $98.3_{\pm 31.4}$ |
| DCDI-G | $5.4_{\pm 4.5}$ | $13.4_{\pm 12.0}$ | $23.7_{\pm 5.6}$ | $112.8_{\pm 41.8}$ |
| DCDI-DSF | $3.6_{\pm 2.7}$ | $6.0_{\pm 5.4}$ | $16.6_{\pm 6.4}$ | $92.5_{\pm 40.1}$ |

Table 5: Results for the additive noise model data set with perfect intervention

| Method | 20 nodes, $e = 1$ | | 20 nodes, $e = 4$ | |
|---|---|---|---|---|
| | SHD | SID | SHD | SID |
| DCDI-linear | $29.6_{\pm 15.4}$ | $24.8_{\pm 18.4}$ | $66.2_{\pm 13.7}$ | $219.0_{\pm 41.7}$ |
| DCDI-G | $21.8_{\pm 30.1}$ | $11.6_{\pm 13.1}$ | $35.2_{\pm 13.2}$ | $109.8_{\pm 44.6}$ |
| DCDI-DSF | $4.3_{\pm 1.9}$ | $19.7_{\pm 12.6}$ | $26.7_{\pm 16.9}$ | $105.3_{\pm 22.7}$ |

Table 6: Results for the nonlinear with non-additive noise data set with perfect intervention

| Method | 20 nodes, $e = 1$ | | 20 nodes, $e = 4$ | |
|---|---|---|---|---|
| | SHD | SID | SHD | SID |
| DCDI-linear | $19.8_{\pm 12.7}$ | $14.2_{\pm 9.2}$ | $45.6_{\pm 12.0}$ | $177.9_{\pm 27.6}$ |
| DCDI-G | $13.9_{\pm 20.3}$ | $13.7_{\pm 8.1}$ | $16.8_{\pm 8.7}$ | $82.5_{\pm 38.1}$ |
| DCDI-DSF | $8.3_{\pm 4.1}$ | $32.4_{\pm 17.3}$ | $11.8_{\pm 2.1}$ | $102.3_{\pm 34.5}$ |

### C.4.1 Perfect interventions

Table 7: Results for the linear data set with perfect intervention

| Method | 10 nodes, $e = 1$ | | 10 nodes, $e = 4$ | | 20 nodes, $e = 1$ | | 20 nodes, $e = 4$ | |
|---|---|---|---|---|---|---|---|---|
| | SHD | SID | SHD | SID | SHD | SID | SHD | SID |
| DCD | $6.6_{\pm 3.6}$ | $14.1_{\pm 11.5}$ | $24.4_{\pm 6.0}$ | $67.0_{\pm 9.2}$ | $18.2_{\pm 15.8}$ | $30.9_{\pm 21.7}$ | $56.7_{\pm 10.2}$ | $227.0_{\pm 38.6}$ |
| DCD-no-interv | $8.9_{\pm 2.8}$ | $19.5_{\pm 10.9}$ | $26.7_{\pm 5.9}$ | $69.0_{\pm 11.2}$ | $24.6_{\pm 20.5}$ | $31.2_{\pm 22.8}$ | $64.4_{\pm 11.4}$ | $292.9_{\pm 28.9}$ |
| DCDI-G | $1.3_{\pm 1.9}$ | $0.8_{\pm 1.8}$ | $3.3_{\pm 2.1}$ | $10.7_{\pm 12.0}$ | $5.4_{\pm 4.5}$ | $13.4_{\pm 12.0}$ | $23.7_{\pm 5.6}$ | $112.8_{\pm 41.8}$ |
| DCDI-DSF | $0.9_{\pm 1.3}$ | $0.6_{\pm 1.9}$ | $3.7_{\pm 2.3}$ | $18.9_{\pm 14.1}$ | $3.6_{\pm 2.7}$ | $6.0_{\pm 5.4}$ | $16.6_{\pm 6.4}$ | $92.5_{\pm 40.1}$ |

Table 8: Results for the additive noise model data set with perfect intervention

| Method | 10 nodes, $e = 1$ | | 10 nodes, $e = 4$ | | 20 nodes, $e = 1$ | | 20 nodes, $e = 4$ | |
|---|---|---|---|---|---|---|---|---|
| | SHD | SID | SHD | SID | SHD | SID | SHD | SID |
| DCD | $11.5_{\pm 6.6}$ | $18.2_{\pm 11.8}$ | $30.4_{\pm 3.8}$ | $75.5_{\pm 4.6}$ | $39.3_{\pm 28.4}$ | $39.8_{\pm 33.3}$ | $62.7_{\pm 14.2}$ | $241.0_{\pm 44.8}$ |
| DCD-no-interv | $11.6_{\pm 8.8}$ | $15.8_{\pm 12.1}$ | $21.3_{\pm 5.2}$ | $63.5_{\pm 12.3}$ | $41.7_{\pm 44.1}$ | $36.2_{\pm 27.1}$ | $43.7_{\pm 9.2}$ | $226.1_{\pm 42.8}$ |
| DCDI-G | $5.2_{\pm 7.5}$ | $2.4_{\pm 4.9}$ | $4.3_{\pm 2.4}$ | $16.0_{\pm 11.9}$ | $21.8_{\pm 30.1}$ | $11.6_{\pm 13.1}$ | $35.2_{\pm 13.2}$ | $109.8_{\pm 44.6}$ |
| DCDI-DSF | $4.2_{\pm 5.6}$ | $5.6_{\pm 5.5}$ | $5.5_{\pm 2.4}$ | $23.9_{\pm 14.3}$ | $4.3_{\pm 1.9}$ | $19.7_{\pm 12.6}$ | $26.7_{\pm 16.9}$ | $105.3_{\pm 22.7}$ |

Table 9: Results for the nonlinear with non-additive noise data set with perfect intervention

| Method | 10 nodes, $e = 1$ | | 10 nodes, $e = 4$ | | 20 nodes, $e = 1$ | | 20 nodes, $e = 4$ | |
|---|---|---|---|---|---|---|---|---|
| | SHD | SID | SHD | SID | SHD | SID | SHD | SID |
| DCD | $5.9_{\pm 6.9}$ | $10.9_{\pm 10.4}$ | $15.7_{\pm 4.9}$ | $53.0_{\pm 9.9}$ | $28.7_{\pm 13.0}$ | $29.7_{\pm 9.3}$ | $29.3_{\pm 8.9}$ | $163.1_{\pm 48.4}$ |
| DCD-no-interv | $11.0_{\pm 9.3}$ | $9.9_{\pm 11.0}$ | $18.4_{\pm 6.4}$ | $56.4_{\pm 11.0}$ | $16.5_{\pm 22.8}$ | $31.9_{\pm 17.5}$ | $31.6_{\pm 11.3}$ | $160.3_{\pm 46.3}$ |
| DCDI-G | $2.3_{\pm 3.6}$ | $2.7_{\pm 3.3}$ | $2.4_{\pm 1.6}$ | $13.9_{\pm 8.5}$ | $13.9_{\pm 20.3}$ | $13.7_{\pm 8.1}$ | $16.8_{\pm 8.7}$ | $82.5_{\pm 38.1}$ |
| DCDI-DSF | $7.0_{\pm 10.7}$ | $7.8_{\pm 5.8}$ | $1.6_{\pm 1.6}$ | $7.7_{\pm 13.8}$ | $8.3_{\pm 4.1}$ | $32.4_{\pm 17.3}$ | $11.8_{\pm 2.1}$ | $102.3_{\pm 34.5}$ |

### C.4.2 Imperfect interventions

Table 10: Results for the linear data set with imperfect intervention

| Method | 10 nodes, $e = 1$ | | 10 nodes, $e = 4$ | | 20 nodes, $e = 1$ | | 20 nodes, $e = 4$ | |
|---|---|---|---|---|---|---|---|---|
| | SHD | SID | SHD | SID | SHD | SID | SHD | SID |
| DCD | $10.6_{\pm 5.4}$ | $24.6_{\pm 18.2}$ | $24.0_{\pm 4.1}$ | $67.2_{\pm 7.6}$ | $21.2_{\pm 11.5}$ | $56.0_{\pm 31.5}$ | $56.7_{\pm 9.0}$ | $268.0_{\pm 25.4}$ |
| DCD-no-interv | $6.8_{\pm 4.4}$ | $19.5_{\pm 13.2}$ | $27.4_{\pm 4.4}$ | $74.0_{\pm 7.2}$ | $19.8_{\pm 9.2}$ | $48.2_{\pm 30.6}$ | $58.2_{\pm 9.9}$ | $288.6_{\pm 31.6}$ |
| DCDI-G | $2.7_{\pm 2.8}$ | $8.2_{\pm 8.8}$ | $5.2_{\pm 3.5}$ | $25.1_{\pm 12.9}$ | $15.6_{\pm 14.5}$ | $29.1_{\pm 23.4}$ | $34.0_{\pm 7.7}$ | $180.9_{\pm 44.5}$ |
| DCDI-DSF | $1.3_{\pm 1.3}$ | $4.2_{\pm 4.0}$ | $1.7_{\pm 2.4}$ | $10.2_{\pm 14.9}$ | $6.9_{\pm 6.3}$ | $22.7_{\pm 21.9}$ | $21.7_{\pm 8.1}$ | $137.4_{\pm 34.3}$ |

Table 11: Results for the additive noise model data set with imperfect intervention

| Method | 10 nodes, $e = 1$ | | 10 nodes, $e = 4$ | | 20 nodes, $e = 1$ | | 20 nodes, $e = 4$ | |
|---|---|---|---|---|---|---|---|---|
| | SHD | SID | SHD | SID | SHD | SID | SHD | SID |
| DCD | $12.0_{\pm 9.2}$ | $14.8_{\pm 10.4}$ | $24.3_{\pm 3.8}$ | $64.5_{\pm 11.1}$ | $51.7_{\pm 41.7}$ | $44.5_{\pm 20.0}$ | $54.1_{\pm 12.0}$ | $196.6_{\pm 37.2}$ |
| DCD-no-interv | $14.6_{\pm 4.3}$ | $12.1_{\pm 11.8}$ | $24.8_{\pm 4.8}$ | $69.3_{\pm 8.3}$ | $49.5_{\pm 36.0}$ | $32.7_{\pm 22.7}$ | $41.2_{\pm 8.1}$ | $197.7_{\pm 50.1}$ |
| DCDI-G | $6.2_{\pm 5.4}$ | $7.6_{\pm 11.0}$ | $13.1_{\pm 2.9}$ | $48.1_{\pm 9.1}$ | $30.5_{\pm 33.0}$ | $12.5_{\pm 8.8}$ | $43.1_{\pm 10.2}$ | $96.6_{\pm 47.1}$ |
| DCDI-DSF | $13.4_{\pm 8.4}$ | $17.9_{\pm 10.5}$ | $14.4_{\pm 2.4}$ | $53.2_{\pm 8.2}$ | $13.1_{\pm 4.5}$ | $43.5_{\pm 19.2}$ | $50.5_{\pm 11.4}$ | $172.1_{\pm 19.6}$ |

Table 12: Results for the nonlinear with non-additive noise data set with imperfect intervention

| Method | 10 nodes, $e = 1$ | | 10 nodes, $e = 4$ | | 20 nodes, $e = 1$ | | 20 nodes, $e = 4$ | |
|---|---|---|---|---|---|---|---|---|
| | SHD | SID | SHD | SID | SHD | SID | SHD | SID |
| DCD | $12.7_{\pm 8.4}$ | $11.8_{\pm 7.3}$ | $15.2_{\pm 3.7}$ | $52.2_{\pm 9.1}$ | $40.4_{\pm 54.7}$ | $45.2_{\pm 43.9}$ | $30.5_{\pm 8.0}$ | $151.2_{\pm 41.7}$ |
| DCD-no-interv | $13.6_{\pm 9.7}$ | $13.0_{\pm 8.1}$ | $14.8_{\pm 3.5}$ | $51.7_{\pm 12.5}$ | $37.1_{\pm 40.7}$ | $57.1_{\pm 56.2}$ | $31.3_{\pm 5.5}$ | $162.3_{\pm 40.5}$ |
| DCDI-G | $3.9_{\pm 3.9}$ | $7.5_{\pm 6.5}$ | $7.3_{\pm 2.2}$ | $28.0_{\pm 10.5}$ | $18.2_{\pm 28.8}$ | $36.9_{\pm 37.0}$ | $21.7_{\pm 8.0}$ | $127.3_{\pm 40.1}$ |
| DCDI-DSF | $5.3_{\pm 4.2}$ | $16.3_{\pm 10.0}$ | $5.9_{\pm 3.2}$ | $35.1_{\pm 12.3}$ | $13.2_{\pm 5.1}$ | $76.5_{\pm 57.8}$ | $16.8_{\pm 5.3}$ | $143.6_{\pm 48.8}$ |

### C.4.3 Unknown interventions

Table 13: Results for the linear data set with perfect intervention with unknown targets

| Method | 10 nodes, $e = 1$ | | 10 nodes, $e = 4$ | | 20 nodes, $e = 1$ | | 20 nodes, $e = 4$ | |
|---|---|---|---|---|---|---|---|---|
| | SHD | SID | SHD | SID | SHD | SID | SHD | SID |
| DCD | $6.6_{\pm 3.6}$ | $14.1_{\pm 11.5}$ | $24.4_{\pm 6.0}$ | $67.0_{\pm 9.2}$ | $18.2_{\pm 15.8}$ | $30.9_{\pm 21.7}$ | $56.7_{\pm 10.2}$ | $227.0_{\pm 38.6}$ |
| DCD-no-interv | $8.9_{\pm 2.8}$ | $19.5_{\pm 10.9}$ | $26.7_{\pm 5.9}$ | $69.0_{\pm 11.2}$ | $24.6_{\pm 20.5}$ | $31.2_{\pm 22.8}$ | $64.4_{\pm 11.4}$ | $292.9_{\pm 28.9}$ |
| DCDI-G | $5.3_{\pm 3.7}$ | $12.9_{\pm 11.5}$ | $5.2_{\pm 3.0}$ | $24.3_{\pm 15.3}$ | $15.4_{\pm 10.3}$ | $30.8_{\pm 18.6}$ | $39.2_{\pm 8.7}$ | $173.7_{\pm 45.6}$ |
| DCDI-DSF | $3.9_{\pm 4.3}$ | $7.1_{\pm 7.1}$ | $7.1_{\pm 3.6}$ | $35.8_{\pm 12.5}$ | $4.3_{\pm 2.4}$ | $18.4_{\pm 7.3}$ | $29.7_{\pm 12.6}$ | $147.8_{\pm 42.7}$ |

Table 14: Results for the additive noise model data set with perfect intervention with unknown targets

| Method | 10 nodes, $e = 1$ | | 10 nodes, $e = 4$ | | 20 nodes, $e = 1$ | | 20 nodes, $e = 4$ | |
|---|---|---|---|---|---|---|---|---|
| | SHD | SID | SHD | SID | SHD | SID | SHD | SID |
| DCD | $11.5_{\pm 6.6}$ | $18.2_{\pm 11.8}$ | $30.4_{\pm 3.8}$ | $75.5_{\pm 4.6}$ | $39.3_{\pm 28.4}$ | $39.8_{\pm 33.3}$ | $62.7_{\pm 14.2}$ | $241.0_{\pm 44.8}$ |
| DCD-no-interv | $11.6_{\pm 8.8}$ | $15.8_{\pm 12.1}$ | $21.3_{\pm 5.2}$ | $63.5_{\pm 12.3}$ | $41.7_{\pm 44.1}$ | $36.2_{\pm 27.1}$ | $43.7_{\pm 9.2}$ | $226.1_{\pm 42.8}$ |
| DCDI-G | $7.6_{\pm 10.3}$ | $5.0_{\pm 5.4}$ | $9.1_{\pm 3.8}$ | $37.5_{\pm 14.1}$ | $41.3_{\pm 39.2}$ | $22.9_{\pm 15.5}$ | $39.9_{\pm 18.8}$ | $153.7_{\pm 50.3}$ |
| DCDI-DSF | $11.9_{\pm 8.8}$ | $13.8_{\pm 7.9}$ | $6.6_{\pm 2.6}$ | $32.6_{\pm 14.1}$ | $22.3_{\pm 31.9}$ | $33.1_{\pm 17.5}$ | $42.5_{\pm 18.7}$ | $152.9_{\pm 53.4}$ |

Table 15: Results for the nonlinear with non-additive noise data set with perfect intervention with unknown targets

| Method | 10 nodes, $e = 1$ | | 10 nodes, $e = 4$ | | 20 nodes, $e = 1$ | | 20 nodes, $e = 4$ | |
|---|---|---|---|---|---|---|---|---|
| | SHD | SID | SHD | SID | SHD | SID | SHD | SID |
| DCD | $5.9_{\pm 6.9}$ | $10.9_{\pm 10.4}$ | $15.7_{\pm 4.9}$ | $53.0_{\pm 9.9}$ | $28.7_{\pm 13.0}$ | $29.7_{\pm 9.3}$ | $29.3_{\pm 8.9}$ | $163.1_{\pm 48.4}$ |
| DCD-no-interv | $11.0_{\pm 9.3}$ | $9.9_{\pm 11.0}$ | $18.4_{\pm 6.4}$ | $56.4_{\pm 11.0}$ | $16.5_{\pm 22.8}$ | $31.9_{\pm 17.5}$ | $31.6_{\pm 11.3}$ | $160.3_{\pm 46.3}$ |
| DCDI-G | $3.4_{\pm 4.2}$ | $6.9_{\pm 7.5}$ | $3.3_{\pm 1.3}$ | $20.4_{\pm 10.4}$ | $21.8_{\pm 32.1}$ | $20.9_{\pm 12.3}$ | $20.1_{\pm 8.1}$ | $104.6_{\pm 47.1}$ |
| DCDI-DSF | $7.8_{\pm 7.9}$ | $11.8_{\pm 5.7}$ | $3.3_{\pm 1.2}$ | $23.2_{\pm 9.1}$ | $27.4_{\pm 30.9}$ | $49.3_{\pm 15.7}$ | $22.2_{\pm 10.4}$ | $131.0_{\pm 41.0}$ |

## C.5 Different kinds of interventions

In this section, we compare DCDI to IGSP using data sets under different kinds of interventions. We report results in tabular form for 10-node and 20-node graphs. For the perfect interventions, instead of replacing the target conditional distribution by the marginal $\mathcal{N}(2, 1)$ (as in the main results), we used a marginal that doesn't involve a mean-shift: $\mathcal{U}[-1, 1]$. The results reported in Tables 16, 17, 18 of Section C.5.1 are the mean and the standard deviation of SHD and SID over ten data sets of each condition. From these results, we can conclude that DCDI-G still outperforms IGSP and, by comparing to DCD (DCDI-G with a loss that doesn't take into account interventions), that the proposed loss is still beneficial for this kind of interventions. It has competitive results compared to GIES and CAM on the linear data set and it outperforms them on the other data sets.

For imperfect intervention, we tried more modest changes in the parameters. For the linear data set, an imperfect intervention consisted of adding $\mathcal{U}[0.5, 1]$ to $w_j$ if $w_j > 0$ and subtracting if $w_j <= 0$. It was done this way to ensure that the intervention would not remove dependencies between variables. For the additive noise model and the nonlinear with non-additive noise data sets, $\mathcal{N}(0, 0.1)$ was added to each weight of the neural networks. Results are reported in Tables 19, 20, 21 of Section C.5.2. These smaller changes made the difference between DCD and DCDI imperceptible. For sparse

graphs, IGSP has a better or comparable performance to DCDI. For graphs with higher connectivity, DCDI often has a better performance than IGSP.

### C.5.1 Perfect interventions

Table 16: Results for the linear data set with perfect intervention

| Method | 10 nodes, $e=1$ | | 10 nodes, $e=4$ | | 20 nodes, $e=1$ | | 20 nodes, $e=4$ | |
|---|---|---|---|---|---|---|---|---|
| | SHD | SID | SHD | SID | SHD | SID | SHD | SID |
| IGSP | $4.0_{\pm4.8}$ | $15.7_{\pm15.4}$ | $28.8_{\pm2.0}$ | $72.2_{\pm5.1}$ | $9.7_{\pm8.7}$ | $45.1_{\pm45.4}$ | $68.1_{\pm13.6}$ | $295.4_{\pm27.6}$ |
| GIES | $0.3_{\pm0.5}$ | $0.0_{\pm0.0}$ | $4.0_{\pm6.5}$ | $6.7_{\pm17.7}$ | $1.5_{\pm1.2}$ | $0.3_{\pm0.9}$ | $49.4_{\pm22.2}$ | $111.9_{\pm51.4}$ |
| CAM | $0.6_{\pm1.0}$ | $0.0_{\pm0.0}$ | $11.8_{\pm4.3}$ | $32.2_{\pm17.2}$ | $6.3_{\pm7.4}$ | $7.6_{\pm9.8}$ | $91.4_{\pm21.3}$ | $181.7_{\pm60.5}$ |
| DCD | $6.3_{\pm3.4}$ | $14.8_{\pm10.6}$ | $26.1_{\pm3.3}$ | $66.4_{\pm11.4}$ | $11.1_{\pm4.7}$ | $45.8_{\pm22.8}$ | $49.0_{\pm12.0}$ | $258.6_{\pm41.6}$ |
| DCDI-G | $0.4_{\pm0.7}$ | $1.3_{\pm2.1}$ | $7.5_{\pm1.4}$ | $29.7_{\pm8.2}$ | $3.2_{\pm3.2}$ | $12.1_{\pm11.2}$ | $21.0_{\pm4.9}$ | $147.6_{\pm49.5}$ |

Table 17: Results for the additive noise model data set with perfect intervention

| Method | 10 nodes, $e=1$ | | 10 nodes, $e=4$ | | 20 nodes, $e=1$ | | 20 nodes, $e=4$ | |
|---|---|---|---|---|---|---|---|---|
| | SHD | SID | SHD | SID | SHD | SID | SHD | SID |
| IGSP | $5.7_{\pm2.3}$ | $23.4_{\pm13.6}$ | $32.8_{\pm2.4}$ | $79.3_{\pm3.2}$ | $14.9_{\pm8.1}$ | $78.8_{\pm64.6}$ | $80.5_{\pm6.4}$ | $337.6_{\pm27.3}$ |
| GIES | $7.5_{\pm5.1}$ | $2.3_{\pm2.5}$ | $9.2_{\pm2.9}$ | $27.1_{\pm11.5}$ | $23.8_{\pm18.4}$ | $3.1_{\pm4.4}$ | $89.6_{\pm14.7}$ | $143.9_{\pm53.1}$ |
| CAM | $6.3_{\pm6.9}$ | $0.0_{\pm0.0}$ | $6.3_{\pm3.8}$ | $14.6_{\pm20.1}$ | $9.2_{\pm14.3}$ | $13.5_{\pm25.1}$ | $106.2_{\pm14.6}$ | $96.2_{\pm57.9}$ |
| DCD | $6.4_{\pm4.6}$ | $22.0_{\pm14.7}$ | $31.1_{\pm3.4}$ | $77.4_{\pm3.1}$ | $18.1_{\pm8.0}$ | $51.5_{\pm41.5}$ | $55.7_{\pm8.3}$ | $261.3_{\pm22.5}$ |
| DCDI-G | $0.9_{\pm1.2}$ | $3.9_{\pm6.4}$ | $5.2_{\pm1.9}$ | $24.0_{\pm9.3}$ | $6.5_{\pm5.6}$ | $17.9_{\pm19.1}$ | $26.8_{\pm7.0}$ | $94.4_{\pm41.5}$ |

Table 18: Results for the nonlinear with non-additive noise data set with perfect intervention

| Method | 10 nodes, $e=1$ | | 10 nodes, $e=4$ | | 20 nodes, $e=1$ | | 20 nodes, $e=4$ | |
|---|---|---|---|---|---|---|---|---|
| | SHD | SID | SHD | SID | SHD | SID | SHD | SID |
| IGSP | $6.6_{\pm3.9}$ | $25.8_{\pm17.9}$ | $31.1_{\pm3.3}$ | $77.1_{\pm5.7}$ | $14.4_{\pm4.8}$ | $63.8_{\pm26.5}$ | $79.7_{\pm8.1}$ | $341.4_{\pm18.1}$ |
| GIES | $6.2_{\pm3.5}$ | $0.9_{\pm1.5}$ | $9.5_{\pm3.6}$ | $29.0_{\pm17.7}$ | $12.2_{\pm2.1}$ | $3.4_{\pm3.2}$ | $63.8_{\pm11.1}$ | $124.9_{\pm36.9}$ |
| CAM | $4.1_{\pm3.8}$ | $2.3_{\pm3.4}$ | $11.3_{\pm4.2}$ | $35.4_{\pm20.8}$ | $4.2_{\pm2.3}$ | $10.9_{\pm10.3}$ | $106.6_{\pm15.7}$ | $144.2_{\pm51.8}$ |
| DCD | $6.6_{\pm3.5}$ | $18.1_{\pm8.1}$ | $20.6_{\pm3.9}$ | $65.8_{\pm9.9}$ | $9.4_{\pm4.9}$ | $25.6_{\pm16.2}$ | $28.6_{\pm6.8}$ | $188.0_{\pm28.7}$ |
| DCDI-G | $2.1_{\pm1.5}$ | $4.6_{\pm5.4}$ | $5.0_{\pm4.3}$ | $28.8_{\pm17.6}$ | $6.4_{\pm3.8}$ | $15.1_{\pm8.0}$ | $12.2_{\pm2.7}$ | $96.1_{\pm18.9}$ |

### C.5.2 Imperfect interventions

Table 19: Results for the linear data set with imperfect intervention

| Method | 10 nodes, $e=1$ | | 10 nodes, $e=4$ | | 20 nodes, $e=1$ | | 20 nodes, $e=4$ | |
|---|---|---|---|---|---|---|---|---|
| | SHD | SID | SHD | SID | SHD | SID | SHD | SID |
| IGSP | $1.1_{\pm1.1}$ | $5.4_{\pm5.4}$ | $28.7_{\pm3.2}$ | $72.4_{\pm6.7}$ | $4.2_{\pm3.9}$ | $17.7_{\pm12.3}$ | $86.1_{\pm12.3}$ | $289.8_{\pm26.3}$ |
| DCD | $3.8_{\pm3.6}$ | $9.4_{\pm6.4}$ | $27.7_{\pm3.4}$ | $74.6_{\pm3.5}$ | $27.2_{\pm22.3}$ | $39.3_{\pm20.5}$ | $65.0_{\pm8.0}$ | $306.8_{\pm26.3}$ |
| DCDI-G | $4.7_{\pm4.5}$ | $11.5_{\pm9.5}$ | $27.4_{\pm4.9}$ | $73.8_{\pm5.4}$ | $29.6_{\pm16.5}$ | $37.7_{\pm14.5}$ | $62.8_{\pm6.5}$ | $303.2_{\pm27.6}$ |
| DCDI-DSF | $4.1_{\pm2.3}$ | $10.3_{\pm7.5}$ | $24.3_{\pm5.3}$ | $69.1_{\pm8.7}$ | $12.2_{\pm2.9}$ | $42.6_{\pm18.3}$ | $56.1_{\pm9.2}$ | $291.4_{\pm35.7}$ |

Table 20: Results for the additive noise model data set with imperfect intervention

| Method | 10 nodes, $e=1$ | | 10 nodes, $e=4$ | | 20 nodes, $e=1$ | | 20 nodes, $e=4$ | |
|---|---|---|---|---|---|---|---|---|
| | SHD | SID | SHD | SID | SHD | SID | SHD | SID |
| IGSP | $5.7_{\pm4.0}$ | $17.4_{\pm13.4}$ | $30.3_{\pm4.0}$ | $73.9_{\pm11.3}$ | $12.5_{\pm6.6}$ | $44.9_{\pm26.7}$ | $85.8_{\pm4.4}$ | $344.0_{\pm9.8}$ |
| DCD | $12.0_{\pm10.3}$ | $11.3_{\pm8.4}$ | $23.5_{\pm2.1}$ | $69.7_{\pm2.5}$ | $39.5_{\pm42.3}$ | $28.2_{\pm13.9}$ | $50.9_{\pm7.1}$ | $247.8_{\pm36.6}$ |
| DCDI-G | $12.7_{\pm9.1}$ | $11.8_{\pm6.5}$ | $21.7_{\pm4.3}$ | $65.2_{\pm9.2}$ | $16.2_{\pm18.0}$ | $27.8_{\pm13.1}$ | $46.2_{\pm5.9}$ | $240.1_{\pm26.3}$ |
| DCDI-DSF | $8.1_{\pm8.2}$ | $15.8_{\pm9.3}$ | $23.3_{\pm6.3}$ | $68.7_{\pm8.2}$ | $12.3_{\pm4.1}$ | $39.9_{\pm19.5}$ | $51.0_{\pm7.1}$ | $257.7_{\pm31.6}$ |

Table 21: Results for the nonlinear with non-additive noise data set with imperfect intervention

| Method | 10 nodes, $e = 1$ | | 10 nodes, $e = 4$ | | 20 nodes, $e = 1$ | | 20 nodes, $e = 4$ | |
|---|---|---|---|---|---|---|---|---|
| | SHD | SID | SHD | SID | SHD | SID | SHD | SID |
| IGSP | $7.0_{\pm 5.7}$ | $22.7_{\pm 19.5}$ | $29.4_{\pm 5.0}$ | $74.2_{\pm 7.3}$ | $18.7_{\pm 7.1}$ | $86.3_{\pm 37.1}$ | $81.6_{\pm 6.9}$ | $344.4_{\pm 20.5}$ |
| DCD | $9.4_{\pm 8.9}$ | $13.3_{\pm 11.0}$ | $15.1_{\pm 3.7}$ | $54.2_{\pm 9.8}$ | $28.5_{\pm 25.0}$ | $25.5_{\pm 16.8}$ | $32.7_{\pm 9.8}$ | $177.1_{\pm 37.5}$ |
| DCDI-G | $6.7_{\pm 5.1}$ | $13.0_{\pm 9.7}$ | $14.6_{\pm 3.3}$ | $53.9_{\pm 9.1}$ | $28.9_{\pm 33.7}$ | $25.2_{\pm 15.2}$ | $32.3_{\pm 7.9}$ | $177.0_{\pm 55.8}$ |
| DCDI-DSF | $12.8_{\pm 9.6}$ | $22.9_{\pm 14.8}$ | $14.4_{\pm 4.8}$ | $54.2_{\pm 10.3}$ | $13.3_{\pm 5.3}$ | $54.2_{\pm 20.9}$ | $28.6_{\pm 8.9}$ | $199.5_{\pm 32.7}$ |

## C.6 Evaluation on unseen interventional distributions

As advocated by Gentzel et al. [13], we present *interventional performance measures* for the flow cytometry data set of Sachs et al. [40] and for the nonlinear with non-additive noise data set. Interventional performance refers to the ability of the causal graph to model the effect of *unseen* interventions. To evaluate this, methods are trained on all the data, except for data coming from one interventional setting. Then, we evaluate the likelihood of the fitted model on the remaining *unseen* interventional distribution. Since some algorithms do not model distributions, for each method, given its estimated causal graph, we fit a distribution using a normalizing flow model, enabling a fair comparison. We report the log-likelihood evaluated on an unseen intervention. Note that when evaluating the likelihood, we ignore the conditional of the targeted node.

For the nonlinear data sets with non-additive noise, we report in Figure 12 boxplots over 10 dense graphs ($e = 4$) of 10 nodes. For each graph, one interventional setting was chosen randomly as the unseen intervention. DCDI-G and DCDI-DSF have the best performance, as was the case for the SHD and SID.

Figure 12: Log-likelihood on unseen interventional distributions of the nonlinear with non-additive noise data sets.

For Sachs, the data where intervention were applied on the protein *Akt* were used as the "held-out" distribution. We report in Figure 13 the log-likelihood and its standard deviation over these data samples. The ordering of the methods is different from the structural metrics: IGSP has the best performance followed by DCDI-G (whereas CAM seemed to have the best performance with the structural metrics).

## C.7 Comprehensive results of the main experiments

In this section, we report the main results presented in Section 4 in tabular form for 10-node and 20-node graphs. Recall that the hyperparameters of DCDI, CAM and GIES were selected to yield the best likelihood on a held-out data set. However, this is not possible for IGSP, UTIGSP and JCI-PC since they do not have a likelihood model. To make sure these algorithms are represented fairly, we report their performance for different hyperparameter values. For IGSP and UT-IGSP, we report performance for the cutoff hyperparameter $\alpha = \{2e-1, 1e-1, 1e-2, 1e-3, 1e-5, 1e-7, 1e-9\}$. This range was chosen to be around the cutoff values used in [47] and [42]. We used the same range for JCI-PC, but since most runs with $\alpha \leq 1e-5$ would not terminate after 12 hours, we only report results with $\alpha = \{2e-1, 1e-1, 1e-2, 1e-3\}$. The overall ranking of the methods does

Figure 13: Log-likelihood on an unseen interventional distribution of the Sachs data set.

not change for different hyperparameters. To be even fairer to these methods, we also report the performance one obtains by selecting, for every data set, the hyperparameter which yields the lowest SHD. These results are denoted by IGSP*, UTIGSP* and JCI-PC*. Notice that this is unfair to DCDI, CAM and GIES which have not tuned their hyperparameters to minimize SHD or SID. Even in this unfair comparison, DCDI remains very competitive. For IGSP and UTIGSP, we also include results using partial correlation test (indicated with the suffix *-lin*) and KCI-test for every data sets. The reported values in the following tables are the mean and the standard deviation of SHD and SID over ten data sets of each condition. As stated in the main discussion, our conclusions are similar for 10-node graphs: DCDI has competitive performance in almost all conditions and outperforms the other methods for graphs with higher connectivity.

### C.7.1 Perfect interventions

Table 22: Results for linear data set with perfect intervention

| Method | 10 nodes, $e=1$ | | 10 nodes, $e=4$ | | 20 nodes, $e=1$ | | 20 nodes, $e=4$ | |
|---|---|---|---|---|---|---|---|---|
| | SHD | SID | SHD | SID | SHD | SID | SHD | SID |
| IGSP*-lin | $2.2_{\pm2.0}$ | $11.5_{\pm11.4}$ | $23.5_{\pm1.8}$ | $67.3_{\pm3.3}$ | $4.7_{\pm3.7}$ | $19.1_{\pm13.4}$ | $73.4_{\pm7.9}$ | $291.6_{\pm46.4}$ |
| IGSP* | $1.9_{\pm1.8}$ | $8.9_{\pm9.5}$ | $24.6_{\pm3.3}$ | $69.0_{\pm10.3}$ | $9.2_{\pm4.8}$ | $42.5_{\pm31.8}$ | $78.5_{\pm6.8}$ | $337.0_{\pm16.4}$ |
| IGSP($\alpha$=2e-1)-lin | $9.3_{\pm4.1}$ | $18.5_{\pm15.6}$ | $26.4_{\pm3.9}$ | $71.2_{\pm3.9}$ | $37.7_{\pm10.7}$ | $42.9_{\pm37.1}$ | $94.6_{\pm8.9}$ | $271.8_{\pm18.3}$ |
| IGSP($\alpha$=1e-1)-lin | $5.8_{\pm3.5}$ | $17.1_{\pm13.4}$ | $27.4_{\pm2.8}$ | $71.6_{\pm4.0}$ | $18.7_{\pm4.4}$ | $25.9_{\pm12.8}$ | $84.4_{\pm12.2}$ | $264.8_{\pm27.4}$ |
| IGSP($\alpha$=1e-2)-lin | $2.4_{\pm2.1}$ | $11.8_{\pm11.0}$ | $27.6_{\pm4.2}$ | $70.9_{\pm8.2}$ | $7.2_{\pm5.3}$ | $22.8_{\pm17.3}$ | $78.9_{\pm10.6}$ | $278.7_{\pm19.5}$ |
| IGSP($\alpha$=1e-3)-lin | $2.4_{\pm2.1}$ | $11.8_{\pm11.0}$ | $26.9_{\pm4.0}$ | $68.3_{\pm6.8}$ | $8.5_{\pm7.2}$ | $33.3_{\pm29.4}$ | $82.4_{\pm12.1}$ | $304.3_{\pm20.4}$ |
| IGSP($\alpha$=1e-5)-lin | $2.4_{\pm2.1}$ | $11.9_{\pm11.1}$ | $30.6_{\pm3.9}$ | $74.8_{\pm7.0}$ | $9.4_{\pm5.4}$ | $41.1_{\pm36.8}$ | $83.9_{\pm11.1}$ | $327.8_{\pm9.0}$ |
| IGSP($\alpha$=1e-7)-lin | $2.7_{\pm2.5}$ | $13.8_{\pm14.3}$ | $33.7_{\pm3.3}$ | $78.8_{\pm4.8}$ | $8.6_{\pm5.1}$ | $44.2_{\pm36.0}$ | $81.5_{\pm10.6}$ | $338.7_{\pm8.8}$ |
| IGSP($\alpha$=1e-9)-lin | $2.6_{\pm2.5}$ | $13.4_{\pm14.6}$ | $29.3_{\pm3.4}$ | $71.0_{\pm9.7}$ | $11.6_{\pm5.1}$ | $65.1_{\pm45.5}$ | $82.0_{\pm6.4}$ | $341.5_{\pm12.2}$ |
| IGSP($\alpha$=2e-1) | $8.1_{\pm3.4}$ | $10.7_{\pm11.2}$ | $28.6_{\pm5.3}$ | $74.0_{\pm6.3}$ | $51.8_{\pm10.4}$ | $64.7_{\pm46.5}$ | $102.4_{\pm9.8}$ | $311.4_{\pm13.8}$ |
| IGSP($\alpha$=1e-1) | $5.4_{\pm2.8}$ | $13.1_{\pm11.1}$ | $26.7_{\pm3.7}$ | $69.5_{\pm11.1}$ | $31.0_{\pm8.6}$ | $52.0_{\pm31.9}$ | $93.2_{\pm8.2}$ | $314.3_{\pm21.3}$ |
| IGSP($\alpha$=1e-2) | $2.5_{\pm2.0}$ | $10.5_{\pm10.3}$ | $31.0_{\pm3.8}$ | $78.2_{\pm4.8}$ | $12.1_{\pm5.1}$ | $40.4_{\pm22.6}$ | $86.8_{\pm9.5}$ | $336.4_{\pm16.4}$ |
| IGSP($\alpha$=1e-3) | $2.8_{\pm2.5}$ | $13.1_{\pm13.8}$ | $31.3_{\pm2.9}$ | $76.0_{\pm8.1}$ | $12.4_{\pm4.7}$ | $55.6_{\pm30.9}$ | $84.7_{\pm10.1}$ | $346.3_{\pm8.5}$ |
| IGSP($\alpha$=1e-5) | $2.9_{\pm2.7}$ | $13.8_{\pm14.6}$ | $33.3_{\pm2.5}$ | $78.8_{\pm7.1}$ | $12.9_{\pm5.6}$ | $64.9_{\pm35.3}$ | $84.4_{\pm6.1}$ | $347.7_{\pm14.0}$ |
| IGSP($\alpha$=1e-7) | $4.1_{\pm3.9}$ | $15.6_{\pm14.9}$ | $33.0_{\pm3.3}$ | $77.7_{\pm5.4}$ | $15.2_{\pm7.2}$ | $75.6_{\pm43.6}$ | $83.9_{\pm6.6}$ | $350.1_{\pm20.4}$ |
| IGSP($\alpha$=1e-9) | $4.0_{\pm3.6}$ | $16.3_{\pm17.9}$ | $33.6_{\pm3.1}$ | $76.2_{\pm5.6}$ | $16.7_{\pm6.3}$ | $81.9_{\pm35.7}$ | $83.0_{\pm6.7}$ | $339.7_{\pm13.8}$ |
| GIES | $0.6_{\pm1.3}$ | $0.0_{\pm0.0}$ | $2.9_{\pm3.0}$ | $0.0_{\pm0.0}$ | $3.2_{\pm6.3}$ | $1.1_{\pm3.5}$ | $53.1_{\pm25.8}$ | $82.9_{\pm84.9}$ |
| CAM | $1.9_{\pm2.6}$ | $1.7_{\pm3.1}$ | $10.6_{\pm3.1}$ | $34.5_{\pm11.0}$ | $5.4_{\pm7.9}$ | $8.2_{\pm9.6}$ | $91.1_{\pm21.7}$ | $167.8_{\pm55.4}$ |
| DCDI-G | $1.3_{\pm1.9}$ | $0.8_{\pm1.8}$ | $3.3_{\pm2.1}$ | $10.7_{\pm12.0}$ | $5.4_{\pm4.5}$ | $13.4_{\pm12.0}$ | $23.7_{\pm5.6}$ | $112.8_{\pm41.8}$ |
| DCDI-DSF | $0.9_{\pm1.3}$ | $0.6_{\pm1.9}$ | $3.7_{\pm2.3}$ | $18.9_{\pm14.1}$ | $3.6_{\pm2.7}$ | $6.0_{\pm5.4}$ | $16.6_{\pm6.4}$ | $92.5_{\pm40.1}$ |

Table 23: Results for the additive noise model data set with perfect intervention

| Method | 10 nodes, $e=1$ | | 10 nodes, $e=4$ | | 20 nodes, $e=1$ | | 20 nodes, $e=4$ | |
|---|---|---|---|---|---|---|---|---|
| | SHD | SID | SHD | SID | SHD | SID | SHD | SID |
| IGSP*-lin | $7.7_{\pm2.4}$ | $24.1_{\pm11.1}$ | $22.5_{\pm2.0}$ | $64.4_{\pm6.3}$ | $14.2_{\pm5.2}$ | $58.6_{\pm37.5}$ | $75.9_{\pm3.1}$ | $307.1_{\pm25.0}$ |
| IGSP* | $5.3_{\pm3.0}$ | $20.9_{\pm13.9}$ | $25.8_{\pm2.8}$ | $68.0_{\pm9.4}$ | $13.6_{\pm6.6}$ | $69.6_{\pm47.9}$ | $76.7_{\pm6.5}$ | $332.6_{\pm18.2}$ |
| IGSP($\alpha$=2e-1)-lin | $17.0_{\pm5.2}$ | $25.0_{\pm13.1}$ | $27.3_{\pm3.3}$ | $69.2_{\pm7.0}$ | $56.3_{\pm10.5}$ | $78.3_{\pm47.5}$ | $125.3_{\pm7.9}$ | $282.9_{\pm27.2}$ |
| IGSP($\alpha$=1e-1)-lin | $13.2_{\pm5.3}$ | $21.1_{\pm9.8}$ | $27.3_{\pm4.4}$ | $69.4_{\pm5.6}$ | $42.0_{\pm11.9}$ | $73.4_{\pm37.5}$ | $115.8_{\pm11.6}$ | $286.0_{\pm34.6}$ |
| IGSP($\alpha$=1e-2)-lin | $11.4_{\pm4.6}$ | $26.4_{\pm13.9}$ | $27.8_{\pm3.4}$ | $72.4_{\pm4.2}$ | $21.5_{\pm9.6}$ | $64.7_{\pm42.0}$ | $101.0_{\pm10.1}$ | $298.6_{\pm20.2}$ |
| IGSP($\alpha$=1e-3)-lin | $10.4_{\pm3.9}$ | $26.6_{\pm11.8}$ | $26.9_{\pm2.9}$ | $70.2_{\pm7.3}$ | $19.0_{\pm8.0}$ | $58.1_{\pm34.2}$ | $93.2_{\pm4.8}$ | $308.5_{\pm18.3}$ |
| IGSP($\alpha$=1e-5)-lin | $9.7_{\pm2.3}$ | $27.4_{\pm8.8}$ | $28.2_{\pm3.9}$ | $70.2_{\pm9.9}$ | $20.1_{\pm8.6}$ | $84.9_{\pm49.1}$ | $82.9_{\pm5.3}$ | $312.9_{\pm19.6}$ |
| IGSP($\alpha$=1e-7)-lin | $9.2_{\pm2.3}$ | $28.1_{\pm10.4}$ | $27.9_{\pm3.8}$ | $72.5_{\pm8.2}$ | $16.1_{\pm5.2}$ | $63.5_{\pm37.3}$ | $84.1_{\pm8.6}$ | $322.1_{\pm22.4}$ |
| IGSP($\alpha$=1e-9)-lin | $9.8_{\pm2.4}$ | $31.5_{\pm12.3}$ | $30.9_{\pm4.7}$ | $77.7_{\pm5.4}$ | $17.2_{\pm6.3}$ | $73.1_{\pm37.3}$ | $78.7_{\pm5.7}$ | $314.8_{\pm23.9}$ |
| IGSP($\alpha$=2e-1) | $13.3_{\pm4.9}$ | $23.2_{\pm15.9}$ | $28.4_{\pm3.3}$ | $71.5_{\pm8.3}$ | $43.2_{\pm7.6}$ | $55.8_{\pm30.0}$ | $98.0_{\pm11.2}$ | $302.3_{\pm34.7}$ |
| IGSP($\alpha$=1e-1) | $9.7_{\pm5.3}$ | $21.8_{\pm14.6}$ | $29.0_{\pm2.9}$ | $73.4_{\pm4.9}$ | $30.6_{\pm6.4}$ | $64.7_{\pm41.5}$ | $88.9_{\pm9.2}$ | $320.9_{\pm16.2}$ |
| IGSP($\alpha$=1e-2) | $7.3_{\pm3.3}$ | $21.9_{\pm11.3}$ | $31.4_{\pm2.5}$ | $74.3_{\pm9.7}$ | $17.2_{\pm6.0}$ | $74.7_{\pm40.2}$ | $84.1_{\pm10.1}$ | $322.8_{\pm15.8}$ |
| IGSP($\alpha$=1e-3) | $7.8_{\pm3.4}$ | $24.2_{\pm12.1}$ | $29.6_{\pm3.8}$ | $75.1_{\pm5.6}$ | $16.5_{\pm8.9}$ | $79.6_{\pm53.6}$ | $85.1_{\pm7.7}$ | $334.2_{\pm22.0}$ |
| IGSP($\alpha$=1e-5) | $8.1_{\pm4.0}$ | $29.2_{\pm15.3}$ | $30.5_{\pm4.2}$ | $77.3_{\pm4.7}$ | $16.6_{\pm6.6}$ | $79.7_{\pm50.3}$ | $81.2_{\pm8.2}$ | $324.4_{\pm26.0}$ |
| IGSP($\alpha$=1e-7) | $7.3_{\pm2.8}$ | $28.5_{\pm11.1}$ | $33.0_{\pm1.8}$ | $78.3_{\pm4.0}$ | $15.3_{\pm6.2}$ | $75.0_{\pm45.4}$ | $82.5_{\pm6.8}$ | $334.3_{\pm22.8}$ |
| IGSP($\alpha$=1e-9) | $9.4_{\pm5.2}$ | $34.3_{\pm15.6}$ | $30.9_{\pm3.9}$ | $73.7_{\pm10.3}$ | $15.3_{\pm6.7}$ | $78.2_{\pm50.6}$ | $81.6_{\pm10.8}$ | $333.4_{\pm17.2}$ |
| GIES | $9.1_{\pm8.5}$ | $1.8_{\pm3.6}$ | $9.0_{\pm2.7}$ | $23.8_{\pm15.6}$ | $40.3_{\pm61.0}$ | $7.5_{\pm7.2}$ | $103.2_{\pm18.6}$ | $120.1_{\pm68.5}$ |
| CAM | $5.2_{\pm3.0}$ | $1.0_{\pm1.9}$ | $8.5_{\pm3.7}$ | $11.5_{\pm13.4}$ | $7.5_{\pm6.0}$ | $5.6_{\pm4.9}$ | $105.7_{\pm13.2}$ | $108.7_{\pm61.0}$ |
| DCDI-G | $5.2_{\pm7.5}$ | $2.4_{\pm4.9}$ | $4.3_{\pm2.4}$ | $16.0_{\pm11.9}$ | $21.8_{\pm30.1}$ | $11.6_{\pm13.1}$ | $35.2_{\pm13.2}$ | $109.8_{\pm44.6}$ |
| DCDI-DSF | $4.2_{\pm5.6}$ | $5.6_{\pm5.5}$ | $5.5_{\pm2.4}$ | $23.9_{\pm14.3}$ | $4.3_{\pm1.9}$ | $19.7_{\pm12.6}$ | $26.7_{\pm16.9}$ | $105.3_{\pm22.7}$ |

Table 24: Results for the nonlinear with non-additive noise data set with perfect intervention

| Method | 10 nodes, $e=1$ | | 10 nodes, $e=4$ | | 20 nodes, $e=1$ | | 20 nodes, $e=4$ | |
|---|---|---|---|---|---|---|---|---|
| | SHD | SID | SHD | SID | SHD | SID | SHD | SID |
| IGSP*-lin | $4.4_{\pm2.6}$ | $15.2_{\pm11.0}$ | $23.3_{\pm1.9}$ | $66.0_{\pm7.9}$ | $13.4_{\pm3.2}$ | $67.4_{\pm27.8}$ | $67.0_{\pm9.6}$ | $318.4_{\pm19.1}$ |
| IGSP* | $4.1_{\pm2.8}$ | $13.6_{\pm11.9}$ | $25.4_{\pm3.8}$ | $69.4_{\pm5.3}$ | $15.3_{\pm4.5}$ | $73.0_{\pm28.8}$ | $72.7_{\pm9.6}$ | $329.4_{\pm21.5}$ |
| IGSP($\alpha$=2e-1)-lin | $12.4_{\pm4.5}$ | $15.2_{\pm9.1}$ | $27.6_{\pm3.9}$ | $70.1_{\pm6.3}$ | $51.4_{\pm9.1}$ | $72.5_{\pm31.1}$ | $102.2_{\pm11.5}$ | $297.1_{\pm27.5}$ |
| IGSP($\alpha$=1e-1)-lin | $9.7_{\pm4.7}$ | $17.5_{\pm13.4}$ | $26.5_{\pm3.1}$ | $68.5_{\pm8.1}$ | $35.8_{\pm9.2}$ | $83.4_{\pm35.1}$ | $93.6_{\pm10.2}$ | $293.9_{\pm25.3}$ |
| IGSP($\alpha$=1e-2)-lin | $7.1_{\pm3.4}$ | $16.4_{\pm12.5}$ | $28.7_{\pm2.7}$ | $72.7_{\pm4.9}$ | $19.0_{\pm5.1}$ | $73.7_{\pm33.7}$ | $76.0_{\pm12.9}$ | $315.7_{\pm14.2}$ |
| IGSP($\alpha$=1e-3)-lin | $5.9_{\pm3.5}$ | $15.9_{\pm10.5}$ | $29.6_{\pm3.0}$ | $75.0_{\pm2.8}$ | $16.4_{\pm4.4}$ | $77.1_{\pm32.2}$ | $75.0_{\pm11.0}$ | $325.1_{\pm17.7}$ |
| IGSP($\alpha$=1e-5)-lin | $6.6_{\pm3.0}$ | $21.1_{\pm12.3}$ | $27.7_{\pm3.4}$ | $73.6_{\pm4.8}$ | $15.9_{\pm5.7}$ | $79.6_{\pm22.5}$ | $73.3_{\pm12.7}$ | $323.2_{\pm16.0}$ |
| IGSP($\alpha$=1e-7)-lin | $7.2_{\pm4.3}$ | $24.3_{\pm15.9}$ | $30.1_{\pm4.1}$ | $75.4_{\pm5.9}$ | $17.3_{\pm3.9}$ | $84.1_{\pm22.1}$ | $73.2_{\pm11.2}$ | $325.5_{\pm23.1}$ |
| IGSP($\alpha$=1e-9)-lin | $5.9_{\pm3.5}$ | $20.9_{\pm16.1}$ | $31.3_{\pm2.1}$ | $76.6_{\pm4.0}$ | $19.2_{\pm4.2}$ | $94.4_{\pm29.9}$ | $77.4_{\pm11.3}$ | $347.2_{\pm15.5}$ |
| IGSP($\alpha$=2e-1) | $10.6_{\pm2.7}$ | $12.4_{\pm4.9}$ | $27.0_{\pm3.0}$ | $70.8_{\pm4.1}$ | $48.2_{\pm7.7}$ | $97.5_{\pm29.8}$ | $89.5_{\pm15.5}$ | $306.3_{\pm17.1}$ |
| IGSP($\alpha$=1e-1) | $7.7_{\pm4.1}$ | $12.1_{\pm8.8}$ | $27.5_{\pm5.0}$ | $73.0_{\pm5.2}$ | $32.3_{\pm7.1}$ | $87.5_{\pm39.9}$ | $89.4_{\pm16.4}$ | $325.4_{\pm21.6}$ |
| IGSP($\alpha$=1e-2) | $5.4_{\pm2.5}$ | $15.3_{\pm6.4}$ | $29.5_{\pm3.5}$ | $74.2_{\pm4.9}$ | $19.5_{\pm5.2}$ | $82.5_{\pm38.5}$ | $83.0_{\pm9.5}$ | $337.3_{\pm15.9}$ |
| IGSP($\alpha$=1e-3) | $6.6_{\pm4.1}$ | $21.7_{\pm14.5}$ | $31.3_{\pm3.8}$ | $75.9_{\pm7.7}$ | $17.3_{\pm6.1}$ | $83.3_{\pm36.2}$ | $80.4_{\pm11.9}$ | $331.0_{\pm23.7}$ |
| IGSP($\alpha$=1e-5) | $6.3_{\pm3.1}$ | $19.8_{\pm12.1}$ | $34.0_{\pm4.2}$ | $76.8_{\pm12.0}$ | $19.3_{\pm4.6}$ | $90.8_{\pm32.6}$ | $77.0_{\pm9.5}$ | $345.2_{\pm9.8}$ |
| IGSP($\alpha$=1e-7) | $6.3_{\pm3.3}$ | $21.4_{\pm13.1}$ | $34.1_{\pm1.9}$ | $78.5_{\pm8.4}$ | $19.1_{\pm4.0}$ | $91.6_{\pm29.0}$ | $75.8_{\pm11.1}$ | $344.4_{\pm16.6}$ |
| IGSP($\alpha$=1e-9) | $5.9_{\pm3.7}$ | $21.7_{\pm15.9}$ | $34.6_{\pm2.6}$ | $79.7_{\pm6.2}$ | $18.8_{\pm3.9}$ | $94.0_{\pm33.8}$ | $77.5_{\pm9.0}$ | $341.4_{\pm24.5}$ |
| GIES | $4.4_{\pm6.1}$ | $1.0_{\pm1.6}$ | $7.9_{\pm4.7}$ | $25.5_{\pm13.2}$ | $26.9_{\pm50.5}$ | $9.5_{\pm7.4}$ | $80.1_{\pm36.2}$ | $96.7_{\pm59.1}$ |
| CAM | $1.8_{\pm1.5}$ | $2.8_{\pm4.4}$ | $7.9_{\pm3.6}$ | $26.7_{\pm19.0}$ | $6.1_{\pm5.2}$ | $18.1_{\pm16.3}$ | $101.8_{\pm24.5}$ | $142.5_{\pm49.1}$ |
| DCDI-G | $2.3_{\pm3.6}$ | $2.7_{\pm3.3}$ | $2.4_{\pm1.6}$ | $13.9_{\pm8.5}$ | $13.9_{\pm20.3}$ | $13.7_{\pm8.1}$ | $16.8_{\pm8.7}$ | $82.5_{\pm38.1}$ |
| DCDI-DSF | $7.0_{\pm10.7}$ | $7.8_{\pm5.8}$ | $1.6_{\pm1.6}$ | $7.7_{\pm13.8}$ | $8.3_{\pm4.1}$ | $32.4_{\pm17.3}$ | $11.8_{\pm2.1}$ | $102.3_{\pm34.5}$ |

## C.7.2 Imperfect interventions

Table 25: Results for the linear data set with imperfect intervention

| Method | 10 nodes, $e=1$ | | 10 nodes, $e=4$ | | 20 nodes, $e=1$ | | 20 nodes, $e=4$ | |
| --- | --- | --- | --- | --- | --- | --- | --- | --- |
| | SHD | SID | SHD | SID | SHD | SID | SHD | SID |
| IGSP*-lin | $2.1_{\pm0.9}$ | $11.7_{\pm6.7}$ | $20.7_{\pm5.8}$ | $61.4_{\pm11.0}$ | $4.0_{\pm2.9}$ | $17.9_{\pm12.9}$ | $62.2_{\pm12.0}$ | $256.8_{\pm35.5}$ |
| IGSP* | $3.4_{\pm1.8}$ | $14.9_{\pm12.4}$ | $24.1_{\pm2.5}$ | $68.9_{\pm9.3}$ | $8.0_{\pm5.7}$ | $43.8_{\pm33.6}$ | $75.3_{\pm9.2}$ | $338.3_{\pm22.3}$ |
| IGSP($\alpha$=2e-1)-lin | $8.5_{\pm2.7}$ | $15.5_{\pm8.0}$ | $23.2_{\pm7.3}$ | $65.8_{\pm11.3}$ | $45.3_{\pm9.5}$ | $48.0_{\pm28.4}$ | $86.1_{\pm15.0}$ | $253.7_{\pm29.8}$ |
| IGSP($\alpha$=1e-1)-lin | $4.5_{\pm3.3}$ | $15.3_{\pm10.8}$ | $24.4_{\pm6.6}$ | $65.4_{\pm12.6}$ | $23.4_{\pm9.9}$ | $47.3_{\pm31.8}$ | $80.5_{\pm13.7}$ | $259.4_{\pm27.2}$ |
| IGSP($\alpha$=1e-2)-lin | $2.8_{\pm1.9}$ | $12.8_{\pm6.6}$ | $26.1_{\pm4.8}$ | $69.7_{\pm8.8}$ | $6.6_{\pm4.4}$ | $20.2_{\pm13.3}$ | $68.2_{\pm13.7}$ | $279.2_{\pm22.4}$ |
| IGSP($\alpha$=1e-3)-lin | $3.9_{\pm2.8}$ | $17.2_{\pm9.1}$ | $26.4_{\pm5.6}$ | $71.1_{\pm9.7}$ | $7.0_{\pm5.9}$ | $33.2_{\pm26.3}$ | $70.6_{\pm16.2}$ | $296.3_{\pm20.8}$ |
| IGSP($\alpha$=1e-5)-lin | $4.3_{\pm2.6}$ | $21.4_{\pm13.4}$ | $29.2_{\pm5.1}$ | $75.3_{\pm7.4}$ | $8.1_{\pm5.0}$ | $45.4_{\pm39.9}$ | $75.5_{\pm7.7}$ | $325.3_{\pm21.3}$ |
| IGSP($\alpha$=1e-7)-lin | $3.4_{\pm1.3}$ | $19.1_{\pm10.1}$ | $29.1_{\pm3.9}$ | $74.8_{\pm6.6}$ | $10.7_{\pm5.1}$ | $52.8_{\pm33.3}$ | $77.9_{\pm9.2}$ | $333.1_{\pm16.7}$ |
| IGSP($\alpha$=1e-9)-lin | $4.6_{\pm3.3}$ | $23.7_{\pm20.4}$ | $31.3_{\pm4.1}$ | $79.1_{\pm5.7}$ | $10.5_{\pm5.0}$ | $61.6_{\pm33.9}$ | $78.0_{\pm8.1}$ | $343.4_{\pm23.9}$ |
| IGSP($\alpha$=2e-1) | $9.5_{\pm3.6}$ | $21.5_{\pm13.6}$ | $27.7_{\pm5.4}$ | $70.9_{\pm10.4}$ | $46.9_{\pm10.3}$ | $64.1_{\pm34.6}$ | $95.5_{\pm8.6}$ | $306.0_{\pm20.0}$ |
| IGSP($\alpha$=1e-1) | $5.6_{\pm2.2}$ | $15.9_{\pm16.0}$ | $26.8_{\pm5.3}$ | $68.8_{\pm9.8}$ | $32.3_{\pm9.6}$ | $54.3_{\pm30.5}$ | $89.0_{\pm9.7}$ | $315.5_{\pm20.6}$ |
| IGSP($\alpha$=1e-2) | $5.0_{\pm2.8}$ | $20.2_{\pm15.3}$ | $32.0_{\pm3.2}$ | $76.3_{\pm5.3}$ | $11.8_{\pm9.1}$ | $48.8_{\pm43.6}$ | $82.7_{\pm12.5}$ | $339.2_{\pm11.7}$ |
| IGSP($\alpha$=1e-3) | $4.0_{\pm2.7}$ | $19.9_{\pm14.3}$ | $31.0_{\pm4.1}$ | $76.4_{\pm6.8}$ | $10.8_{\pm6.0}$ | $56.6_{\pm32.3}$ | $82.6_{\pm8.6}$ | $347.3_{\pm8.3}$ |
| IGSP($\alpha$=1e-5) | $5.4_{\pm4.4}$ | $23.3_{\pm19.8}$ | $30.9_{\pm4.1}$ | $80.4_{\pm2.9}$ | $12.7_{\pm6.9}$ | $71.2_{\pm41.5}$ | $80.3_{\pm9.6}$ | $347.6_{\pm12.6}$ |
| IGSP($\alpha$=1e-7) | $5.1_{\pm2.4}$ | $21.6_{\pm12.7}$ | $31.4_{\pm2.7}$ | $79.5_{\pm3.4}$ | $13.8_{\pm7.4}$ | $80.4_{\pm42.1}$ | $82.2_{\pm7.3}$ | $351.0_{\pm13.7}$ |
| IGSP($\alpha$=1e-9) | $6.5_{\pm3.3}$ | $28.0_{\pm18.4}$ | $30.6_{\pm3.9}$ | $78.3_{\pm4.4}$ | $15.3_{\pm7.7}$ | $80.3_{\pm45.2}$ | $83.0_{\pm8.8}$ | $351.4_{\pm8.6}$ |
| GIES | $13.7_{\pm11.9}$ | $20.9_{\pm19.4}$ | $14.2_{\pm7.1}$ | $47.1_{\pm16.8}$ | $33.7_{\pm48.8}$ | $20.8_{\pm22.4}$ | $78.7_{\pm40.4}$ | $194.1_{\pm61.0}$ |
| CAM | $8.1_{\pm6.2}$ | $22.6_{\pm18.8}$ | $19.4_{\pm4.7}$ | $56.0_{\pm10.1}$ | $10.5_{\pm5.8}$ | $36.3_{\pm23.6}$ | $111.7_{\pm16.5}$ | $232.5_{\pm23.4}$ |
| DCDI-G | $2.7_{\pm2.8}$ | $8.2_{\pm8.8}$ | $5.2_{\pm3.5}$ | $25.1_{\pm12.9}$ | $10.8_{\pm12.0}$ | $27.0_{\pm21.3}$ | $34.7_{\pm7.1}$ | $188.0_{\pm48.8}$ |
| DCDI-DSF | $1.3_{\pm1.3}$ | $4.2_{\pm4.0}$ | $1.7_{\pm2.4}$ | $10.2_{\pm14.9}$ | $7.0_{\pm4.0}$ | $21.0_{\pm12.5}$ | $18.9_{\pm5.9}$ | $133.6_{\pm33.9}$ |

Table 26: Results for the additive noise model data set with imperfect intervention

| Method | 10 nodes, $e=1$ | | 10 nodes, $e=4$ | | 20 nodes, $e=1$ | | 20 nodes, $e=4$ | |
| --- | --- | --- | --- | --- | --- | --- | --- | --- |
| | SHD | SID | SHD | SID | SHD | SID | SHD | SID |
| IGSP*-lin | $9.1_{\pm4.4}$ | $23.6_{\pm12.7}$ | $22.8_{\pm4.6}$ | $62.0_{\pm9.7}$ | $18.1_{\pm6.0}$ | $67.5_{\pm26.2}$ | $81.2_{\pm8.8}$ | $322.9_{\pm13.8}$ |
| IGSP* | $6.2_{\pm3.4}$ | $15.8_{\pm9.9}$ | $27.6_{\pm2.3}$ | $67.2_{\pm8.7}$ | $17.0_{\pm3.9}$ | $79.7_{\pm33.9}$ | $75.7_{\pm7.4}$ | $321.0_{\pm23.8}$ |
| IGSP($\alpha$=2e-1)-lin | $19.7_{\pm3.4}$ | $29.9_{\pm14.9}$ | $26.0_{\pm5.0}$ | $67.0_{\pm11.7}$ | $59.0_{\pm11.9}$ | $87.0_{\pm40.0}$ | $123.7_{\pm10.4}$ | $279.5_{\pm27.6}$ |
| IGSP($\alpha$=1e-1)-lin | $17.8_{\pm5.5}$ | $35.4_{\pm14.1}$ | $26.1_{\pm5.5}$ | $68.9_{\pm9.5}$ | $40.1_{\pm12.0}$ | $71.4_{\pm39.3}$ | $119.2_{\pm10.8}$ | $285.5_{\pm21.3}$ |
| IGSP($\alpha$=1e-2)-lin | $13.0_{\pm4.7}$ | $28.1_{\pm12.0}$ | $27.7_{\pm3.0}$ | $70.0_{\pm5.8}$ | $24.9_{\pm9.9}$ | $67.1_{\pm35.0}$ | $109.6_{\pm11.6}$ | $291.6_{\pm29.8}$ |
| IGSP($\alpha$=1e-3)-lin | $13.1_{\pm6.0}$ | $30.6_{\pm16.0}$ | $28.7_{\pm3.6}$ | $71.8_{\pm6.2}$ | $24.4_{\pm9.0}$ | $68.8_{\pm24.9}$ | $96.5_{\pm10.6}$ | $303.7_{\pm17.3}$ |
| IGSP($\alpha$=1e-5)-lin | $11.5_{\pm7.3}$ | $31.0_{\pm17.4}$ | $28.8_{\pm6.0}$ | $69.6_{\pm12.8}$ | $21.6_{\pm5.1}$ | $81.3_{\pm32.2}$ | $90.4_{\pm10.8}$ | $314.1_{\pm15.3}$ |
| IGSP($\alpha$=1e-7)-lin | $10.6_{\pm5.8}$ | $31.0_{\pm15.8}$ | $29.5_{\pm5.0}$ | $74.1_{\pm8.1}$ | $23.3_{\pm5.1}$ | $93.2_{\pm35.9}$ | $84.2_{\pm8.9}$ | $329.3_{\pm15.6}$ |
| IGSP($\alpha$=1e-9)-lin | $11.0_{\pm6.4}$ | $34.0_{\pm20.7}$ | $29.7_{\pm2.8}$ | $69.7_{\pm9.5}$ | $21.3_{\pm5.7}$ | $86.3_{\pm29.7}$ | $83.4_{\pm8.1}$ | $328.5_{\pm19.2}$ |
| IGSP($\alpha$=2e-1) | $11.4_{\pm4.2}$ | $23.8_{\pm16.0}$ | $29.0_{\pm3.2}$ | $72.1_{\pm7.5}$ | $48.0_{\pm8.3}$ | $77.8_{\pm42.6}$ | $97.5_{\pm12.8}$ | $307.5_{\pm23.7}$ |
| IGSP($\alpha$=1e-1) | $10.6_{\pm5.1}$ | $26.2_{\pm15.8}$ | $31.3_{\pm3.3}$ | $73.7_{\pm7.1}$ | $36.9_{\pm6.1}$ | $86.9_{\pm42.6}$ | $88.8_{\pm11.1}$ | $318.5_{\pm25.8}$ |
| IGSP($\alpha$=1e-2) | $9.1_{\pm4.4}$ | $24.3_{\pm11.5}$ | $32.4_{\pm4.1}$ | $76.9_{\pm6.8}$ | $20.9_{\pm6.2}$ | $84.8_{\pm39.9}$ | $86.1_{\pm8.4}$ | $334.3_{\pm14.2}$ |
| IGSP($\alpha$=1e-3) | $8.2_{\pm4.5}$ | $24.5_{\pm13.5}$ | $32.7_{\pm2.2}$ | $78.2_{\pm8.3}$ | $19.3_{\pm4.4}$ | $78.8_{\pm32.2}$ | $82.9_{\pm5.7}$ | $325.1_{\pm19.7}$ |
| IGSP($\alpha$=1e-5) | $8.0_{\pm3.8}$ | $25.8_{\pm14.2}$ | $33.8_{\pm2.4}$ | $79.4_{\pm4.1}$ | $21.4_{\pm5.4}$ | $91.8_{\pm40.5}$ | $83.1_{\pm7.8}$ | $343.4_{\pm14.3}$ |
| IGSP($\alpha$=1e-7) | $8.4_{\pm4.3}$ | $27.6_{\pm15.3}$ | $33.2_{\pm1.9}$ | $78.1_{\pm5.9}$ | $20.3_{\pm4.7}$ | $87.2_{\pm39.6}$ | $85.6_{\pm7.4}$ | $334.9_{\pm25.2}$ |
| IGSP($\alpha$=1e-9) | $8.4_{\pm4.5}$ | $28.3_{\pm16.3}$ | $34.4_{\pm3.4}$ | $79.9_{\pm4.4}$ | $19.6_{\pm3.1}$ | $90.1_{\pm33.1}$ | $79.1_{\pm7.4}$ | $332.5_{\pm20.9}$ |
| GIES | $19.9_{\pm10.4}$ | $23.0_{\pm10.1}$ | $18.9_{\pm6.0}$ | $59.5_{\pm11.2}$ | $74.4_{\pm59.8}$ | $56.4_{\pm43.1}$ | $112.2_{\pm23.8}$ | $245.2_{\pm36.1}$ |
| CAM | $11.2_{\pm9.3}$ | $7.8_{\pm8.7}$ | $9.6_{\pm3.0}$ | $25.2_{\pm10.8}$ | $16.3_{\pm9.9}$ | $26.7_{\pm27.2}$ | $121.9_{\pm11.6}$ | $155.4_{\pm41.5}$ |
| DCDI-G | $6.2_{\pm5.4}$ | $7.6_{\pm11.0}$ | $13.1_{\pm2.9}$ | $48.1_{\pm9.1}$ | $26.0_{\pm34.6}$ | $23.3_{\pm25.7}$ | $36.4_{\pm13.4}$ | $88.5_{\pm43.8}$ |
| DCDI-DSF | $13.4_{\pm8.4}$ | $17.9_{\pm10.5}$ | $14.4_{\pm2.4}$ | $53.2_{\pm8.2}$ | $15.2_{\pm2.7}$ | $49.4_{\pm26.7}$ | $44.6_{\pm15.4}$ | $149.8_{\pm26.0}$ |

Table 27: Results for the nonlinear with non-additive noise data set with imperfect intervention

| Method | 10 nodes, $e=1$ SHD | SID | 10 nodes, $e=4$ SHD | SID | 20 nodes, $e=1$ SHD | SID | 20 nodes, $e=4$ SHD | SID |
|---|---|---|---|---|---|---|---|---|
| IGSP*-lin | $5.6_{\pm 3.6}$ | $23.0_{\pm 19.6}$ | $22.5_{\pm 2.9}$ | $63.4_{\pm 6.7}$ | $13.8_{\pm 6.9}$ | $86.0_{\pm 71.7}$ | $65.1_{\pm 12.0}$ | $315.4_{\pm 46.2}$ |
| IGSP* | $5.1_{\pm 4.3}$ | $20.8_{\pm 16.5}$ | $24.3_{\pm 2.9}$ | $69.1_{\pm 5.5}$ | $18.2_{\pm 7.9}$ | $100.3_{\pm 74.7}$ | $71.7_{\pm 5.2}$ | $331.7_{\pm 35.9}$ |
| IGSP($\alpha$=2e-1)-lin | $14.1_{\pm 6.1}$ | $30.8_{\pm 21.8}$ | $26.9_{\pm 4.1}$ | $70.1_{\pm 5.8}$ | $49.7_{\pm 13.2}$ | $89.7_{\pm 64.3}$ | $100.2_{\pm 8.8}$ | $297.2_{\pm 13.9}$ |
| IGSP($\alpha$=1e-1)-lin | $9.8_{\pm 4.8}$ | $24.9_{\pm 23.0}$ | $25.5_{\pm 4.6}$ | $68.1_{\pm 7.1}$ | $39.7_{\pm 12.3}$ | $104.9_{\pm 62.7}$ | $90.2_{\pm 13.0}$ | $289.0_{\pm 32.7}$ |
| IGSP($\alpha$=1e-2)-lin | $8.0_{\pm 4.4}$ | $29.6_{\pm 22.6}$ | $26.4_{\pm 3.8}$ | $69.9_{\pm 4.0}$ | $18.1_{\pm 8.2}$ | $88.6_{\pm 58.7}$ | $70.6_{\pm 13.0}$ | $301.0_{\pm 40.9}$ |
| IGSP($\alpha$=1e-3)-lin | $7.6_{\pm 4.8}$ | $26.9_{\pm 22.4}$ | $28.4_{\pm 2.3}$ | $73.7_{\pm 3.7}$ | $16.3_{\pm 8.8}$ | $88.5_{\pm 72.6}$ | $72.9_{\pm 8.7}$ | $326.0_{\pm 18.1}$ |
| IGSP($\alpha$=1e-5)-lin | $7.7_{\pm 5.3}$ | $29.2_{\pm 24.7}$ | $27.2_{\pm 4.0}$ | $69.3_{\pm 8.6}$ | $18.9_{\pm 6.9}$ | $112.2_{\pm 64.6}$ | $70.7_{\pm 9.8}$ | $320.2_{\pm 27.6}$ |
| IGSP($\alpha$=1e-7)-lin | $6.7_{\pm 4.6}$ | $26.3_{\pm 19.9}$ | $28.8_{\pm 3.9}$ | $73.1_{\pm 5.8}$ | $16.8_{\pm 7.2}$ | $106.1_{\pm 63.8}$ | $72.6_{\pm 9.9}$ | $338.0_{\pm 17.2}$ |
| IGSP($\alpha$=1e-9)-lin | $7.7_{\pm 4.3}$ | $29.2_{\pm 17.9}$ | $30.0_{\pm 3.2}$ | $74.4_{\pm 7.4}$ | $17.7_{\pm 6.8}$ | $119.8_{\pm 77.9}$ | $72.3_{\pm 9.6}$ | $337.1_{\pm 23.8}$ |
| IGSP($\alpha$=2e-1) | $12.5_{\pm 5.5}$ | $27.9_{\pm 21.0}$ | $26.7_{\pm 4.4}$ | $71.7_{\pm 4.2}$ | $52.9_{\pm 6.6}$ | $113.0_{\pm 64.2}$ | $91.7_{\pm 7.6}$ | $311.0_{\pm 15.9}$ |
| IGSP($\alpha$=1e-1) | $9.5_{\pm 5.4}$ | $26.7_{\pm 24.0}$ | $26.2_{\pm 4.7}$ | $70.6_{\pm 6.4}$ | $37.1_{\pm 10.1}$ | $113.0_{\pm 79.7}$ | $79.5_{\pm 9.0}$ | $318.2_{\pm 30.3}$ |
| IGSP($\alpha$=1e-2) | $7.3_{\pm 4.5}$ | $26.9_{\pm 19.4}$ | $28.4_{\pm 3.3}$ | $73.9_{\pm 4.3}$ | $20.9_{\pm 7.7}$ | $100.1_{\pm 71.9}$ | $77.5_{\pm 7.5}$ | $324.7_{\pm 28.7}$ |
| IGSP($\alpha$=1e-3) | $7.4_{\pm 5.2}$ | $29.8_{\pm 21.8}$ | $29.6_{\pm 2.9}$ | $76.0_{\pm 3.0}$ | $22.4_{\pm 7.8}$ | $125.9_{\pm 89.4}$ | $76.2_{\pm 7.6}$ | $343.4_{\pm 21.3}$ |
| IGSP($\alpha$=1e-5) | $6.6_{\pm 5.1}$ | $24.9_{\pm 20.4}$ | $31.0_{\pm 2.4}$ | $76.5_{\pm 4.7}$ | $19.6_{\pm 8.4}$ | $114.6_{\pm 79.9}$ | $74.4_{\pm 5.4}$ | $335.7_{\pm 24.3}$ |
| IGSP($\alpha$=1e-7) | $6.8_{\pm 5.2}$ | $25.5_{\pm 20.2}$ | $32.6_{\pm 3.3}$ | $77.7_{\pm 7.2}$ | $21.3_{\pm 10.0}$ | $129.2_{\pm 92.4}$ | $76.4_{\pm 5.6}$ | $341.0_{\pm 26.0}$ |
| IGSP($\alpha$=1e-9) | $6.8_{\pm 4.4}$ | $25.7_{\pm 18.9}$ | $33.0_{\pm 2.4}$ | $77.2_{\pm 6.7}$ | $21.3_{\pm 9.1}$ | $127.6_{\pm 92.8}$ | $76.8_{\pm 6.5}$ | $348.4_{\pm 18.5}$ |
| GIES | $13.2_{\pm 11.2}$ | $16.7_{\pm 13.9}$ | $18.1_{\pm 5.6}$ | $53.7_{\pm 15.0}$ | $36.8_{\pm 41.1}$ | $67.0_{\pm 46.3}$ | $92.7_{\pm 29.4}$ | $215.8_{\pm 63.9}$ |
| CAM | $4.3_{\pm 3.3}$ | $9.3_{\pm 6.8}$ | $14.7_{\pm 5.1}$ | $45.7_{\pm 14.9}$ | $20.7_{\pm 16.2}$ | $53.9_{\pm 32.9}$ | $121.5_{\pm 9.3}$ | $194.1_{\pm 40.3}$ |
| DCDI-G | $3.9_{\pm 3.9}$ | $7.5_{\pm 6.5}$ | $7.4_{\pm 2.7}$ | $29.8_{\pm 11.0}$ | $10.0_{\pm 14.0}$ | $39.2_{\pm 41.5}$ | $20.9_{\pm 7.2}$ | $124.0_{\pm 39.0}$ |
| DCDI-DSF | $5.3_{\pm 4.2}$ | $16.3_{\pm 10.0}$ | $5.6_{\pm 3.1}$ | $32.4_{\pm 14.6}$ | $12.4_{\pm 5.3}$ | $70.3_{\pm 55.2}$ | $16.4_{\pm 4.9}$ | $139.7_{\pm 42.6}$ |

### C.7.3 Unknown interventions

Table 28: Results for the linear data set with perfect intervention with unknown targets

| Method | 10 nodes, $e=1$ SHD | SID | 10 nodes, $e=4$ SHD | SID | 20 nodes, $e=1$ SHD | SID | 20 nodes, $e=4$ SHD | SID |
|---|---|---|---|---|---|---|---|---|
| UTIGSP*-lin | $0.7_{\pm 1.6}$ | $3.4_{\pm 8.4}$ | $21.1_{\pm 3.6}$ | $62.9_{\pm 6.0}$ | $3.9_{\pm 3.6}$ | $14.6_{\pm 9.1}$ | $67.9_{\pm 10.8}$ | $271.6_{\pm 38.6}$ |
| UTIGSP* | $1.7_{\pm 2.0}$ | $7.4_{\pm 10.7}$ | $25.8_{\pm 2.5}$ | $67.4_{\pm 8.7}$ | $14.3_{\pm 4.8}$ | $65.5_{\pm 32.2}$ | $77.9_{\pm 5.5}$ | $332.2_{\pm 19.7}$ |
| UTIGSP($\alpha$=2e-1)-lin | $7.7_{\pm 3.7}$ | $15.1_{\pm 15.4}$ | $24.5_{\pm 6.1}$ | $67.6_{\pm 8.0}$ | $37.6_{\pm 10.2}$ | $44.4_{\pm 32.6}$ | $95.9_{\pm 9.7}$ | $265.6_{\pm 24.5}$ |
| UTIGSP($\alpha$=1e-1)-lin | $3.7_{\pm 3.2}$ | $10.2_{\pm 12.6}$ | $26.4_{\pm 2.9}$ | $68.9_{\pm 6.5}$ | $18.4_{\pm 5.1}$ | $16.8_{\pm 7.4}$ | $83.4_{\pm 13.1}$ | $255.8_{\pm 20.3}$ |
| UTIGSP($\alpha$=1e-2)-lin | $1.7_{\pm 2.1}$ | $7.0_{\pm 9.3}$ | $27.2_{\pm 5.8}$ | $70.1_{\pm 9.8}$ | $4.6_{\pm 4.0}$ | $13.9_{\pm 11.1}$ | $70.1_{\pm 12.0}$ | $271.2_{\pm 19.9}$ |
| UTIGSP($\alpha$=1e-3)-lin | $1.6_{\pm 2.2}$ | $7.2_{\pm 10.1}$ | $29.6_{\pm 5.5}$ | $73.1_{\pm 9.4}$ | $6.9_{\pm 6.5}$ | $25.6_{\pm 31.6}$ | $81.0_{\pm 12.7}$ | $301.1_{\pm 17.6}$ |
| UTIGSP($\alpha$=1e-5)-lin | $1.2_{\pm 1.9}$ | $5.1_{\pm 8.7}$ | $29.4_{\pm 4.2}$ | $73.2_{\pm 7.1}$ | $8.8_{\pm 6.0}$ | $36.7_{\pm 29.9}$ | $81.5_{\pm 11.7}$ | $323.1_{\pm 14.1}$ |
| UTIGSP($\alpha$=1e-7)-lin | $1.8_{\pm 2.6}$ | $7.6_{\pm 13.4}$ | $29.4_{\pm 3.4}$ | $72.3_{\pm 9.6}$ | $8.8_{\pm 5.5}$ | $43.3_{\pm 40.1}$ | $84.8_{\pm 9.7}$ | $339.6_{\pm 11.8}$ |
| UTIGSP($\alpha$=1e-9)-lin | $1.8_{\pm 2.4}$ | $7.8_{\pm 13.5}$ | $29.2_{\pm 3.8}$ | $70.2_{\pm 7.5}$ | $11.6_{\pm 7.3}$ | $57.3_{\pm 48.4}$ | $81.2_{\pm 5.7}$ | $339.4_{\pm 13.7}$ |
| UTIGSP($\alpha$=2e-1) | $8.5_{\pm 3.0}$ | $9.6_{\pm 8.6}$ | $27.8_{\pm 4.7}$ | $70.7_{\pm 10.4}$ | $50.3_{\pm 15.2}$ | $65.1_{\pm 49.2}$ | $106.7_{\pm 9.7}$ | $315.7_{\pm 24.0}$ |
| UTIGSP($\alpha$=1e-1) | $6.2_{\pm 3.2}$ | $13.0_{\pm 10.9}$ | $30.5_{\pm 2.4}$ | $74.3_{\pm 6.7}$ | $32.5_{\pm 7.0}$ | $57.5_{\pm 35.9}$ | $97.4_{\pm 9.8}$ | $317.5_{\pm 22.1}$ |
| UTIGSP($\alpha$=1e-2) | $2.6_{\pm 2.7}$ | $8.6_{\pm 9.7}$ | $30.4_{\pm 4.0}$ | $74.6_{\pm 7.3}$ | $17.9_{\pm 5.6}$ | $60.5_{\pm 27.1}$ | $85.9_{\pm 8.1}$ | $328.2_{\pm 20.1}$ |
| UTIGSP($\alpha$=1e-3) | $2.7_{\pm 2.2}$ | $9.3_{\pm 10.2}$ | $32.1_{\pm 3.0}$ | $78.1_{\pm 4.6}$ | $16.9_{\pm 6.5}$ | $70.2_{\pm 34.1}$ | $83.2_{\pm 8.6}$ | $341.4_{\pm 8.0}$ |
| UTIGSP($\alpha$=1e-5) | $4.3_{\pm 2.6}$ | $15.2_{\pm 11.5}$ | $31.5_{\pm 2.2}$ | $78.4_{\pm 8.0}$ | $17.0_{\pm 6.6}$ | $82.8_{\pm 37.4}$ | $82.2_{\pm 5.2}$ | $344.2_{\pm 14.1}$ |
| UTIGSP($\alpha$=1e-7) | $5.0_{\pm 3.9}$ | $18.2_{\pm 16.6}$ | $32.0_{\pm 2.8}$ | $77.1_{\pm 5.9}$ | $19.5_{\pm 6.9}$ | $89.7_{\pm 37.7}$ | $82.8_{\pm 4.9}$ | $346.0_{\pm 17.4}$ |
| UTIGSP($\alpha$=1e-9) | $6.0_{\pm 3.7}$ | $22.2_{\pm 18.0}$ | $31.7_{\pm 3.8}$ | $73.6_{\pm 7.1}$ | $18.8_{\pm 6.7}$ | $87.4_{\pm 41.2}$ | $81.4_{\pm 5.7}$ | $345.8_{\pm 15.4}$ |
| JCI-PC* | $5.7_{\pm 2.6}$ | $23.6_{\pm 13.2}$ | $35.9_{\pm 1.7}$ | $83.0_{\pm 6.5}$ | $13.1_{\pm 3.5}$ | $77.4_{\pm 22.2}$ | $76.2_{\pm 7.0}$ | $341.9_{\pm 22.5}$ |
| JCI-PC($\alpha$=2e-1) | $7.4_{\pm 2.1}$ | $28.4_{\pm 13.8}$ | $36.1_{\pm 1.8}$ | $83.2_{\pm 6.7}$ | $17.6_{\pm 4.2}$ | $84.9_{\pm 26.2}$ | $76.2_{\pm 7.0}$ | $341.9_{\pm 22.5}$ |
| JCI-PC($\alpha$=1e-1) | $6.9_{\pm 2.0}$ | $26.2_{\pm 13.0}$ | $36.1_{\pm 1.8}$ | $83.2_{\pm 6.7}$ | $15.2_{\pm 3.7}$ | $83.1_{\pm 25.3}$ | $76.2_{\pm 7.0}$ | $341.9_{\pm 22.5}$ |
| JCI-PC($\alpha$=1e-2) | $5.9_{\pm 2.3}$ | $23.6_{\pm 13.2}$ | $36.1_{\pm 1.8}$ | $83.2_{\pm 6.7}$ | $13.4_{\pm 3.4}$ | $79.0_{\pm 23.1}$ | $76.2_{\pm 7.0}$ | $341.9_{\pm 22.5}$ |
| JCI-PC($\alpha$=1e-3) | $5.7_{\pm 2.6}$ | $23.6_{\pm 13.2}$ | $36.1_{\pm 1.8}$ | $83.2_{\pm 6.7}$ | $13.1_{\pm 3.5}$ | $77.4_{\pm 22.2}$ | $76.2_{\pm 7.0}$ | $341.9_{\pm 22.5}$ |
| DCDI-G | $10.1_{\pm 4.2}$ | $12.4_{\pm 8.6}$ | $16.4_{\pm 5.3}$ | $52.3_{\pm 15.2}$ | $14.3_{\pm 18.8}$ | $23.3_{\pm 13.6}$ | $59.9_{\pm 10.5}$ | $237.6_{\pm 40.8}$ |
| DCDI-DSF | $4.4_{\pm 5.3}$ | $9.4_{\pm 9.4}$ | $9.3_{\pm 4.0}$ | $36.9_{\pm 11.9}$ | $4.9_{\pm 3.1}$ | $20.0_{\pm 12.0}$ | $32.5_{\pm 7.8}$ | $161.3_{\pm 37.1}$ |

Table 29: Results for the additive noise model data set with perfect intervention with unknown targets

| Method | 10 nodes, $e=1$ | | 10 nodes, $e=4$ | | 20 nodes, $e=1$ | | 20 nodes, $e=4$ | |
|---|---|---|---|---|---|---|---|---|
| | SHD | SID | SHD | SID | SHD | SID | SHD | SID |
| UTIGSP*-lin | $7.1_{\pm 2.3}$ | $20.5_{\pm 12.5}$ | $22.6_{\pm 3.0}$ | $59.2_{\pm 12.6}$ | $14.1_{\pm 4.8}$ | $56.8_{\pm 32.0}$ | $76.4_{\pm 5.7}$ | $312.5_{\pm 24.3}$ |
| UTIGSP* | $7.0_{\pm 4.3}$ | $20.6_{\pm 13.7}$ | $24.9_{\pm 2.3}$ | $70.8_{\pm 5.9}$ | $16.8_{\pm 7.0}$ | $87.1_{\pm 52.7}$ | $77.9_{\pm 6.6}$ | $333.4_{\pm 18.7}$ |
| UTIGSP($\alpha$=2e-1)-lin | $16.9_{\pm 4.1}$ | $24.2_{\pm 12.5}$ | $25.9_{\pm 5.0}$ | $66.5_{\pm 9.3}$ | $58.0_{\pm 10.8}$ | $73.7_{\pm 31.9}$ | $125.5_{\pm 11.0}$ | $275.8_{\pm 23.0}$ |
| UTIGSP($\alpha$=1e-1)-lin | $13.8_{\pm 6.0}$ | $20.8_{\pm 15.0}$ | $26.9_{\pm 4.1}$ | $67.1_{\pm 11.8}$ | $40.0_{\pm 11.7}$ | $67.0_{\pm 50.1}$ | $117.9_{\pm 6.3}$ | $290.7_{\pm 16.1}$ |
| UTIGSP($\alpha$=1e-2)-lin | $11.2_{\pm 4.3}$ | $25.2_{\pm 13.1}$ | $26.4_{\pm 4.6}$ | $66.5_{\pm 13.4}$ | $20.5_{\pm 10.5}$ | $54.8_{\pm 41.6}$ | $101.5_{\pm 7.6}$ | $298.6_{\pm 19.3}$ |
| UTIGSP($\alpha$=1e-3)-lin | $10.3_{\pm 3.7}$ | $28.1_{\pm 13.2}$ | $26.2_{\pm 3.6}$ | $64.6_{\pm 7.5}$ | $17.3_{\pm 7.2}$ | $47.6_{\pm 24.4}$ | $94.5_{\pm 7.9}$ | $306.8_{\pm 20.1}$ |
| UTIGSP($\alpha$=1e-5)-lin | $9.3_{\pm 2.5}$ | $27.4_{\pm 9.8}$ | $29.0_{\pm 3.7}$ | $73.0_{\pm 5.4}$ | $18.3_{\pm 6.9}$ | $73.0_{\pm 42.4}$ | $87.9_{\pm 7.8}$ | $325.2_{\pm 14.9}$ |
| UTIGSP($\alpha$=1e-7)-lin | $8.1_{\pm 2.1}$ | $24.9_{\pm 11.6}$ | $28.2_{\pm 3.7}$ | $72.4_{\pm 8.6}$ | $16.6_{\pm 5.7}$ | $65.8_{\pm 40.3}$ | $80.2_{\pm 8.4}$ | $316.4_{\pm 22.1}$ |
| UTIGSP($\alpha$=1e-9)-lin | $8.2_{\pm 2.8}$ | $27.5_{\pm 10.7}$ | $30.7_{\pm 3.9}$ | $76.7_{\pm 5.3}$ | $16.7_{\pm 5.9}$ | $70.2_{\pm 42.0}$ | $78.3_{\pm 4.0}$ | $318.9_{\pm 20.7}$ |
| UTIGSP($\alpha$=2e-1) | $13.5_{\pm 3.9}$ | $22.2_{\pm 17.2}$ | $27.6_{\pm 3.7}$ | $73.7_{\pm 3.5}$ | $45.6_{\pm 9.3}$ | $66.2_{\pm 43.7}$ | $98.6_{\pm 10.0}$ | $297.3_{\pm 36.4}$ |
| UTIGSP($\alpha$=1e-1) | $10.6_{\pm 6.1}$ | $20.1_{\pm 12.8}$ | $26.7_{\pm 2.9}$ | $71.9_{\pm 6.7}$ | $31.3_{\pm 5.3}$ | $68.3_{\pm 45.8}$ | $87.8_{\pm 10.0}$ | $301.0_{\pm 35.3}$ |
| UTIGSP($\alpha$=1e-2) | $9.1_{\pm 4.2}$ | $25.3_{\pm 10.3}$ | $29.0_{\pm 2.6}$ | $73.1_{\pm 3.1}$ | $20.8_{\pm 7.6}$ | $97.6_{\pm 53.0}$ | $84.4_{\pm 9.6}$ | $328.2_{\pm 17.4}$ |
| UTIGSP($\alpha$=1e-3) | $10.4_{\pm 4.1}$ | $28.1_{\pm 12.9}$ | $30.5_{\pm 4.7}$ | $77.8_{\pm 5.4}$ | $18.6_{\pm 7.0}$ | $84.5_{\pm 45.4}$ | $83.6_{\pm 5.3}$ | $335.0_{\pm 25.3}$ |
| UTIGSP($\alpha$=1e-5) | $9.9_{\pm 4.3}$ | $33.6_{\pm 12.0}$ | $32.1_{\pm 3.9}$ | $77.4_{\pm 6.7}$ | $19.5_{\pm 6.6}$ | $95.6_{\pm 50.9}$ | $81.9_{\pm 7.1}$ | $341.3_{\pm 12.1}$ |
| UTIGSP($\alpha$=1e-7) | $9.4_{\pm 4.9}$ | $33.3_{\pm 14.4}$ | $33.7_{\pm 3.9}$ | $76.8_{\pm 9.4}$ | $18.5_{\pm 6.9}$ | $92.3_{\pm 49.0}$ | $83.3_{\pm 8.1}$ | $337.5_{\pm 21.5}$ |
| UTIGSP($\alpha$=1e-9) | $9.4_{\pm 5.2}$ | $32.1_{\pm 15.2}$ | $33.0_{\pm 4.2}$ | $77.7_{\pm 8.7}$ | $18.7_{\pm 6.8}$ | $93.8_{\pm 52.0}$ | $82.9_{\pm 7.0}$ | $329.4_{\pm 28.2}$ |
| JCI-PC* | $8.5_{\pm 2.7}$ | $33.6_{\pm 12.0}$ | $35.5_{\pm 3.0}$ | $76.5_{\pm 8.7}$ | $15.2_{\pm 5.0}$ | $90.8_{\pm 52.1}$ | $72.4_{\pm 5.4}$ | $330.6_{\pm 12.8}$ |
| JCI-PC($\alpha$=2e-1) | $10.2_{\pm 3.3}$ | $35.8_{\pm 13.1}$ | $35.5_{\pm 3.0}$ | $75.6_{\pm 8.0}$ | $21.0_{\pm 3.6}$ | $92.0_{\pm 49.6}$ | $72.9_{\pm 5.4}$ | $328.7_{\pm 13.8}$ |
| JCI-PC($\alpha$=1e-1) | $9.5_{\pm 3.0}$ | $35.2_{\pm 12.9}$ | $35.5_{\pm 3.0}$ | $75.6_{\pm 8.0}$ | $17.5_{\pm 3.8}$ | $91.2_{\pm 51.2}$ | $72.9_{\pm 5.4}$ | $328.7_{\pm 13.8}$ |
| JCI-PC($\alpha$=1e-2) | $9.1_{\pm 3.0}$ | $35.4_{\pm 13.8}$ | $35.5_{\pm 3.0}$ | $75.6_{\pm 8.0}$ | $15.2_{\pm 5.0}$ | $90.8_{\pm 52.1}$ | $72.5_{\pm 5.4}$ | $330.5_{\pm 12.9}$ |
| JCI-PC($\alpha$=1e-3) | $8.6_{\pm 2.8}$ | $33.7_{\pm 12.1}$ | $35.5_{\pm 3.0}$ | $75.6_{\pm 8.0}$ | $15.2_{\pm 5.0}$ | $90.8_{\pm 52.1}$ | $72.4_{\pm 5.4}$ | $330.6_{\pm 12.8}$ |
| DCDI-G | $18.2_{\pm 10.1}$ | $16.4_{\pm 5.8}$ | $20.4_{\pm 6.8}$ | $64.8_{\pm 10.4}$ | $28.0_{\pm 33.5}$ | $39.1_{\pm 29.5}$ | $65.5_{\pm 11.6}$ | $249.8_{\pm 26.1}$ |
| DCDI-DSF | $10.6_{\pm 7.0}$ | $15.3_{\pm 10.5}$ | $9.1_{\pm 3.8}$ | $42.2_{\pm 12.4}$ | $28.0_{\pm 29.9}$ | $37.8_{\pm 22.6}$ | $42.4_{\pm 15.6}$ | $168.5_{\pm 37.8}$ |

Table 30: Results for the nonlinear with non-additive noise data set with perfect intervention with unknown targets

| Method | 10 nodes, $e=1$ | | 10 nodes, $e=4$ | | 20 nodes, $e=1$ | | 20 nodes, $e=4$ | |
|---|---|---|---|---|---|---|---|---|
| | SHD | SID | SHD | SID | SHD | SID | SHD | SID |
| UTIGSP*-lin | $3.6_{\pm 2.2}$ | $14.5_{\pm 11.1}$ | $23.1_{\pm 3.4}$ | $66.3_{\pm 6.4}$ | $13.7_{\pm 3.6}$ | $67.2_{\pm 28.8}$ | $68.0_{\pm 11.8}$ | $323.6_{\pm 15.7}$ |
| UTIGSP* | $4.1_{\pm 2.7}$ | $13.9_{\pm 9.5}$ | $24.2_{\pm 3.8}$ | $64.2_{\pm 11.1}$ | $17.8_{\pm 3.7}$ | $87.2_{\pm 25.8}$ | $73.4_{\pm 7.6}$ | $328.7_{\pm 24.9}$ |
| UTIGSP($\alpha$=2e-1)-lin | $11.3_{\pm 2.8}$ | $13.7_{\pm 6.9}$ | $24.7_{\pm 4.7}$ | $67.5_{\pm 7.4}$ | $50.2_{\pm 5.4}$ | $66.2_{\pm 29.4}$ | $104.3_{\pm 13.6}$ | $292.4_{\pm 18.5}$ |
| UTIGSP($\alpha$=1e-1)-lin | $8.5_{\pm 3.2}$ | $13.2_{\pm 10.3}$ | $27.0_{\pm 4.2}$ | $70.5_{\pm 6.3}$ | $36.7_{\pm 8.5}$ | $81.7_{\pm 38.1}$ | $91.7_{\pm 7.6}$ | $288.6_{\pm 20.4}$ |
| UTIGSP($\alpha$=1e-2)-lin | $6.6_{\pm 2.6}$ | $17.0_{\pm 9.9}$ | $27.4_{\pm 3.4}$ | $67.9_{\pm 8.7}$ | $18.3_{\pm 3.7}$ | $71.5_{\pm 37.0}$ | $77.6_{\pm 12.1}$ | $304.2_{\pm 22.4}$ |
| UTIGSP($\alpha$=1e-3)-lin | $4.4_{\pm 2.5}$ | $14.5_{\pm 10.8}$ | $27.8_{\pm 3.6}$ | $72.2_{\pm 6.0}$ | $16.1_{\pm 5.2}$ | $77.2_{\pm 38.4}$ | $72.2_{\pm 13.4}$ | $319.0_{\pm 16.9}$ |
| UTIGSP($\alpha$=1e-5)-lin | $6.3_{\pm 3.6}$ | $20.8_{\pm 14.8}$ | $28.7_{\pm 3.4}$ | $72.6_{\pm 5.7}$ | $15.7_{\pm 3.7}$ | $80.5_{\pm 19.0}$ | $71.5_{\pm 9.3}$ | $323.3_{\pm 17.0}$ |
| UTIGSP($\alpha$=1e-7)-lin | $5.7_{\pm 2.9}$ | $21.6_{\pm 15.3}$ | $29.9_{\pm 2.8}$ | $75.1_{\pm 5.4}$ | $15.7_{\pm 4.2}$ | $77.7_{\pm 25.7}$ | $73.0_{\pm 12.8}$ | $325.7_{\pm 17.8}$ |
| UTIGSP($\alpha$=1e-9)-lin | $5.3_{\pm 3.3}$ | $19.6_{\pm 15.3}$ | $30.2_{\pm 4.0}$ | $74.3_{\pm 8.6}$ | $17.1_{\pm 4.1}$ | $81.3_{\pm 28.2}$ | $76.2_{\pm 11.3}$ | $345.8_{\pm 17.9}$ |
| UTIGSP($\alpha$=2e-1) | $10.4_{\pm 4.4}$ | $12.4_{\pm 10.0}$ | $26.4_{\pm 4.4}$ | $69.3_{\pm 9.7}$ | $48.5_{\pm 7.8}$ | $93.9_{\pm 37.5}$ | $90.3_{\pm 15.5}$ | $306.8_{\pm 19.9}$ |
| UTIGSP($\alpha$=1e-1) | $8.1_{\pm 3.3}$ | $14.0_{\pm 7.6}$ | $26.9_{\pm 4.1}$ | $70.6_{\pm 6.8}$ | $35.6_{\pm 6.5}$ | $103.2_{\pm 28.8}$ | $86.5_{\pm 13.9}$ | $319.6_{\pm 28.1}$ |
| UTIGSP($\alpha$=1e-2) | $6.1_{\pm 4.1}$ | $16.6_{\pm 12.5}$ | $28.1_{\pm 4.8}$ | $68.4_{\pm 14.3}$ | $23.0_{\pm 5.7}$ | $107.1_{\pm 27.5}$ | $84.5_{\pm 8.9}$ | $327.3_{\pm 20.4}$ |
| UTIGSP($\alpha$=1e-3) | $6.4_{\pm 3.6}$ | $19.5_{\pm 14.5}$ | $31.0_{\pm 3.1}$ | $76.8_{\pm 4.3}$ | $20.6_{\pm 3.5}$ | $97.3_{\pm 20.8}$ | $81.1_{\pm 6.2}$ | $338.5_{\pm 10.8}$ |
| UTIGSP($\alpha$=1e-5) | $6.8_{\pm 3.5}$ | $21.1_{\pm 12.9}$ | $35.0_{\pm 2.2}$ | $80.6_{\pm 4.8}$ | $20.5_{\pm 4.2}$ | $95.8_{\pm 23.2}$ | $79.4_{\pm 8.8}$ | $338.1_{\pm 16.0}$ |
| UTIGSP($\alpha$=1e-7) | $6.2_{\pm 3.5}$ | $20.0_{\pm 11.5}$ | $32.5_{\pm 2.1}$ | $75.2_{\pm 9.9}$ | $20.0_{\pm 4.5}$ | $97.4_{\pm 22.2}$ | $78.8_{\pm 9.3}$ | $348.1_{\pm 12.2}$ |
| UTIGSP($\alpha$=1e-9) | $7.6_{\pm 3.8}$ | $22.3_{\pm 13.4}$ | $33.9_{\pm 2.0}$ | $78.6_{\pm 6.9}$ | $19.4_{\pm 3.9}$ | $94.3_{\pm 27.1}$ | $77.9_{\pm 7.5}$ | $342.3_{\pm 18.7}$ |
| JCI-PC* | $8.1_{\pm 2.6}$ | $26.7_{\pm 13.4}$ | $38.8_{\pm 1.9}$ | $80.8_{\pm 7.6}$ | $16.3_{\pm 3.5}$ | $89.8_{\pm 34.7}$ | $73.7_{\pm 7.7}$ | $335.8_{\pm 15.1}$ |
| JCI-PC($\alpha$=2e-1) | $10.5_{\pm 2.0}$ | $27.3_{\pm 14.3}$ | $39.2_{\pm 2.2}$ | $82.9_{\pm 6.6}$ | $23.4_{\pm 4.6}$ | $99.4_{\pm 34.8}$ | $73.8_{\pm 7.7}$ | $334.4_{\pm 18.4}$ |
| JCI-PC($\alpha$=1e-1) | $9.6_{\pm 2.0}$ | $27.8_{\pm 14.2}$ | $39.2_{\pm 2.2}$ | $82.9_{\pm 6.6}$ | $20.5_{\pm 3.9}$ | $100.0_{\pm 33.3}$ | $73.9_{\pm 7.7}$ | $336.2_{\pm 15.4}$ |
| JCI-PC($\alpha$=1e-2) | $8.2_{\pm 2.5}$ | $26.7_{\pm 13.4}$ | $39.4_{\pm 2.2}$ | $84.8_{\pm 4.6}$ | $16.8_{\pm 3.5}$ | $88.8_{\pm 36.2}$ | $74.0_{\pm 7.7}$ | $340.0_{\pm 14.3}$ |
| JCI-PC($\alpha$=1e-3) | $8.1_{\pm 2.6}$ | $26.7_{\pm 13.4}$ | $39.5_{\pm 2.1}$ | $84.9_{\pm 4.5}$ | $16.4_{\pm 3.6}$ | $90.9_{\pm 37.0}$ | $74.1_{\pm 7.8}$ | $340.1_{\pm 14.4}$ |
| DCDI-G | $6.6_{\pm 10.1}$ | $9.2_{\pm 9.4}$ | $8.5_{\pm 4.2}$ | $37.1_{\pm 15.3}$ | $16.5_{\pm 22.8}$ | $20.8_{\pm 10.5}$ | $35.4_{\pm 8.4}$ | $177.3_{\pm 38.8}$ |
| DCDI-DSF | $8.3_{\pm 11.4}$ | $12.1_{\pm 6.6}$ | $4.3_{\pm 2.6}$ | $28.6_{\pm 14.2}$ | $17.0_{\pm 13.5}$ | $52.6_{\pm 20.2}$ | $27.7_{\pm 10.0}$ | $126.9_{\pm 36.6}$ |