[Reviews · NeurIPS 2020]

Review 1

Summary and Contributions: The authors provide a score-based algorithm for learning causal graphs using interventional data. They first construct a score function, show that it achieves the optimum at an I-Markov equivalent graph. They later relax this score/loss to be able to differentiably optimize it using neural nets, using approaches from the existing literature.

Strengths: The paper proposes a score-based approach for learning causal graphs with interventions which is supported by theory. I think this is a very important direction.

Weaknesses: The unknown intervention section is unclear to me. It is not clear why the additional mask to learn intervention targets would work. For me, this section only makes the paper weaker. I would recommend the authors to only mention this in the experimental section since I do not think there is enough motivation to present it as a sound method.

Correctness: Yes, the proposed theory (Theorem 1) is correct. The methodology of Section 3.3 about unknown interventions, however, is not clear.

Clarity: Yes, the paper is mostly well written.

Relation to Prior Work: Related work section is missing some references. The authors cite "Joint causal inference from multiple contexts" by Mooij et al. in bibliography but not in the text of the paper. Another related work that is not included is the "Characterization and learning of causal graphs with latent variables from soft interventions" by Kocaoglu et al.

Reproducibility: Yes

Additional Feedback: POST-REBUTAL FEEDBACK Thank you for your response. I do think it's a good idea to add the hinted theorem on operating under unknown intervention targets. I recommend the authors to emphasize the assumptions even more for this one, i.e., causal sufficiency and Assumption 2, which could make this possible. Clearly, there are some proof details the authors will figure out, but the proof sketch seems reasonable and I will be excited to read once they are available in camera-ready. Thank you for the good work. Authors show how they use NNs to model the conditionals in (7). It might be better to explicitly state Assumption 1 here, which restricts the considered interventions in this way. For I-Markov, please cite related work in the main text. There are multiple definitions with the same name in the literature, which can be confusing. I believe the finite entropy assumption should be decoupled from Assumption 1, which is currently described only as "sufficient capacity neural nets" assumption. It might be good to provide intuition on why this is necessary. Assumption 2 seems tied to the intervention targets, which could have created problems with perfect interventions. However, so long as obs. data is assumed to be available, this is simply the original faithfulness assumption. This point could be emphasized. line 575: "This means that KL>0". For clarity, please consider adding the argument that differing CI statement means we cannot fit the exact distribution and KL=0 iff dist's are the same. I want to say that the argument for Case 5 is very well done. Proof of Theorem 1 looks correct. The proposed approach in Section 3.3 seems a bit arbitrary. Moreover, there are a lot of works that could be used even when intervention targets are unknown. Please see Mooij et al. "Joint causal inference" Table 4 for these. The experiments seem okay, but definitely the authors could have provided comparisons with the other existing work, as detailed in the paper by Mooij et al. mentioned above. Minor comments: line 116: the use a->the use of a in (7), f is used for density whereas p is used in (8). line 564: their immoralities implying ->their immoralities including


Review 2

Summary and Contributions: This paper proposes a neural-network-based method for causal discovery that can leverage interventional data. - A causal discovery method using continuous constrained optimization that could utilize interventional data. - A score for interventional data with theoretical justification of its validity - Applicable to both imperfect and perfect intervention. - provide theoretical results of the method's identifiability

Strengths: - This work is of good novelty. It proposes the first causal discovery method using continuous constrained optimization considering interventional data. - From the theoretical aspects, it provides sufficient theoretical results to justify the validity of the score used in the proposed method. - In the empirical evaluation, it conducted extensive synthetic experiments which provide comprehensive understanding of the properties in various factors (e.g., type of interventions, graph size, density, type of mechanisms) of the proposed method.

Weaknesses: - Assumptions used in theorem 1 are not included in the main text. As manuscripts should be as self-contained as possible, it may be better to move assumptions 1 and 2 from Appendix A.2 to the main text. Furthermore, if possible, it would be better to extend intuitive explanations of these assumptions to help readers have an idea of their limitations. - The idea of how to prove theorem 1 is not mentioned. Similar to the comment above, a general description of the proof (e.g., a proof sketch) may help readers accept the theorem more easily. It would be better to give a brief description in the main text. - The empirical results can be further analyzed. For example, what contribute to the good performance of DCDI in cases with higher number of average edges?

Correctness: Both the theoretical result and empirical methodology are technically correct.

Clarity: This paper is well written. It gives sufficient background knowledge in the Appendix and adds references of the Appendix in the main text.

Relation to Prior Work: Yes, previous works are carefully reviewed and the references are sufficient. It is a meaningful extension of existing works using continuous constrained optimization.

Reproducibility: Yes

Additional Feedback: - The reference of RMSprop in line 189 is missing.


Review 3

Summary and Contributions: This paper works on causal discovery in the presence of interventional data. The proposed algorithm is in line with the recent works of differentiable score-based learning. The score is a penalized interventional log-likelihood of the data, where the likelihood is either 1) a Gaussian parameterized by nonlinear functions of parents; 2) a nonparametric tractable density modeled by DSF. Theorem 1 gives justification for this kind of scores. The DAG constraint is enforced by introducing a stochastic masking matrix that is learned with Monte Carlo gradient estimates. Furthermore, it also introduces a way to deal with unknown interventions by estimating the binary intervention target matrix in a similar stochastic manner. ---------------- Thanks for the clarifications. Hopefully this can be incorporated into the final version.

Strengths: - To my knowledge first differentiable causal discovery method for interventional data. - Extensive experiments and detailed description.

Weaknesses: - Algorithm idea is not new.

Correctness: Did not check the proof for Theorem 1. Methodology seems correct.

Clarity: Yes

Relation to Prior Work: Yes

Reproducibility: Yes

Additional Feedback: The algorithm itself is perhaps less exciting since most of the techniques (differentiable causal discovery, neural network likelihood models, stochastic mask and gradient estimates) have existed in the literature. However, the paper did a great job executing these ideas and recording all the details. Additional questions: - Would it be interesting to try linear Gaussian likelihood against linear Gaussian model just for sanity check? - It seems that on ANM data DCDI-G should in principal outperform DCDI-DSF. However it is not always the case. Is there an explanation for this?


Review 4

Summary and Contributions: The authors extend a continuous optimization technique for causal structure discovery to include a combination of observational and perfect and imperfect interventional data, rather than just observational data.

Strengths: The paper addresses an important problem, and builds off of recent advances using continuous optimization for causal structure discovery. The authors provide a thorough description of necessary background and related work on approaches for causal structure discovery with interventional data.

Weaknesses: The paper has two major weaknesses, (1) the description of the methodology lacks clarity and (2) the empirical results do not provide compelling evidence that the methodology is effective. Methodology: The paper does not clearly explain what I understand as a central methodological contribution, section 3.1. Several questions remain unanswered. 1. Why is each intervention set parameterized by independent sets of neural network weights? 2. How is background knowledge about the intervention assignment incorporated into the likelihood function? For example, with an encouragement design we know that a random variables distributions will place higher density on larger values than the observational distribution. 3. What is the score function when interventions are perfect? It is not enough to say that “the idea is simple” without providing a mathematical expression for the likelihood. Empirical Results: In addition to the structural measures you report, it is important to report on some metric of the accuracy of effect estimates for some previously unseen set of interventions. See (Gentzel et al., NeurIPS 2019) for a discussion of using interventional distributions to evaluate causal discovery algorithms. This should be straightforward, as DCDI jointly learns structure and conditional probability distributions. I strongly disagree with moving the cytometry evaluation to the appendix. When making parametric and semi-parametric assumptions (such as a particular NN architecture), it is important to understand the algorithms’ performance when those assumptions are violated. The synthetic experiments provide useful “knobs” to twist, but they do not provide insight into the key question, “how will this work in the real world?” The visual presentation of the results makes it difficult to determine where DCDI outperforms the alternative approaches. See the additional feedback below for additional details.

Correctness: Given that the empirical evaluation does not provide metrics of interventional distributions and focusses on synthetic data, it is difficult to determine whether the claims and methods are correct. On the real-world dataset, the empirical results appear to be inconclusive. As the authors note, CAM performs better than DCDI on several metrics.

Clarity: The paper is mostly well written, although some additional editorial review would have strengthened the submission. See the additional feedback below.

Relation to Prior Work: Yes, the authors appear to have touched on the relevant areas of prior work. In particular, prior approaches either (1) don’t account for interventional data or (2) don’t use a continuous-optimization formulation of the structure discovery problem.

Reproducibility: Yes

Additional Feedback: Line 36: “Constrained-based” -> “constraint-based” Line 63: we call interventional target -> we call the interventional target Line 82: “identifiable (more severe without interventional data)” -> “identifiable, which is more severe without interventional data.” Line 98: “space of DAGs is enormous” -> “space of DAGS is super-exponential in the number of variables” Line 112: “which serves as basis “ -> “which serves as the basis” Line 157: “Intuitively, this score favors graphs in which a conditional p(x_j|_{\pi_j}G) is invariant across all interventional distributions in which x_j is not a target, i.e. j \not \in I_k.” -> This expression appears to one of the central claims in the paper. As written the paper does not clarify how this is true, or why it is important. Line 246: “although performance is sensible to this hyperparameter” -> “although performance is sensitive to this hyperparameter” Line 257: “how two DAGs differ with respect to their causal inference statements” -> This needs to be made more precise. Figure 2, 3, 4 Captions: This will be clearer at first glance if you spell out Structural Hamming Distance and Structural Interventional Distance. Section 3.3: Please clarify exactly what is meant by “unknown interventions”. Is the set of random variables intervened upon unknown? The intervention assignment? Whether it is a perfect or imperfect intervention? Equation 8: My understanding is that the \lambda |G| term induces an acyclicity constraint. Is there any reason not to include any other regularization terms? This seems important when each interventional dataset is relatively small. POST REBUTTAL FEEDBACK: Based on the authors' response I have increased my score from a 4 to a 6. The authors' response effectively addresses my concerns about clarity with a thorough discussion of how background knowledge is incorporated and some more intuition about the score for unknown targets. These concerns are now completely addressed. Thank you! Regarding empirical evaluation: I am very glad to see that the authors have included an additional experiment evaluating DCDI's ability to produce accurate interventional distributions. However, I still would have liked to see more emphasis placed on real or semi-synthetic benchmarks.

[Author Response · NeurIPS 2020]

We thank the reviewers for their time and detailed reviews. We are happy to integrate their comments in our revision.

**R4 How is background knowledge about the intervention assignment incorporated into the likelihood function?**
In Sections 3.1 & 3.2, the *only* information about interventions that we make use of is the data itself and the set of
targeted variables $I_k$. The conditionals of targeted variables, parameterized by a separate $\phi^{(k)}$, are learned from scratch.
Exploiting domain expertise about the nature of interventions is an interesting direction left as future work.

**R4 Invariance in score - "L157 [...] clarify how this is true, or why it is important."** Inspection of score (8) and
model (7) should make clear that learned conditionals are assumed to be invariant across interventional distributions *in*
*which they are not targeted*. Thus, its maximization will favor graphs for which this invariance holds in the data, which
is a central property of causal graphs (see definition of intervention (2) and Peters et al., 2017 Sec 2.1).

**R2 Proof sketch and assumptions explanations.** This would strengthen the paper and we will add it to the main text.

**R4 Perfect interventions.** To obtain the score for such interventions, the conditional densities of targeted nodes are
removed from the likelihood. This is justified at L171-175, but we will add a formal explanation in the appendix. The
essence is that the score (8) separates into two pieces, one of which does not depend on $\mathcal{G}$ and can thus be removed.

**R4 Clarification of the $\lambda|\mathcal{G}|$ term.** This term is not an acyclicity constraint, it serves to encourage graph sparsity.

**R1 R4 Clarification and justification of the score for unknown targets.** In Section 3.3, the phrase "unknown
intervention" should be replaced by "unknown targets", which makes more explicit what is actually unknown. In this
setting, the interventional targets $I_k$ are unknown and must be learned (except for $I_1$ which is known to be observational;
to be clarified in main text). The approach of Section 3.3 can be seen as a relaxed maximization of the score

$$\mathcal{S}(\mathcal{G},\mathcal{I}) := \max_\phi \sum_{k=1}^K \mathbb{E}_{X \sim p^{(k)}} \log f^{(k)}(X; M^\mathcal{G}, R^\mathcal{I}, \phi) - \lambda|\mathcal{G}| - \lambda_R|\mathcal{I}|, \tag{1}$$

where $|\mathcal{I}|$ is the total number of targeted nodes in $\mathcal{I}$, $R^\mathcal{I} \in \{0,1\}^{K \times d}$ is such that $R^\mathcal{I}_{kj} = 1 \iff j \in I_k$ and
$f^{(k)}(X; M^\mathcal{G}, R^\mathcal{I}, \phi)$ is defined in Section 3.3. **If reviewers agree**, we could include the following result to justify (1).

▶ *Extension of Theorem 1 to unknown targets.* Let $\mathcal{I}^*$ be the ground truth intervention family. Under assumptions
identical to Theorem 1, we showed that, for $\lambda > 0$ and $\lambda_R > 0$ small enough, if $(\hat{\mathcal{G}}, \hat{\mathcal{I}})$ maximizes score (1), then
$\hat{\mathcal{G}} \in \mathcal{I}^*\text{-MEC}(\mathcal{G}^*)$ and $\hat{\mathcal{I}} = \mathcal{I}^*$. The idea is to add two steps at the beginning of the proof of Theorem 1 showing
$\mathcal{I} \neq \mathcal{I}^* \implies \mathcal{S}(\mathcal{G}^*, \mathcal{I}^*) > \mathcal{S}(\mathcal{G}, \mathcal{I})$. The argument is similar to Case 5 & 6, except we need to make sure $\lambda$ and $\lambda_R$ are
small enough (via an argument similar to Case 1). Then, we can resume to the proof of Theorem 1 assuming $\mathcal{I} = \mathcal{I}^*$.

**R4 Evaluation on unseen interventional distributions.**
Such evaluations are uncommon in the closely related
causal discovery literature, since some algorithms do not
even model distributions. However, we agree that this
has scientific value and thus added such an experiment.

Hence, we learn a graph using each method and fit a distribution to each graph using a normalizing flow model, enabling
a fair comparison. We report the log likelihood evaluated on an *unseen* intervention. For the NN data set, we report
boxplots over 10 graphs. DCDI-G and DCDI-DSF have the best performance like for the structural metrics. For Sachs,
we report the log-likelihood and its standard deviation (over data samples). The ordering of the methods is different
from the structural metrics: IGSP has the best performance followed by DCDI-G. This will be added to the appendix.

**R3 Performance of DCDI-G on the ANM data sets. R3** correctly notes that DCDI-G should outperform DCDI-DSF
on ANM data since it has the right inductive bias. We believe that this is not always noticeable in our results due to the
relatively high sample size ($n = 10^4$). Hence, we performed a small scale experiment on data sets with a smaller sample
size ($n = 10^3$) for perfect interventions. The results support our hypothesis, DCDI-G does outperform DCDI-DSF:

(sparse graph) **DCDI-G** SHD: $5.2 \pm 2.5$, SID: $20.0 \pm 20.8$ **DCDI-DSF** SHD: $45.2 \pm 12.7$, SID: $20.8 \pm 9.4$
(dense graph) **DCDI-G** SHD: $23.6 \pm 9.4$, SID: $127.2 \pm 37.8$ **DCDI-DSF** SHD: $37.1 \pm 10.7$, SID: $125.9 \pm 44.9$

**R3 Linear DCDI as a sanity check.** We trained DCDI as a linear Gaussian model (i.e. no hidden layer in NN). As
expected, this version of DCDI obtained competitive results for the linear data set, but poorer results on nonlinear data
sets, showing the interest of using high capacity models. We will include this in the appendix.

**R2 What contributes to the good performance of DCDI in cases with higher number of average edges?** While
we do not have a definitive explanation for this, it might be that continuous search has an advantage over discrete greedy
search in this setting. This trend has also been noted by Zheng et al. (2018, Section 5.2) (DAGs with NOTEARS).

**R1 Additional unknown target methods.** While we had seen the list of methods compiled by Mooij et al. (2020),
many methods did not have an implementation available or addressed a different setting. Recently (May 2020), the code
for Joint Causal Inference (JCI) has been released. We plan to a add a comparison to JCI to the camera-ready version.

[Meta-Review · NeurIPS 2020]

The reviewers appreciated the methodological and theoretical contribution of this paper and voted to accept. I strongly encourage the authors to take the reviewer comments and incorporate them into the final draft of the paper.